# Systematic review and meta-analysis of the evidence for an illusory truth effect and its determinants

Steeven Ye[1,2], David Attali [3,4], Maria Ghazi [1,5], Arnaud Cachia[1,6], Mathieu Cassotti[1,7,9,10] & Grégoire Borst [1,7,8,9,10] ✉

Over the last four decades, studies provided evidence that individuals tend to rate statements as being more truthful when they are re-exposed to them, the so-called 'illusory truth effect'. In light of a growing number of studies published since the previous meta-analysis in 2006 and concern of publishing biases, we conduct a meta-analysis on 182 studies and 366 effect sizes (N = 31,184 participants) published from 1977 to 2025. After correcting for small-study effects, we observe a small illusory truth effect (g = 0.37, 95% confidence interval [0.30, 0.44]), with a substantial within and between-study heterogeneity. Here, we show that multiple variables accounted for such heterogeneity, including the type of item, the instructions during the first exposure, the presence of veracity cues, and the duration of presentation on first exposure to the statement. We highlight the importance of the initial exposure and discuss practical implications regarding the current misinformation crisis.

The illusory truth effect, or repetition-induced truth effect refers to the tendency for individuals to judge repeated statements as more true than novel statements[1]. In a typical experiment, participants are first exposed to a set of ambiguous statements (i.e., statements for which participants are unlikely to know the actual truth status, as established by pretests) or neutral statements (i.e., statements equally likely to be rated as true in both their true and false version) during the exposure phase. At the test phase, participants are instructed to judge the truth of another set of statements, usually consisting of statements repeated from the exposure phase and new statements previously unseen in a random order. The illusory truth effect is typically operationalized as the difference in truth ratings between repeated and novel items.

While the design of primary studies varies considerably, the illusory truth effect appears robust across populations and types of content[2,3]. The effect is observed regardless of the population type[4,5]

(clinical vs typical), or of age[6,7] (from 5 to 80 years old). Additionally, the illusory truth effect seems to appear regardless of the source of the information[8], or whether it is a trivia statement[9,10], a general knowledge question[7,11], a (fake) news headline[12,13], or less-tangible information such as a rumor[14], an opinion in an argument[15], or a marketing claim[16]. However, the effect of repetition on perceived truth was not consistently observed. For instance, repetition did not reliably increase the perceived truth for extremely implausible statements[12,17], or headlines in the form of questions[18].

The rise of digital media and the rapid pace of online communication have amplified concerns about how repetition might increase belief in misinformation[19]. Repeated exposure to false news headlines not only increases perceived accuracy but can also raise the intention to share such content on social media[20]. While individual factors such as motivation, cognitive abilities, thinking style, or knowledge

[1]Université Paris Cité, LaPsyDÉ, CNRS, Paris, France. [2]Global Science Organization, IPSOS, Paris, France. [3]GHU Paris Psychiatrie et Neurosciences, Hôpital Sainte-Anne, Unité de Neuropsychiatrie interventionnelle, Paris, France. [4]Institute Physics for Medicine Paris, Inserm U1273, CNRS UMR 8063, ESPCI Paris, PSL University, Paris, France. [5]Editions Nathan, Paris, France. [6]Institute of Psychiatry and Neuroscience of Paris (IPNP), INSERM U1266, Imaging biomarkers for brain development and disorders, Université Paris Cité, Paris, France. [7]Institut Universitaire de France, Paris, France. [8]Laboratory for Interdisciplinary Evaluation of Public Policies (LIEPP), Sciences Po, Paris, France. [9]These authors contributed equally: Mathieu Cassotti, Grégoire Borst. [10]These authors jointly supervised this work: Mathieu Cassotti, Grégoire Borst. ✉e-mail: gregoire.borst@u-paris.fr

influence belief in misinformation[21,22], these factors appear to have little impact on the illusory truth effect. The effect is robust across individual differences in cognitive abilities[10] and persists even when participants are incentivized for accuracy[23], or possess relevant factual knowledge[7,24]. Several intervention strategies have been tested to reduce or eliminate the illusory truth effect. These include warning participants about the effect itself[25,26], alerting them prior to initial exposure that some of the information might be false[27], or asking them to rate the accuracy of information during initial exposure[11,28]. However, other approaches have produced mixed results. While labeling false information as disputed by a third-party fact-checker had little impact, more explicit labels (e.g., false) successfully eliminated the effect[29]. Similarly, the benefit of labeling source reliability may depend on whether participants recall this information during judgment[8,30,31]. Given that false information can spread more widely and rapidly than true information online[32], understanding the determinants of the illusory truth effect has become increasingly urgent. Such insights could clarify the conditions under which repetition amplifies perceived truth, thereby guiding the design of targeted interventions to reduce the risk of increasing gullibility to misinformation when people are massively and repeatedly exposed to them.

Previous research has proposed several theoretical frameworks to explain why repetition increases perceived truth of information[33]. Dominant explanations of the illusory truth effect attribute a significant role to processing fluency, which is defined as the perceived ease with which information is processed[34]. According to this view, repeated exposure to information increases processing fluency, which in turn increases the perceived truth. Supporting this framework, prior research has shown that the illusory truth effect can be reversed when participants form an implicit association between fluency and falseness[35]. While fluency can be a legitimate indicator of truth when it aligns with valid cues[36,37], the illusory truth effect arises when people misattribute incidental increases in fluency from mere repetition as evidence of truth.

An emerging alternative, non-exclusive explanation is the referential theory. This framework argues that truth judgments are driven by the presence and coherence of corresponding references in an individual's semantic memory[38]. When processing a statement, individuals rely on pre-existing memory nodes – concepts and associations linked to the statement's referents. According to this theory, repeated exposure to a statement activates these memory nodes, but also strengthens the connections between them, leading to a more integrated and coherent network. This coherence makes repeated statements easier to retrieve and process, thereby enhancing their perceived truthfulness in later evaluations. Udry and Barber[39] argue that the corresponding references must also be semantically coherent across the repetitions (i.e., repeated statements must convey the same meaning to exhibit the illusory truth effect). For instance, even if repeated statements share some of the referents, they will not induce the illusory truth effect if these referents are associated in semantically inconsistent ways. This suggests that both fluency and strengthened connections between concepts alone are insufficient to increase the perceived truth of information without maintaining semantic consistency. Dechene et al.'s meta-analysis[2] of 51 studies published up to 2006 and reporting 102 effect sizes revealed a small ($d = 0.39$, 95% confidence interval [0.32, 0.47]) to medium ($d = 0.50$, 95% confidence interval [0.43, 0.57]) illusory truth effect across studies depending on the type of analysis (i.e., fixed-effects model or random-effects model) and the method used to compute the effect (i.e., within-item or between-item criterion). In particular, the illusory truth effect seems to be smaller when the truthfulness ratings for the same items are compared between the exposure and the test phases (within-item criterion, $d = 0.39$) compared to when repeated items are compared with new items in the test phase (between-item criterion, $d = 0.53$).

This first meta-analysis on the illusory truth effect provided a valuable foundation by synthesizing early evidence for the phenomenon. Since then, the field has expanded rapidly, with increasing diversity in study designs, populations, and materials, warranting an updated and more comprehensive synthesis. At the same time, concerns have been raised about the evidential quality of the literature: a recent review found that 96% of studies on the illusory truth effect reported statistically significant results, raising the possibility of publication bias[3].

The present meta-analysis was designed to address these developments through four core objectives. First, to provide an updated and comprehensive estimate of the illusory truth effect by synthesizing the greatly expanded body of research with 182 studies and 366 effect sizes published between 1977 and 2025. Second, to improve the estimation of the effect size of the illusory truth effect and the potential moderators of such effect by applying more advanced techniques than the ones available when the previous meta-analysis was conducted. These include three-level random-effects models to account for the non-independence of multiple estimates from the same sample[40,41], publication bias diagnostics[42], risk of bias assessments (RoB2[43]), multivariate imputation for missing moderator data, and sensitivity analyses to evaluate the impact of missing outcome data. Third, to examine a broader set of moderators to investigate the boundary conditions of the illusory truth effect (see Table 1). Moderator selection prioritized variables with practical relevance (e.g., misinformation susceptibility) and empirical prevalence. The results are discussed within the processing fluency account and the referential theory of the truth effect, while also considering alternative explanations for moderators not directly addressed by these frameworks. Fourth, drawing from this moderator analysis, to identify those with direct implications for misinformation mitigation, including the design of effective interventions.

In this work, we confirm that repeated exposure produces a small but robust increase in perceived truth (bias-corrected $g = 0.37$, 95% CI [0.30, 0.44]). The effect is systematically moderated by the nature of the item type and the design of the initial exposure, particularly the initial task, the presentation duration, and the presence of explicit veracity cues. The meta-regression indicates that the combined moderators account for approximately 37% of the between-study variance, highlighting their explanatory value. Together, these findings suggest the importance of designing interventions that target the initial moments of exposure. Encouraging early truth evaluation or embedding explicit accuracy signals during initial encounters with information may significantly reduce the later influence of repetition.

## Results

### Study Selection and Study Characteristics

The current meta-analysis included $n = 99$ articles describing $k = 182$ studies from which $u = 366$ effect sizes were extracted from 31,184 participants. Of the 93 articles identified in Henderson et al.[3], 27 were excluded according to our inclusion and exclusion criteria. Compared to a previous meta-analysis conducted by Dechêne et al.[2], 6 out of 25 articles were excluded. Included articles were published between 1977 and 2025, with a median year of 2019. Most studies were published in a journal ($n = 77$) while other were unpublished studies ($n = 9$) or data set ($n = 1$), PhD ($n = 6$) or Master's thesis ($n = 5$) and conference presentations ($n = 1$). Most studies were conducted in Europe ($n = 49$) and in North America ($n = 47$), with a few in Oceania ($n = 3$), Asia ($n = 1$), and South America ($n = 1$). Characteristics and references for all studies included in the meta-analysis are provided in the Supplementary Data 1 and the Supplementary References.

### Publication Bias

Visual inspection of the contour-enhanced funnel plot[42] revealed asymmetry in the distribution of effect sizes (see Fig. 1). Most

**Table 1 | Description of moderators included in the meta-analysis, with definitions, coding scheme, and conceptual rationale**

| Moderator | Definition | Coding Scheme | Conceptual Rationale |
|---|---|---|---|
| Item type. | The format of the statements used. | Standard statements, headlines, claims. | Tests the generalizability of the illusory truth effect across different content formats. |
| Repetition study design. | The experimental design used to manipulate repetition. | Within-participant, between-participant. | Clarifies whether awareness of repetition influences subsequent truth judgments. |
| Instructions during the exposure phase. | The task participants completed during the exposure phase. | Truth judgment, irrelevant task, passive processing. | Evaluates how levels of cognitive engagement during encoding influence susceptibility to repetition. |
| Truth warning at the exposure phase. | The presence of explicit instructional warning before first exposure that some statements can be false. | Warning, no warning. | Determines whether warnings before first exposure attenuate the illusory truth effect. |
| Truth warning at the test phase. | The presence of explicit instructional warning before the test phase that some statements can be false. | Warning, no warning. | Clarifies whether the timing of warnings (exposure vs. test) affects belief correction. |
| Delay between exposure and test phases. | The time interval between the exposure and test phases. | No delay, within day, up to one week, more than one week. | Evaluates whether the illusory truth effect decays across time intervals. |
| Veracity cues. | The valence of indicator of factuality during exposure. | No cue, true cues, false cues. | Examines whether veracity cues mitigate the illusory truth effect. |
| Repetition type. | The nature of repetition between exposure and test phases. | Verbatim, gist. | Tests whether the effect extends to semantically coherent but non-verbatim repetitions (gist). |
| Modality of presentation. | The sensory channel through which statements were presented. | Visual, verbal, both. | Evaluates the robustness of the effect across sensory channels and media contexts. |
| Presentation time during exposure. | The time duration of exposure to each statement at the exposure phase. | 5 seconds or less, more than 5 seconds, participant-paced. | Investigates whether exposure duration influences the illusory truth effect (e.g., fast vs. extended processing). |
| Presentation time during test. | The time duration allowed for participants to perform truth judgment for each statement at test phase. | 5 seconds or less, more than 5 seconds, participant-paced. | Examines whether response time constraints affect the illusory truth effect. |
| Response scale | The format of the scale used for truth judgments in the test phase. | Binary, Likert, 100-point slider. | Tests how response influences measured effect sizes. |
| | | Even, odd. | |
| Proportion of true statements. | The proportion of factually true items in the stimulus pool. | Continuous variable from 0 to 1. | Clarifies whether varying base rates of truth moderate susceptibility to repetition. |
| Veracity. | The factual status of the item. | True, false, none | Examines whether repetition biases judgments equally for true, false, and unverifiable statements. |
| Testing environment. | The context in which data collection occurred. | Online, face-to-face. | Evaluates whether laboratory findings generalize to online settings where misinformation is prevalent. |

imprecise studies (lower part of the plot) fall in the $p < 0.05$ regions to the right of zero, whereas few comparable studies appear in the adjacent above the conventional 0.05% area. This pattern suggests that non-significant or negative small-study results are underrepresented. Additionally, most highly precise estimates at the top of the funnel cluster around the bias-adjusted effect size, whereas the unadjusted pooled effect estimate falls farther to the right, implying inflation by small-study effects. This observation was confirmed using an Egger's regression test[44] accounting for dependencies across effect sizes[45]. Results indicate asymmetry in the funnel plot, $t(364) = 9.77$, $p < 0.001$, which suggests the presence of small-study effects, one possible component of publication bias[46].

Following the evidence of funnel-plot asymmetry, we implemented a Precision-Effect Estimate with Standard Error (PEESE) correction to obtain a bias-adjusted overall effect. Furthermore, this adjusted model served as the baseline for all subsequent moderator analyses to support an unbiased interpretation of moderator effects.

**Overall Effect Size and Sensitivity Analysis**
The three-level meta-analysis of the illusory truth effect for 366 effect sizes retrieved from 182 studies described in 99 unique articles yielded an estimated small to medium size effect. The PEESE-corrected overall effect size ranged from $g = 0.35$, 95% CI [0.29, 0.42] to $g = 0.44$, 95% CI [0.37, 0.51] depending on the values imputed for missing standard deviations ($u = 136$) and missing correlation coefficients ($u = 282$). This interval was obtained from a sensitivity analysis using the minimum

and maximum estimated values (i.e., minimum/maximum of previously observed values) for imputed standard deviations and correlation coefficients. When the median estimated values were taken, we found an effect size $g = 0.37$, 95% CI [0.30, 0.44] (see Table 2).

Additionally, a substantial variability was found both for effect sizes extracted from the same study (i.e., level 2 within-study heterogeneity) and from different studies (i.e., level 3 between-study heterogeneity). Depending on the computed missing values, the estimated variance components varied for $\tau^2_{Level\ 2} = 0.054$ to 0.101 and $\tau^2_{Level\ 3} = 0.099$ to 0.148. The related I² value indicated that 26% to 36% of the total variance could be attributed to within-study differences in effect sizes, and that between-study heterogeneity explained I² = 36% to 70% of the total variation. We found that the three-level model provided a significantly better fit compared to a two-level model with level 3 heterogeneity constrained to zero ($\chi^2(1) = 32.65$; $p < 0.001$).

The strong within and between-study heterogeneity, $Q(364) = 4445$, $p < 0.001$, calls for caution in interpreting the estimated overall effect size and suggests that differences in the methods used in the different studies may be at the root of the observed heterogeneity (see the meta-regression below for such analysis).

To investigate whether heterogeneity in standard deviations contributed to variation in effect sizes, we conducted a three-level meta-regression using the coefficient of variation (CV) as a moderator. This analysis revealed a significant negative relationship between CV and the standardized mean differences, $b = 0.53$, 95% CI [0.39, 0.58], $t(330) = -4.30$, $p < 0.001$, indicating that effect sizes were smaller in

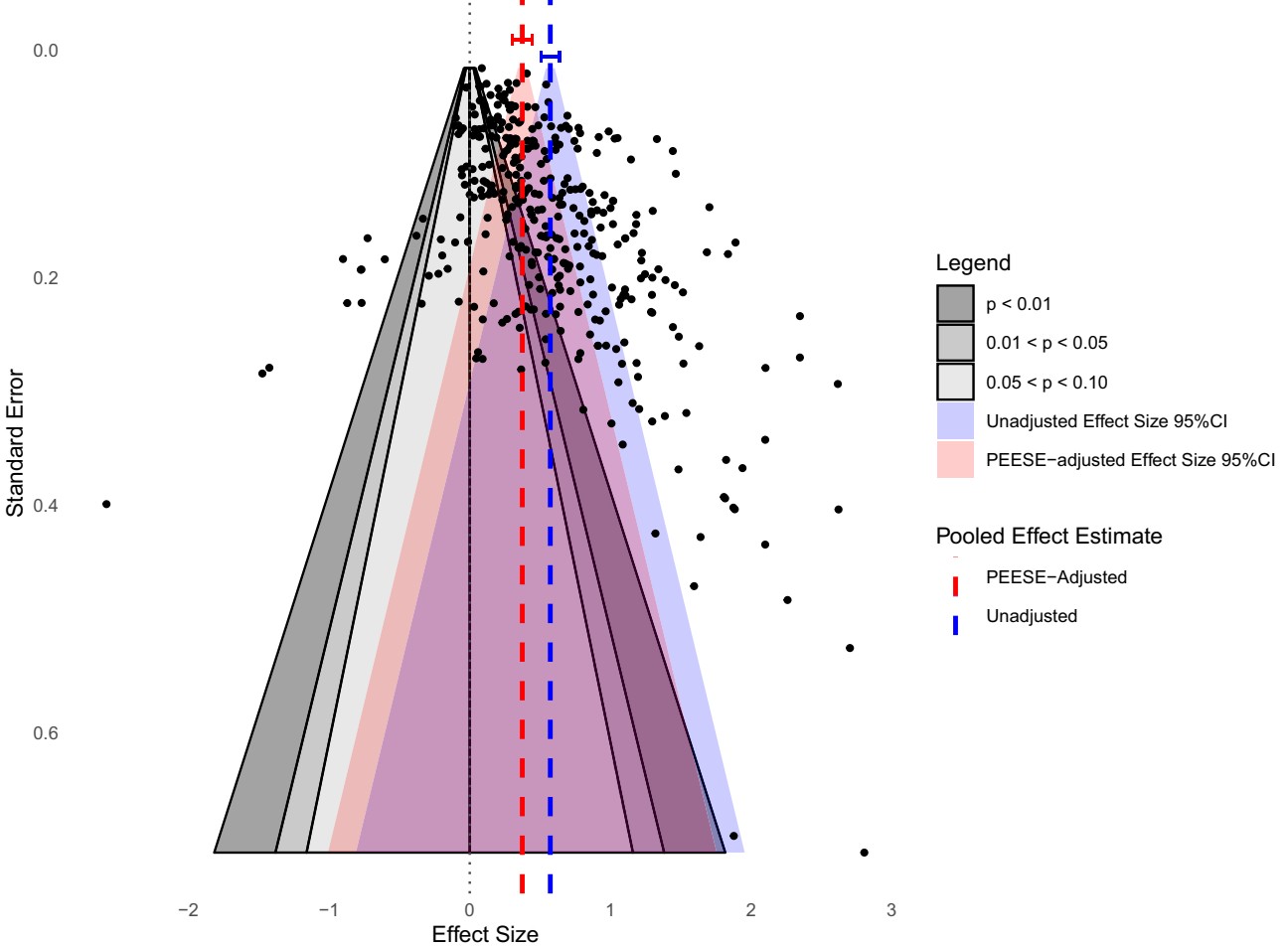

**Fig. 1 | Contour-enhanced funnel plot to visualize publication bias in the illusory truth effect literature.** Each point represents a single effect size estimate. Contour shading reflects two-sided statistical significance thresholds based on null-centered z-tests ($p < 0.10$, $p < 0.05$, $p < 0.01$). Dashed vertical lines mark the pooled point estimates from the unadjusted (blue) and Precision-Effect Estimate with Standard Error (PEESE; red)-adjusted models, with their 95% confidence intervals (CI) regions. No adjustment for multiple comparisons was applied.

**Table 2 | Estimated PEESE-adjusted overall effect contrasted with sensitivity analysis of missing outcomes imputation methods**

| Imputation method | g | SE | 95% CI | | p | Q(364) | τ² level | | I² level | |
|---|---|---|---|---|---|---|---|---|---|---|
| | | | LL | UL | | | 1 | 2 | 1 | 2 |
| Median | 0.37 | 0.036 | 0.30 | 0.44 | < 0.001 | 4446 | 0.071 | 0.131 | 29.9% | 55.0% |
| Minimum | 0.44 | 0.036 | 0.37 | 0.51 | < 0.001 | 4064 | 0.101 | 0.099 | 36.0% | 35.5% |
| Maximum | 0.35 | 0.033 | 0.29 | 0.42 | < 0.001 | 28327 | 0.054 | 0.148 | 25.7% | 70.0% |

Effect sizes are reported as Hedges' g with 95% confidence intervals. Effect size estimates were pooled using the restricted maximum-likelihood (REML) estimators. Statistical inference was based on two-sided t tests with Knapp–Hartung adjustment. All P-values are two-sided and no adjustment for multiple comparisons was applied. Heterogeneity statistics (τ², I²) are reported at within- and between-study levels.
*SE* standard error, *CI* confidence interval, *LL* lower limit, *UL* upper limit.

studies with higher relative dispersion of truth judgments. This finding suggests that variability in outcome precision may partially account for between-study heterogeneity.

**Risk of Bias**
Using the RoB-2 framework, 43 of the 182 studies were judged at low overall risk, while the remaining 139 raised some concerns. Among the five RoB-2 domains, deviation from intended intervention (Domain 2) raised concerns in 86 studies and selection of the reported results (Domain 5) in 138 studies, primarily resulting from lack of pre-registration processes. Supplementary Data 2 describes every study by the five RoB-2 domains. Figure 2 provides a weighted summary.

A three-level meta-regression using overall RoB grade as a moderator showed no significant difference between effect sizes from studies at low risk of bias and those with some concerns, $b = 0.12$, 95% CI [−0.02, 0.26], $t(363) = 1.63$, $p = 0.104$. Thus, the risk-of-bias analyses suggested that the pooled illusory truth effect remained stable after controlling for study-level biases.

**Meta-regression**
We performed a meta-regression to investigate the moderating factors at the root of the within-study and between-study heterogeneity. To handle missing values in moderator variables, we used multivariate imputation by chained equations (mice)[47]. This method fills in missing

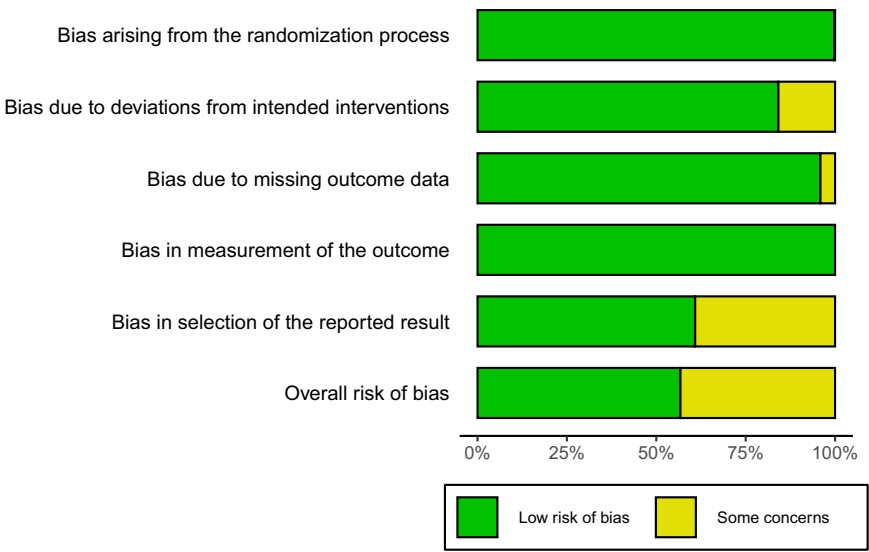

**Fig. 2 | Weighted risk-of-bias summary (RoB 2) across all included experiments.** Percentage of effect sizes that were judged Low risk of bias (green) or Some concerns (yellow) in each Cochrane RoB 2 domain and for the overall risk of bias judgment. Segments widths are weighted by the inverse-variance weights of each effect size in the meta-analysis.

moderator values by estimating them based on patterns in the observed data (e.g., other moderators and effect sizes), preserving the relationships among variables and reducing the loss of information that would result from excluding cases with missing values. We created 20 imputed data sets, each with a maximum of 20 iterations to ensure convergence of the chained equations. We then pooled the resulting coefficients and their variances according to Rubin's rules to obtain the meta-analytic estimates[47]. As a result, the adjusted degrees of freedom associated with t-statistics also account for the variability introduced by multiple imputations[48]. The results of each meta-regression reported below presents the effect of one moderating factor while all other moderating factors are held constant. To provide an overview of effect sizes within each subgroup, we also report the subgroup estimates. Because these values are not adjusted for the other moderators, they are not directly comparable to the coefficients from the multivariate meta-regression. Supplementary Data 3 summarizes the subgroup analyses contrasted with Dechêne et al.'s meta-analysis[2]. A visualization of the meta-regression results is presented in Fig. 3.

For brevity, omnibus test results described below are those obtained using the median to impute missing standard deviations and correlation coefficients. Any inconsistent results obtained through sensitivity analyses of the missing outcome of interest (i.e., minimum-maximum estimations) are further specified. Supplementary Data 4 provides the full comparative results of the moderator analysis, including sensitivity analysis with the minimum and maximum estimations for missing outcomes.

Studies on the illusory truth effect use a variety of materials such as trivia statements, topic-specific or general knowledge statements, social media news headlines, conspiracy theories, political claims, marketing advertisements, and meaningless sentences. In our coding, these were grouped in three categories: standard statements, headlines, and claims. The illusory truth effect was significantly larger in studies using standard statements ($g = 0.41$, 95% CI [0.33, 0.48]; observed u = 309) compared to headlines ($g = 0.17$, 95% CI [0.01, 0.34]; observed u = 43), $b = 0.25$, 95% CI [0.01, 0.48], $t(147) = 2.10$, $p = .038$, but not compared to claims ($g = 0.47$, 95% CI [0.20, 0.73]; observed u = 12), $t(147) < 1$. Effect sizes did not significantly differ between news headlines and opinion-based claims, $t(138) < 1$.

Repetition can be manipulated either within-participants (e.g., each participant judges both repeated and new items), or between-

participants (e.g., one group judges only repeated items and another only new items). The illusory truth effect did not significantly differ whether repetition was manipulated within-participants ($g = 0.38$, 95% CI [0.30, 0.45]; observed u = 361) or between participants ($g = 0.17$, 95% CI [−0.28, 0.61]; observed u = 5), $t(252) < 1$.

In primary studies, the task participants perform during initial exposure to information varies widely, from passively reading statements, to making unrelated judgments (e.g., rating interestingness, categorizing), to explicitly evaluating truthfulness. A significant moderating effect was found for the content of the task used during the exposure phase. When participants were asked for a truth judgment during the exposure phase ($g = 0.10$, 95% CI [−0.03, 0.23]; observed u = 78), the illusory truth effect was consistently reduced compared to irrelevant tasks ($g = 0.42$, 95% CI [0.34, 0.50]; observed u = 213), $b = −0.32$, 95% CI [−0.47, −0.16], $t(264) = 4.02$, $p < .001$, or passive processing ($g = 0.47$, 95% CI [0.34, 0.60]; observed u = 73), $b = −0.46$, 95% CI [−0.70, −0.23], $t(200) = 3.94$, $p < .001$. However, the illusory truth effect did not significantly vary when the exposure task involved passive processing compared to performing an irrelevant task, $t(195) = 1.69$, $p = .093$.

Some studies warn participants that some information may be false, either before the exposure phase or before the test phase. We coded these as separate moderators because their timing may inform of the engagement of different cognitive processes. Providing a truth warning did not moderate the illusory truth effect at either phase of the experiment. During the exposure phase, the effect did not differ significantly with a truth warning ($g = 0.34$, 95% CI [0.25, 0.43]; observed u = 163) or without one ($g = 0.44$, 95% CI [0.32, 0.57]; observed u = 41), after controlling for other moderating variables, $t(86) < 1$. Similarly, the illusory truth effect with a warning ($g = 0.38$, 95% CI [0.27, 0.48]; observed u = 66) did not differ compared to no warning ($g = 0.37$, 95% CI [0.26, 0.49]; observed u = 41) in the test phase, $t(27) < 1$.

Veracity cues refer to any explicit or implicit indicators of a statement's truth status, such as labels (disputed by a third-party fact-checker), epistemic qualifiers (it is unlikely that…), source reliability cues (all statements voiced by female are false), or corrective feedback (e.g., immediately after initial exposure, or at the end of exposure phase). In our coding scheme, we classified these cues by valence (e.g., true, false, or no cue) regardless of the type of cue. The valence of

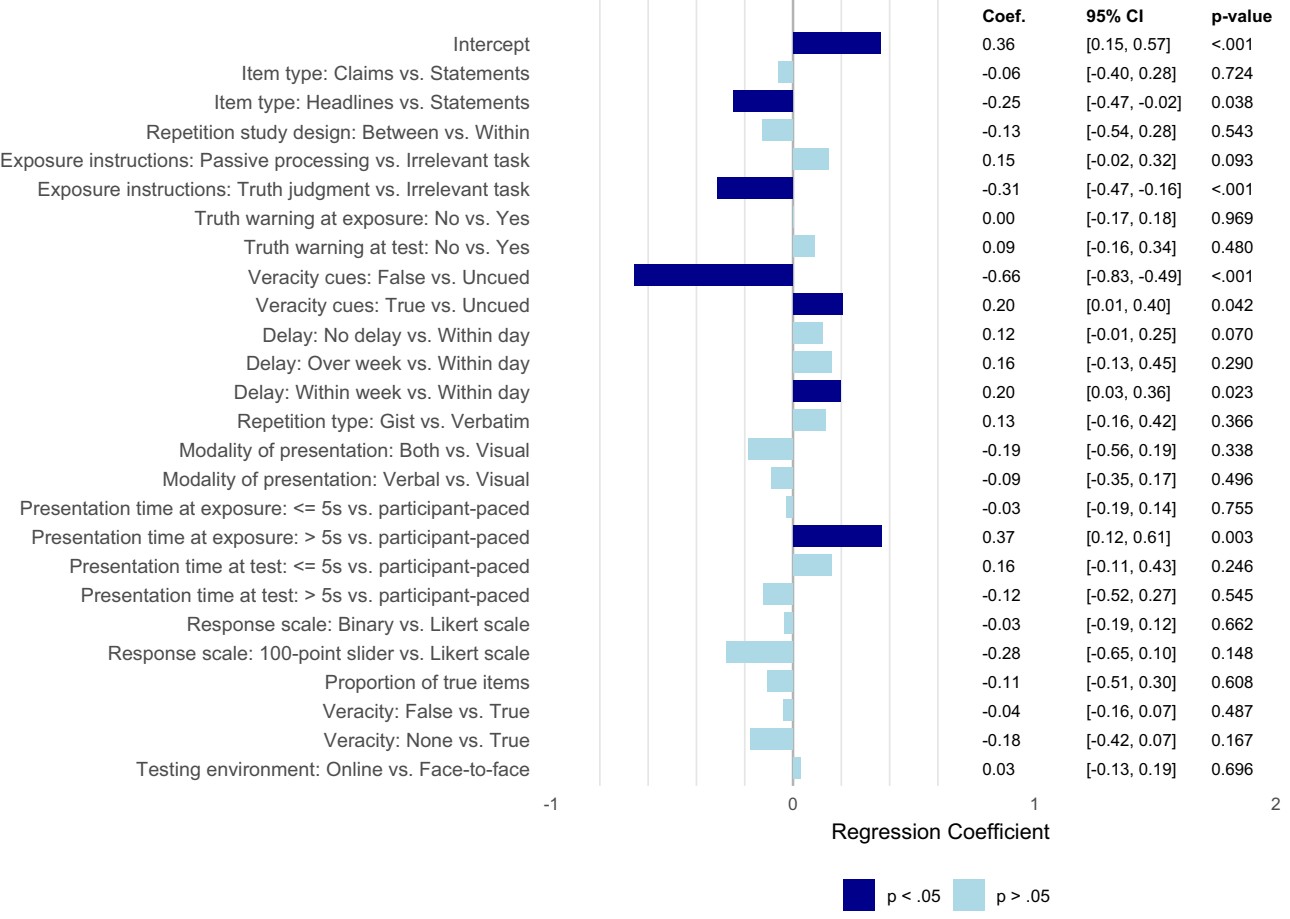

| | Coef. | 95% CI | p-value |
|---|---|---|---|
| Intercept | 0.36 | [0.15, 0.57] | <.001 |
| Item type: Claims vs. Statements | -0.06 | [-0.40, 0.28] | 0.724 |
| Item type: Headlines vs. Statements | -0.25 | [-0.47, -0.02] | 0.038 |
| Repetition study design: Between vs. Within | -0.13 | [-0.54, 0.28] | 0.543 |
| Exposure instructions: Passive processing vs. Irrelevant task | 0.15 | [-0.02, 0.32] | 0.093 |
| Exposure instructions: Truth judgment vs. Irrelevant task | -0.31 | [-0.47, -0.16] | <.001 |
| Truth warning at exposure: No vs. Yes | 0.00 | [-0.17, 0.18] | 0.969 |
| Truth warning at test: No vs. Yes | 0.09 | [-0.16, 0.34] | 0.480 |
| Veracity cues: False vs. Uncued | -0.66 | [-0.83, -0.49] | <.001 |
| Veracity cues: True vs. Uncued | 0.20 | [0.01, 0.40] | 0.042 |
| Delay: No delay vs. Within day | 0.12 | [-0.01, 0.25] | 0.070 |
| Delay: Over week vs. Within day | 0.16 | [-0.13, 0.45] | 0.290 |
| Delay: Within week vs. Within day | 0.20 | [0.03, 0.36] | 0.023 |
| Repetition type: Gist vs. Verbatim | 0.13 | [-0.16, 0.42] | 0.366 |
| Modality of presentation: Both vs. Visual | -0.19 | [-0.56, 0.19] | 0.338 |
| Modality of presentation: Verbal vs. Visual | -0.09 | [-0.35, 0.17] | 0.496 |
| Presentation time at exposure: <= 5s vs. participant-paced | -0.03 | [-0.19, 0.14] | 0.755 |
| Presentation time at exposure: > 5s vs. participant-paced | 0.37 | [0.12, 0.61] | 0.003 |
| Presentation time at test: <= 5s vs. participant-paced | 0.16 | [-0.11, 0.43] | 0.246 |
| Presentation time at test: > 5s vs. participant-paced | -0.12 | [-0.52, 0.27] | 0.545 |
| Response scale: Binary vs. Likert scale | -0.03 | [-0.19, 0.12] | 0.662 |
| Response scale: 100-point slider vs. Likert scale | -0.28 | [-0.65, 0.10] | 0.148 |
| Proportion of true items | -0.11 | [-0.51, 0.30] | 0.608 |
| Veracity: False vs. True | -0.04 | [-0.16, 0.07] | 0.487 |
| Veracity: None vs. True | -0.18 | [-0.42, 0.07] | 0.167 |
| Testing environment: Online vs. Face-to-face | 0.03 | [-0.13, 0.19] | 0.696 |

Regression Coefficient

■ p < .05    ■ p > .05

**Fig. 3 | Bar plot of moderator analysis of the illusory truth effect.** The regression coefficient values and their 95% confidence intervals (CI) were computed using median estimates imputation for missing outcomes of interest. All tests are two-sided. The reference group for each categorical moderator was selected based on the most frequently observed subgroup in the dataset. Negative coefficients indicate a reduction in the illusory truth effect relative to the reference group, positive coefficients indicate an increase. All coefficients are corrected for small-study effects using the PEESE method (precision-effect estimate with standard error). Exact test statistics (t values) and degrees of freedom are reported in the Results section and Supplementary Data 2. No adjustment for multiple comparisons was applied.

veracity cues significantly moderated the illusory truth effect. Across cue types (e.g., any timing or format), providing false cues during the exposure phase ($g = -0.18$, 95% CI [$-0.33$, $-0.03$]; observed $u = 40$) decreased the effect compared to no cues ($g = 0.43$, 95% CI [0.36, 0.50]; observed $u = 287$), $b = -0.66$, 95% CI [$-0.83$, $-0.49$], $t(219) = 7.70$, $p < .001$, and providing true cues ($g = 0.70$, 95% CI [0.51, 0.88]; observed $u = 33$), $b = -0.86$, 95% CI [$-1.04$, $-0.68$], $t(244) = 9.47$, $p < .001$). Additionally, explicitly labelling statements as true yielded a larger illusory truth effect compared to no veracity indication, $b = 0.20$, 95% CI [0.01, 0.40], $t(229) = 2.05$, $p = .042$. However, this was not robust with sensitivity analysis using maximum estimates of missing values, $t(275) = 1.15$, $p = .253$. No other moderating effects were found.

We also conducted an exploratory analysis to assess the effect of the type of cue on the illusory truth effect. We found a significant decrease in the illusory truth effect when using false cues involving epistemic qualifiers ($g = -0.57$, 95% CI [$-0.87$, $-0.29$]; observed $u = 7$), source reliability cues ($g = -0.73$, 95% CI [$-0.95$, $-0.50$]; observed $u = 16$), immediate feedback ($g = -0.91$, 95% CI [$-1.44$, $-0.39$]; observed $u = 2$), and delayed feedback ($g = -1.12$, 95% CI [$-1.51$, $-0.73$]; observed $u = 7$). Among these, false delayed feedback was significantly more effective than false epistemic qualifiers in reducing the effect ($b = -0.55$, 95% CI [$-1.04$, $-0.06$], $t(263) = 2.20$, $p = .029$). In contrast, false labels showed no reliable effect ($g = 0.04$, 95% CI [$-0.29$, 0.38]; observed $u = 5$). Given the small number of effect sizes in several

categories, these findings should be interpreted cautiously (see Supplementary Methods).

The length of time between initial exposure to information and subsequent truth judgment varies considerably across studies, ranging from immediate testing to delays of several weeks. We coded this as a four-level moderator: no delay, within day delay, up to one week delay, and more than one week delay. The illusory truth effect was significantly smaller when the test phase followed the exposure phase in the same day (e.g., after a distractor task) ($g = 0.33$, 95% CI [0.23, 0.43]; observed $u = 138$) compared to when test phase followed exposure phase in the same week ($g = 0.36$, 95% CI [0.21, 0.50]; observed $u = 47$), $b = -0.20$, 95% CI [$-0.37$, 0.03], $t(278) = 2.29$, $p = .023$, and compared to no delay conditions; however, this difference did not reach significance ($g = 0.46$, 95% CI [0.35, 0.56]; observed $u = 135$), $b = 0.12$, 95% CI [$-0.26$, 0.01], $t(254) = 1.82$, $p = .070$. No significant difference was found when comparing studies with a same day delay to those with a delay over a week ($g = 0.25$, 95% CI [0.06, 0.44]; observed $u = 41$), $t(82) = 1.07$, $p = .290$. No other moderating effect of delay was found.

Repeated information can be presented verbatim (e.g., identical wording as in the initial exposure), or as a gist repetition (e.g., preserves the core meaning while altering surface details). No moderating effect of the repetition type was found. The illusory truth effect did not differ significantly between studies in which the statement was repeated verbatim ($g = 0.36$, 95% CI [0.29, 0.44]; observed $u = 350$) or gist

($g = 0.63$, 95% CI [0.35, 0.91]; observed u = 16) in the test phase, $t(243) = 1.19$, $p = .237$.

Repeated statements are presented either visually (e.g., written text), verbally (e.g., spoken), or through a combination of both modalities. No effect of the presentation mode on the illusory truth effect was found. The effect sizes did not differ between study designs with visually presented statements ($g = 0.38$, 95% CI [0.31, 0.46]; observed u = 257), compared to verbally presented statements ($g = 0.30$, 95% CI [0.12, 0.48]; observed u = 62), $t(158) < 1$, or both modes of presentation ($g = 0.32$, 95% CI [0.02, 0.62]; observed u = 17), $t(61) < 1$.

Across studies, the duration participants are given to view or hear each statement varies considerably, both during the initial exposure and at the later truth judgment. To ensure a balanced distribution of effect sizes across categories, we coded presentation time into three categories: 5 seconds or less, more than 5 seconds, and participant-paced. By coding presentation time at both stages, we can assess whether the illusory truth effect is more strongly moderated by encoding or retrieval conditions.

During the exposure phase, studies with a presentation time longer than 5 seconds reported larger effect sizes ($g = 0.48$, 95% CI [0.33, 0.63]; observed u = 92) than those with self-paced presentation time ($g = 0.34$, 95% CI [0.26, 0.43]; observed u = 179), $b = 0.37$, 95% CI [0.12, 0.61], $t(184) = 2.96$, $p = .003$, and those with a presentation time 5 seconds or less ($g = 0.39$, 95% CI [0.26, 0.52]; observed u = 77), $b = 0.39$, 95% CI [0.15, 0.64], $t(214) = 3.14$, $p = .002$. However, in sensitivity analysis using maximum estimates, the effect size difference was only marginal between self-paced presentation and presentations of more than 5 seconds, $t(189) = 1.86$, $p = .065$, and non-significant between presentations of 5 seconds or less and more than 5 seconds, $t(186) = 1.62$, $p = .108$. Effect sizes did not differ between studies in which presentation time was 5 seconds or less and those in which presentation was self-paced, $t(265) < 1$.

During the test phase, no moderating effect of presentation time on the magnitude of the illusory truth effect was found. A self-paced presentation yielded similar effect sizes ($g = 0.37$, 95% CI [0.30, 0.45]; observed u = 314) compared to a presentation of 5 seconds or less ($g = 0.42$, 95% CI [0.19, 0.64]; observed u = 29), $t(245) = 1.16$, $p = .246$, or more than 5 seconds ($g = 0.27$, 95% CI [−0.02, 0.56]; observed u = 19), $t(125) < 1$. Similarly, no difference of effect was found for a presentation of 5 seconds or less or a presentation of more than 5 seconds, $t(122) = 1.08$, $p = 0.281$.

In primary studies, truth judgments can be assessed using different response formats, such as binary (e.g., true/false), Likert-type scales, or continuous sliders. In our coding, we grouped these into three categories: binary, Likert, and 100-point slider scale. We also distinguished scales with an odd number of points (i.e., allowing a midpoint) from those with an even number of points (i.e., forcing a choice). The size of the illusory truth effect did not vary between studies using Likert scale ($g = 0.36$, 95% CI [0.29, 0.44]; observed u = 269) vs. binary scales ($g = 0.43$, 95% CI [0.30, 0.56]; observed u = 91), $t(260) < 1$, or 100-point sliders ($g = 0.23$, 95% CI [−0.17, 0.62]; observed u = 6), $t(212) = 1.45$, $p = .148$. Similarly, the effect size did not vary between odd ($g = 0.30$, 95% CI [0.18, 0.43]; observed u = 119) vs. even ($g = 0.40$, 95% CI [0.32, 0.47]; observed u = 241) scales, $t(267) < 1$.

The factual status of items varies across experiments. In addition, the overall proportion of true statements in the stimulus set differs across studies, from 0 (all false) to 100 (all true), with many using mixed proportions. We coded these features as two moderators: statements' veracity (true, false, none), and proportion of true statements (continuous variable from 0 to 1). The size of the illusory truth effect was not modulated the factual truth of the exposed statements. Repetition of true statements ($g = 0.46$, 95% CI [0.36, 0.56]; observed u = 83) yielded similar effect sizes compared to false statements ($g = 0.33$, 95% CI [0.23, 0.42]; observed u = 81), $t(63) < 1$, or non-verifiable statements (e.g., opinions, meaningless statements)

($g = 0.31$, 95% CI [0.12, 0.49]; observed u = 12), $t(52) = 1.40$, $p = 0.167$. Similarly, the proportion of true or false statements showed no influence on the effect size ($b = −0.02$, 95% CI [−0.39, 0.35]; observed u = 339), $t(175) < 1$, indicating that the illusory truth effect is similar across different proportions of true vs. false statements.

Studies on the illusory truth effect have been conducted in both face-to-face laboratory settings and online environment with surveys. The illusory-truth effect did not differ between online ($g = 0.40$, 95% CI [0.32, 0.48]; observed u = 196) and face-to-face ($g = 0.31$, 95% CI [0.20, 0.43]; observed u = 169) testing contexts, $t(231) < 1$.

Averaged across the 20 multiple imputed datasets, the pseudo-$R^2$ for the between-study variance was 0.37 (range = 0.35–0.41). On average, 37% of the between-study heterogeneity relative to the PEESE-adjusted baseline model was accounted for by the moderator set. Supplementary Figs. 1–16 display forest plots stratified by item type, exposure instructions, veracity cues, and presentation time at exposure.

## Discussion

This systematic review and three-level meta-analysis provides robust evidence that repetition increases the perceived truth of information, with growing relevance in today's media landscape. Synthesizing evidence from 182 studies (366 effect sizes; $N = 31,184$), our analysis provides robust confirmation of the illusory truth effect, indicating a small effect size ($g = 0.37$, 95% CI [0.30, 0.44]), after correcting for publication bias. These findings suggest that repetition reliably increases perceived truth across diverse methodological contexts, highlighting the relevance for contemporary concerns regarding misinformation.

Our methodological approach improved upon the previous meta-analysis by incorporating more recent statistical techniques, including a three-level random-effects model to account for within-study dependence of effect sizes, multivariate imputation to handle missing moderator data, and sensitivity analyses to evaluate the impact of missing outcome data. We also assessed study-level quality using the Risk of Bias 2 tool, which indicated that most studies raised some concerns primarily due to a lack of pre-registration. However, effect sizes did not differ significantly by overall risk of bias level, suggesting that the pooled estimate was not substantially biased by study quality. In addition, while publication bias diagnostics revealed evidence of small-study effects, the illusory truth effect remained robust after applying appropriate correction methods.

Beyond the overall effect, substantial between-study heterogeneity in effect sizes warranted moderator analyses to clarify the conditions under which repetition influences the perceived truth. To structure this discussion, we group moderators by stimulus characteristics, exposure phase, test phase, and study design. Averaged across multiple imputed datasets, the moderator set explained approximately 37% of the observed variance. This indicates meaningful heterogeneity beyond sampling error.

Regarding stimulus characteristics, the illusory truth effect was robust across content types (e.g., trivia statements, news headlines, opinions), repetition formats (verbatim vs. gist), factual truth status (true vs. false), or proportion of true items. The absence of moderation by truth status aligns with prior findings and reinforces the idea that fluency affects perceived truth independently of objective accuracy[30,49–51]. Similarly, the comparable effects observed for verbatim and gist repetitions are consistent with both the fluency account and referential theory. While the former emphasizes the role of conceptual (vs. perceptual) processing ease[52], the latter attributes the effect to stronger semantic coherence across repetitions[38,39]. We also found that studies using news headlines tend to yield smaller effect sizes than those using standard trivia statements. One possible explanation is that many headline-based studies assessed the illusory truth effect in the context of social media misinformation, where items

were not systematically chosen for neutrality or ambiguity. Moreover, a recent meta-analysis[53] found that participants were generally able to distinguish real from fake news. One possibility is that the format of social media content may cue readers to engage in more cautious evaluation. This could reduce reliance on fluency during truth judgments, thereby attenuating the illusory truth effect.

In contrast, several moderators related to the exposure phase significantly influenced the magnitude of the illusory truth effect. Asking participants to perform a truth judgment in the exposure phase was associated with a weaker illusory truth effect compared to irrelevant tasks or passive processing. One explanation, consistent with the fluency discount hypothesis, is that individuals prompted with an initial truth judgment may activate relevant knowledge, thereby reducing reliance on fluency as a cue of truth. This attenuation appears to be limited in time, emerging primarily when the test phase follows shortly after exposure and dissipating after approximately two days[11,54]. Importantly, this pattern remains after controlling for participants' memory of the statements, suggesting that factors beyond simple memory retention are involved. From a referential theory perspective, truth judgments may create links between referents in memory whose valence (e.g., excitatory vs. inhibitory) depends on whether the information is encoded as true or false. In contrast, in irrelevant-task or passive-processing conditions, statements may be implicitly assumed to be true, leading to predominantly excitatory links. Furthermore, individual retrieval strategies may also influence the illusory truth effect: some participants may attempt to retrieve their previous judgment stored in memory, whereas others may primarily rely on fluency or adopt a different strategy.

Additionally, longer presentation time at the exposure phase (more than 5 seconds) was associated with a larger truth effect, although this relationship was only marginal in one of the sensitivity analyses. Accordingly, this result should be interpreted with some caution. To our knowledge, no prior work has systematically isolated the exposure phase to examine how stimulus duration influences the illusory truth effect. This potential relationship has thus not been reported previously. From a processing fluency perspective, longer exposure may facilitate more complete perceptual encoding, making the statement easier to process at test and thereby more likely to be judged as true. From a referential theory perspective, extended exposure may provide more opportunity to activate and integrate related concepts in memory, increasing the coherence of the referential network associated with the statement. Future work should systematically investigate whether limiting presentation time during initial exposure could effectively attenuate the illusory truth effect.

Crucially, our findings suggested that the valence of veracity cues significantly moderate the illusory truth effect. In particular, false cues not only attenuated but, on average, reversed the illusory truth effect, such that repeated statements were judged as less true than new statements. In contrast, true cues did not reliably alter the magnitude of the illusory truth effect. This pattern suggests that explicit falsity cues can override the influence of repetition, providing further support for the potential effectiveness of debunking interventions. However, because this reversal is observed in subgroup estimates that are not adjusted for the full set of moderators, it should be interpreted as descriptive of existing study patterns. To explore how variability in the type of veracity cue (e.g., source reliability cues, epistemic qualifiers, labels, immediate or delayed feedback) contribute to inconsistent findings in the literature, we conducted an exploratory analysis. We found that the moderating influence of veracity cues may depend on cue type, with stronger effects of feedback, source reliability cues, and epistemic qualifiers compared to labels. However, the reliability of these conclusions is limited by the small number of effect sizes per category, and they should thus be interpreted with caution (see Supplementary Methods). Understanding these mechanisms is crucial for developing effective interventions to mitigate the spread of misinformation[19]. Future studies should systematically investigate how different types of veracity cues influence the illusory truth effect.

Surprisingly, we found no effect of explicit instructional warnings at the exposure phase on the illusory truth effect. This contrasts with Jalbert et al.'s findings[27], which showed that explicit warnings reduced the illusory truth effect, but only when given during exposure alongside reading instructions. A possible explanation is that implicit warnings (e.g., initial truth ratings) and explicit warnings may overlap. Once participants are instructed to evaluate veracity, an additional instructional warning may provide only few additional information, which makes its impact difficult to detect at the aggregate level.

In addition, we found little evidence that variables in the test phase (e.g., warnings, presentation time) reliably moderated the illusory truth effect. These null results align with prior work showing that time pressure or warnings during the test phase do not alter the effect[27,55], suggesting that repetition may exert its strongest influence during initial exposure rather than later evaluation. Finally, the effect was generally consistent across study design features (e.g., delay length, repetition design, modality of presentation, response scale, testing environment). Although some primary studies have reported smaller effects in between-participant designs[56–58], these patterns did not replicate in our aggregate analyses and were likely constrained by the limited number of studies in some categories (e.g., only five effect sizes extracted from between-participant designs). Similarly, we observed a relatively smaller truth effect when the test phase followed exposure within the same day versus within the same week, Primary studies suggested that the moderating effect of delay may interact with other variables influencing the illusory truth effect during the exposure phase, such as the type of task[59] or the type of repetition[60]. As such, it may be difficult to isolate the sheer effect of delay in aggregate data. Null effects for moderators such as the modality of presentation, the response scale or testing environment were consistent with earlier meta-analytic findings[2], suggesting that the illusory truth effect extends across commonly used research contexts. However, recent evidence suggests that divided attention (e.g., experimenter presence, environmental distractions) may still influence the illusory truth effect and warrant closer examination in future research[61].

Synthesizing across moderators, the results indicate that illusory truth effect may be shaped primarily by processes occurring during the initial encounter with information, and several moderators point to potential leverage for interventions. Performing an initial truth judgment was associated with a weaker illusory truth effect, suggesting that accuracy-focused evaluation at first exposure may help to attenuate the influence of repetition. This interpretation is consistent with intervention studies showing that accuracy prompts and early veracity judgments can reduce susceptibility to misinformation[11,28]. Similarly, the presence of falsity cues was reliably linked to a reduced effect, pointing to the potential utility of clear validity signals during initial encounter with potential misinformation[51]. In contrast, longer exposure durations tended to amplify the effect, raising the possibility that stimulus timing may be an overlooked target for intervention, though this relationship warrants further investigation.

Together, these findings indicate that interventions may be most effective when they encourage accuracy-focused processing and make falsity salient at the point of first exposure. Promising approaches include metacognitive prompts[26], encouraging early reflection with accuracy checks[28], and debunking strategies[62]. Importantly, the concern that repeating false claims during debunking might backfire appears less warranted than previously feared; current evidence suggests that corrective interventions can be effective without amplifying beliefs in the exposed claims[63,64]. At the same time, the illusory truth effect captures only one mechanism among many that contribute to the belief and spread of misinformation. While it cannot solve

misinformation alone, reducing the illusory truth effect is a practical way to limit its impact.

Several limitations of the current meta-analysis should be acknowledged. While we applied the PEESE method to gauge and correct potential publication bias, its accuracy deteriorates under high heterogeneity[65], the corrected estimate should thus be interpreted with caution. Recent methodological works suggest that using standardized mean difference (e.g., Hedge's g) may inflate between-study heterogeneity estimates. For instance, heterogeneity in standard deviations alone can raise I² even when underlying mean differences are relatively stable[66]. Thus, part of the observed heterogeneity may be a statistical artefact of standardization rather than true differences in the illusory truth effect. To explore this possibility, we examined whether differences in the relative variability of truth judgments across studies contributed to inconsistencies in effect sizes. Results from this additional analysis indicated that studies with more dispersed outcomes tended to report smaller standardized effects, suggesting that statistical noise in measurement precision may inflate heterogeneity estimates. This finding supports the view that variability in study-level standard deviations may contribute to between-study heterogeneity. Furthermore, the potential influence of hidden multiplicity warrants caution. Multiplicity often leads to selective reporting, where researchers may preferentially publish outcomes that align with their hypotheses, while omitting null or contrary results. Many included experiments reported multiple eligible effect sizes (e.g., across different time points: immediate vs. delayed test). While we extracted all eligible effects and used a multilevel meta-analytic approach to account for within-study dependence, this does not fully eliminate the risk of bias introduced by multiplicity. The analytic choices made by original authors (e.g., which outcomes to report or emphasize), as well as those made during data extraction, may still influence aggregated effect estimates despite statistical adjustment.

Another key limitation of the current meta-analysis is the substantial methodological diversity across the included studies. Procedures varied in initial rating tasks, delay intervals, presentation timing, and other moderators. While our three-level meta-regression approach is appropriate for estimating overall trends and accounting for clustered data, it lacks the statistical power to detect interaction effects given the uneven distribution of study designs across moderator combinations. Thus, moderator effects should be interpreted as indicators of relative salience rather than definitive estimates, as potential interactions may remain undetected in aggregate-level analyses. Moderator selection was primarily guided by pragmatic considerations than by alignment to a specific theoretical framework. Thus, they should not be interpreted as direct tests of cognitive models. In fact, the moderators that could be reliably extracted across studies do not systematically align with key constructs from existing theoretical accounts. Moreover, the limited number of effect sizes per moderator combination prevents us from testing interaction effects that would be necessary to evaluate core theoretical predictions. While our findings provide relevant insights to theory-relevant processes (e.g., fluency, depth of processing, encoding conditions), they should not be interpreted as conclusions about the validity of competing cognitive models.

Some questions remain to be addressed in future primary studies on the illusory truth effect. For instance, up to now most studies have been conducted on WEIRD (i.e., Western Educated Industrialized Rich and Democratic) population raising the question of the universality of such an effect. It remains unclear whether individuals from non-WEIRD cultures or lower socioeconomic background would show the same susceptibility to the illusory truth effect.

In conclusion, this updated meta-analysis confirms the robustness of the illusory truth effect, with a small but consistent overall effect (g = 0.37) after correcting for publication bias. While repetition reliably increases perceived truth, the effect varies substantially across study designs. Our findings highlight the central role of initial exposure in shaping later truth judgments, with robust moderators including the type of item, participants' task during the exposure phase, and the valence of veracity cues. These insights have direct practical relevance for developing scalable interventions to limit the impact of repeated misinformation.

## Methods

### Transparency and Openness
We followed the PRISMA 2020 guidelines for systematic reviews[67] (with the checklist available in Supplementary Table 1). Data analysis and visualization were conducted in the statistical environment R (v.4.4.2)[68] running under RStudio (2024.12.1 + 563)[69], with the code written following[70] guidelines and using the package "metafor" (v.4.8.0)[71]. Data processing and reshaping were performed using "tidyverse" (v.2.0.0)[72], visualizations were produced with "ggplot2" (v.3.5.1)[73], missing-data patterns were inspected using "VIM" (v.6.2.2)[74], and risk-of-bias figures were generated using "robvis" (v.0.3.0)[75]. Missing moderating factors were imputed with the package "mice" (v.3.17.0)[47]. This review was not pre-registered. No review protocol was prepared.

### Eligibility Criteria
Relevant studies were identified based on the following inclusion criteria:

1. Population: human population of any age, including those from clinical groups
2. Intervention: verbatim or gist repetition of statements (e.g., trivia, political, marketing) presented verbally or visually
3. Comparator: within-subjects (repeated vs. non-repeated statements) or between-subjects (non-repetition control vs. repetition group)
4. Outcome: numerical (Likert-type scale, slider or similar) or binary (true/false) measure of subjective truth judgments, comparing truth ratings for new versus once-repeated items and non-repeated items (between-item criterion)
5. Study type: empirical quantitative studies

Any document written in English with unique data was considered regardless of publication status, publication type, or year of release. We used several exclusion criteria to reduce heterogeneity in effect sizes. We excluded studies that involved (1) veracity cues during truth judgment in the test phase, (2) multiple exposure to a statement before evaluation (e.g., participants are exposed to a same statement twice or more before test phase), (3) labels tagging repeated and new statements, or (4) an intervention to reduce the illusory truth effect. We also excluded studies in which the illusory truth effect was measured (5) by repeating contradictory or incongruent statements compared to the first exposure to that statement. (6) In contrast to the previous meta-analysis[2], all studies that used a within-item criterion to detect the illusory truth effect were excluded because most studies since 2006 have used the between-item criterion to measure illusory truth effect. Moreover, we excluded studies that (7) did not ask participants to rate the veracity of the statements (e.g., agreement or belief ratings), (8) did not report effect sizes or enough data to compute such effect sizes (i.e., mean of repeated or new statements). Finally, we excluded any (9) duplicated data. If the same data from multiple eligible publications fitted our inclusion criteria, we included only the data from the first peer-reviewed publication.

### Information Sources, Search Strategy and Selection Process
First, we retrieved all articles (n = 89) included in the most recent systematic literature review of the illusory truth effect[3]. To identify articles published since the ones included in the systematic review (i.e., from February 2020), we screened the following database

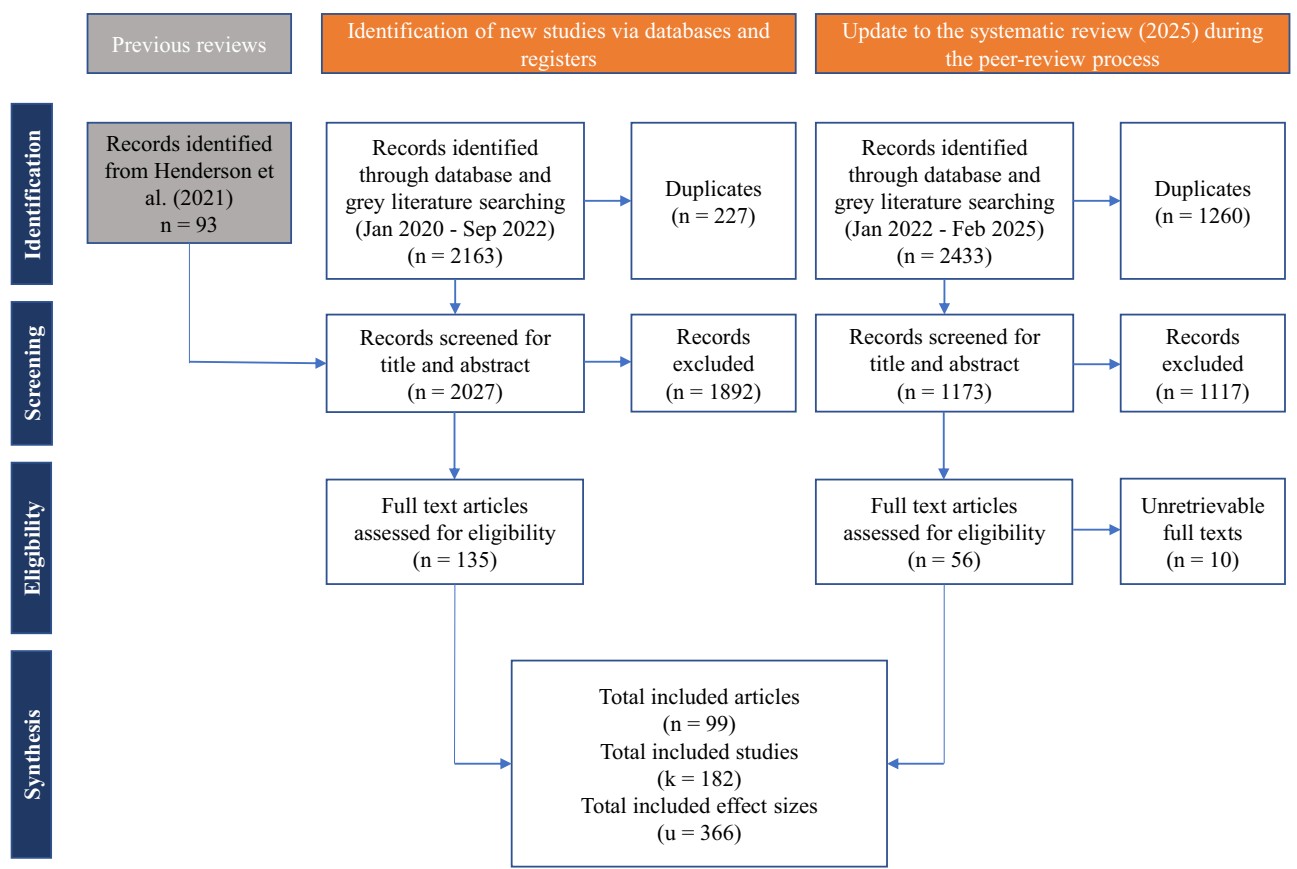

**Fig. 4 | Flow diagram of systematic review and study selection.** This flow diagram was inspired by the PRISMA (Preferred Reporting Items for Systematic Reviews and Meta-Analyses) guidelines and has been adapted to better represent our study selection process and data sources. Source: Page MJ, et al. BMJ 2021;372:n71. https://doi.org/10.1136/bmj.n71. This work is licensed under CC BY 4.0. To view a copy of this license, visit https://creativecommons.org/licenses/by/4.0/.

EBSCOHOST (APA PsycArticles, APA PsycInfo, Business Source Premier, EconLit, Psychology and Behavioral Sciences Collection), NBCI (PubMed), Elsevier (Scopus), Web of Sciences, OSF Preprints (PsyArXiv, Thesis Commons), Google Scholar, and ProQuest (Dissertations & Thesis) with the keywords used by Henderson et al.[3]: "illusory truth", "illusory truth effect", "illusions of truth", "reiteration effect", "repetition induced truth effect", "repetition based truth effect", "truth effect", "truth judgment". These keywords were validated by Henderson et al.[3] to identify a predetermined set of 20 eligible articles in Web of Science and Scopus. Additionally, we searched database using their extensive revised keywords (cf. Supplementary Methods).

We used "Zotero" and "Publish or Perish"[76] software to export results and retrieve the title and the abstract of each article. This combined search conducted in September 2022 yielded a total of 2166 articles, from which 192 duplicates were removed using an automatic tool from Zotero, and 37 additional articles were manually identified as duplicates and were removed. Out of the 1971 remaining articles, we added the articles identified by Henderson et al.[3], for a total of 2027 articles. Titles and abstracts were screened to determine whether full text should be retrieved based on relevance (i.e., experimental articles, mention of repetition or fluency...). All titles and abstracts were screened by two reviewers (S.Y. & M.G.) independently, and every article selected by any of the two reviewers was retained for the full-text check. The title and abstract check yielded a total of 135 articles, for which full texts were retrieved. Finally, inclusion and exclusion criteria were assessed for each full text article by three reviewers (S.Y., M.G., & Y.B.).

We conducted an update to the systematic review on 24 February 2025, using the same databases, keyword string, and screening procedures that were applied to the original review. Limiting to records published after September 2022 returned 2433 titles/abstracts. After removing 1234 duplicates with Zotero's automatic tool and 26 duplicates through manual inspection, 1173 unique records remained. Title and abstract screening excluded 1117 of these records, leaving 56 additional articles to be assessed for eligibility, 10 of which could not be retrieved even after messaging the authors. Figure 4 provides the flow diagram of systematic review and study selection, and Supplementary Data 5 lists all excluded effect sizes and studies with justifications.

## Data Extraction and Coding

The following information was extracted from each study: (1) year of publication, (2) publication journal (i.e., name of the journal, or unpublished manuscript/thesis), (3) sample size, (4) country in which the study was conducted, (5) experimental design, (6) design of repetition manipulation (e.g., within-participants, between-participants), (7) type of stimulus (e.g., trivia statements, news headlines...), (8) means and standard deviations for repeated and new statements, (9) reported effect size, (10) reported effect size's standard error, (11) correlation between repeated and new statements. For all outcome of interest that were not reported, the reviewers searched in open data and supplementary materials when available.

When a study included multiple conditions such that only some conditions met our inclusion criteria, we extracted data exclusively from those qualifying conditions. Specifically, if a study provided distinct outcome metrics for each condition or made raw data available, we retained only the eligible subsets that satisfied our criteria and excluded the ineligible ones. In instances where outcomes for eligible and ineligible conditions were reported only as combined results,

making it impossible to isolate appropriate data, we excluded the entire study to maintain methodological consistency.

In total, three reviewers collected data from every article independently: S.Y. coded data for all articles, M.G. coded data of half of the articles (first half ranked in alphabetical order of articles' first author name), and Y.A. coded data of the other half of the articles. All the 135 articles selected by title and abstract were at least double-checked, and the final 70 included articles based on full text check of inclusion and exclusion criteria were double-coded. The intercoder agreement for inclusion/exclusion was 91% (Kappa = 0.84). All disagreements between reviewers were discussed until consensus was reached.

### Extraction of Moderating Variables

To determine whether the size of the illusory truth effect was moderated by specific variables, we systematically coded a targeted set of information for each study. When recoding was necessary, especially in cases where theoretical guidance for categorization was limited, we ensured a balanced distribution of effect sizes across categories to ensure that each category was adequately represented. When a study included effect sizes potentially matching multiple categories of a moderator, we extracted each effect size individually whenever the data (e.g., distinct outcomes or open datasets) permitted clear separation. In instances where effect sizes were reported in a combined form and could not be disentangled by moderator category, we coded the moderator as missing. The intercoder agreement for moderating variables was on average 92% (ranging from 60% to 100%).

### Item type

We included item type (e.g., trivia, headlines, opinions) to examine whether the illusory truth effect generalizes across common real-world formats, especially given the prominence of news headlines and unverifiable claims in online misinformation. We coded the item type used in each study to examine differences in effect sizes across three categories: standard statements ($u = 309$), defined as any declarative sentences; headlines ($u = 43$), defined as items formatted as social media news headlines, and claims ($u = 12$) included unverifiable assertions (e.g., any kind of opinions or marketing claims).

**Repetition study design.** To evaluate whether the illusory truth effect varies based on repetition format, we examined whether participants were exposed to repeated and new statements within the same condition or in isolation. We coded whether the effect size was extracted from a within-participant ($u = 361$) or a between-participant ($u = 5$) manipulation of the repetition. In between-participant designs, two groups of participants evaluated either all-repeated or all-new statements during the test phase.

**Instructions during the exposure phase.** We examined the moderating effect of participants' task during the initial exposure to determine how different levels of cognitive engagement during encoding influence susceptibility to the illusory truth effect. Participants completed a range of tests (e.g., read, rate interestingness, categorize, rate veracity, rate familiarity, evaluate the source of information, evaluate belief or doubt, guess the gender of the author, rate humor, rate sharing intention, rate understanding, or shake/nod their head). Participants could perform more than one task (e.g., evaluate interest and veracity of a statement). These categories were recoded in three categories: truth judgment ($u = 78$, including all effect sizes in which participants had to rate veracity during exposure phase), passive processing ($u = 73$, including all effect sizes in which participants had to read, listen, or perform any action not requiring to process and manipulate information in working memory such as nodding or shaking their head), and irrelevant task ($u = 213$, including all effect sizes in which participants had to manipulate information in working memory, or perform a judgment unrelated to veracity rating).

**Truth warning at the exposure phase.** This moderator reflected whether explicitly alerting participants to potential falsehoods prior to exposure modulates the illusory truth effect. We coded whether participants were explicitly instructed that some or half of the statements were true or false in the exposure phase, including the presence of cues ($u = 163$), or did not provide any explicit warning ($u = 41$). To prevent any risk of miscategorization, we coded studies that did not specify whether participants were warned about the statement's veracity or did not provide the full verbatim of the instructions to participants as missing data ($u = 162$). Instructions to rate the truth of statements were not coded as explicit warnings.

**Truth warning at the test phase.** As warnings at test and exposure are likely to influence truth judgments through distinct cognitive mechanisms and serve different theoretical functions[27], we coded truth warnings at the exposure and test phases as separate moderators. We followed the same procedure as for the warning during the exposure phase, coding studies in which participants were warned ($u = 66$) or not ($u = 41$) that statements could be false in the test phase. Similarly, we coded missing data ($u = 259$) if the warning status is not specified (or respecified for the test phase).

**Veracity cues.** We included the valence of veracity cues as a moderator to examine the impact of cues that signal truth or falsity, which are commonly used in misinformation interventions such as fact-checking. We coded each effect size that involved any indication of a statement's factual status during the exposure phase. We classified veracity cues into three subgroups: true cues ($n = 33$), false cues ($n = 40$), or uncued ($n = 287$).

**Delay.** We examined the influence of delay between exposure and test to assess the temporal durability of the illusory truth effect. This moderator helps understanding the long-term influence of repeated information in both experimental and applied settings. We coded each study for the delay introduced between the exposure and test phases, from no delay to a few minutes to a few months, in 4 categories: no delay ($u = 135$, with effect sizes extracted from a procedure involving test phase immediately following exposure phase), within day ($u = 138$, including all effect sizes with a delay ranging from a few minutes to one day), up to a week ($u = 47$, including all effect sizes with a delay ranging from two days to one week), more than a week ($u = 41$, when the delay was superior to a week). When the delay was not specified, we assumed that the test phase immediately followed the exposure phase.

**Repetition type.** This moderator reflected whether exact repetition (verbatim) and paraphrased repetition (gist) differentially influence perceived truth, as these forms commonly occur in real-world contexts. We coded whether the statement repeated in the test phase was a verbatim ($u = 350$) or a gist ($u = 16$) of the statement presented in the exposure phase. When no information was provided, we assumed that the statements were verbatim repetitions.

**Modality of presentation.** We included the modality of presentation to understand how the presentation channel affects the illusory truth effect. This moderator accounts for variability in how content is encountered across studies and reflects real-world differences in media consumption. We coded whether the statements were presented visually ($u = 257$) (i.e., written) or verbally ($u = 62$) (i.e., read aloud). We coded the modality of presentation as missing data when the modality of the statement presentation differed between the exposure and the test phase (e.g., verbally in the exposure phase and visually in the test phase, $u = 25$) or when statements were presented both verbally and visually ($u = 17$).

**Presentation time during exposure**. This moderator reflected whether the duration of initial exposure affects the illusory truth effect. Exposure time is a key methodological factor that may affect encoding depth and fluency. This moderator also reflects real-world variability in how long individuals attend to repeated information, making it relevant for interpreting the robustness of the effect in ecological settings. We coded the presentation time of the statement presentation in the exposure phase. When the presentation time was not specified and the statements were presented visually, we assumed that presentation time was participant- paced. We coded the presentation time as missing data when the statements were presented verbally ($u = 5$). To ensure a balanced distribution of effect sizes across categories, we grouped the effect sizes based on presentation time: 5 seconds or less ($u = 77$), more than 5 seconds ($u = 92$), and participant-paced ($u = 179$).

**Presentation time during test**. To examine whether the amount of time available for truth judgments moderates the illusory truth effect, we examined the presentation time during the test phase. Varying test durations may influence the extent to which participants rely on quick, fluency-based heuristics versus more deliberative evaluation. We coded the extent to which the presentation time of the statements was the same between the exposure and the test phase. When not specified, we assumed that the presentation time was participant-paced. There was $u = 43$ effect sizes with a presentation time during test of 5 s or less, $u = 54$ of more than 5 s, and $u = 265$ of participant-paced.

**Response scale**. We coded whether the response scale was a Likert scale ($u = 269$), a binary scale ($u = 91$) (e.g., yes vs. no), or a slider from 0 to 100 ($u = 6$) to examine whether the precision of the response format used to evaluate truth judgments influenced the effect size. Additionally, we distinguished response scales based on whether they had an odd number of points ($u = 119$) or an even number of points ($u = 241$). The goal was to explore the moderating role of a neutral or midpoint option (e.g., 3 on a 5-point scale), versus even-numbered scales that force participants to express a choice that leans in one direction or another (i.e., there is no true midpoint).

**Proportion of true statements**. We included the proportion of true statements to evaluate whether the overall truth base rate within a study influences participants' sensitivity to repetition. This moderator captures variation in the informational context (e.g., all-true, all-false, or mixed item sets), which may affect the baseline likelihood of endorsing repeated items as true. To capture design variability across all-true, all-false, and mixed item sets, we created a continuous moderator (range 0–1) that records the proportion of factually true statements in each study's stimulus pool (e.g., 0 = all false; 0.50 = 50 % true/ 50 % false; 1 = all true).

**Veracity**. We investigated whether repetition increases perceived truth for true vs. false statements equally which relates to the illusory truth effect in the misinformation context. Effects sizes obtained using factually true ($u = 83$) or false ($u = 81$) statements were coded accordingly. In cases where the statement could not be true or false (e.g., opinion-based statements), the effect size was recoded as none ($u = 12$). We coded the veracity as missing data when the factual veracity of a statement was not indicated or when the statements were a mix of true and false statements ($u = 190$).

**Testing environment**. We examined the effect of testing environment to assess whether the data collection setting (online vs. face-to-face) introduces variability in effect sizes. This is especially relevant considering the increasing use of online platforms in the illusory truth effect research. It also helps evaluate the generalizability of laboratory findings to digital contexts where repeated misinformation is most encountered. We coded whether data were collected online ($u = 169$) or in face-to-face ($u = 196$).

## Missing Data in Moderating Variables
When data on the moderating variables considered in this meta-analysis were not directly available, the reviewers examined open data and supplementary materials when available. Missing data were then imputed using multivariate imputation by chained equations with "mice" package[47] in R. This method allowed us to preserve the relationships between variables while avoiding data losses associated with deletion methods. During the imputation process, we generated 20 datasets ensuring the robustness of the imputation. This approach allows to overcome the risks of producing inaccurate results when running the meta-regression due to the number of missing data for moderating variables[77].

## Effect Size Calculation and Missing Outcome of Interest
The outcome of interest was the standardized mean difference $g$[78] of the truth judgment in the test phase of the new statements and of the ones already presented in the exposure phase. To compute $g$, we first computed the standardized mean difference $d$[79] by dividing the mean difference in a given study by its standard deviation[80]. For missing effect sizes, we followed the general guidelines of the Cochrane handbook for systematic reviews of interventions[81]. When the effect size was not reported, we estimated $d$ from the mean and the standard deviation reported in the study. The standard error of the effect size was computed based on the sample size of each study to assign a weight to each of them. When the exact number of participants in each group in a between-subject design was missing, or could not be estimated (e.g., from open data or df), we assumed that the two groups included the same number of participants. Missing effect sizes were thus computed as follows[80]:

$$d = \frac{\bar{Y}_1 - \bar{Y}_2}{SD_{within}}$$

with standard deviation within groups computed as follows:

$$SD_{within} = \frac{SD_{diff}}{\sqrt{2(1 - r)}}$$

the standard deviation of the difference as follows[81]:

$$SD_{diff} = \sqrt{SD_1^2 + SD_2^2 - 2 \times r \times SD_1 \times SD_2}$$

and the standard error of the effect size as follows:

$$SE_d = \sqrt{V_d} = \sqrt{\left(\frac{1}{n} + \frac{d^2}{2n}\right)2(1 - r)}$$

When the standard deviation of each group was not provided, we estimated one based on the median of the truth judgment for the new and repeated statements in the test phase in the other included studies using a similar scale. Given that the standard mean error $d$ tends to overestimate the effect size of small samples, we computed the Hedge's $g$[78], an unbiased estimate of $d$. We applied the following correction using the correction factor $J$[80]:

$$J = 1 - \frac{3}{4(n - 1) - 1}$$

We obtained $g$ as follows:

$$g = J \times d$$

And we computed the standard error as follows:

$$SE_g = \sqrt{V_g} = \sqrt{J^2 \times V_d}$$

For studies adopting a between-subject design to evaluate the illusory truth effect (i.e., in which truth judgment was compared between a group exposed to statements repeated from the exposure phase and a control group exposed to new statements), we used the formula proposed by Borenstein et al. [80]. to measure the standardized effect size.

We also conducted a sensitivity analysis using Becker's formula to compute the standardized mean difference, which only requires standard deviations from the first assessment and therefore avoids the need to impute correlations between repeated measures[82]. This approach yielded effect sizes close to those from Borenstein's formula but with slightly wider confidence intervals, reflecting greater uncertainty (see Supplementary Table 2). We selected Borenstein's method for the main analyses because it yielded more stable estimates across studies, which facilitates interpretation and comparability of effect sizes in the synthesis. However, we acknowledge that its greater precision is contingent on the imputation of missing correlations, which may lead to potentially overconfident uncertainty estimates.

### Missing Outcomes of Interest and Sensitivity Analyses

Considering that: (1) calculating a standardized mean difference in a repeated measure study design requires the correlation ($r$ coefficient) between matched groups (i.e., repeated and new statements)[80], (2) none of the included publication reported the correlation coefficient in the full-text article, and (3) some of the included studies that did not report the effect size also did not report the standard deviation of both repeated and new statements, we computed missing values (i.e., standard deviations and correlation coefficients) from previous studies and conducted sensitivity analyses to measure a range of uncertainty.

We estimated missing standard deviations with the median of the standard deviations reported in other studies using the same measuring scale for truth judgment in the test phase (e.g., binary scale, 4-point Likert scale…). Missing correlation coefficients were computed in a similar way: we computed the coefficient of correlation between repeated statements and new statements when data were available. Otherwise, we imputed the median of such coefficients of correlation.

Sensitivity analyses were performed to determine whether our results would be affected if, instead of using the median, we used the minimum or maximum of the standard deviations or correlation coefficients of other studies to impute missing data.

### Statistical Analyses

We applied a random-effect approach, in light of the between-study heterogeneity reported in the previous meta-analysis[2]. We used a three-level meta-analysis model[40,41] to control for the dependence between the effect sizes reported on the same sample in studies manipulating several potential moderating factors. This method appropriately handles the clustering of effect sizes within individual studies and allows for estimation of between-study and within-study variance.

In a three-level meta-analysis model, the variance within each study is adjusted from the number of reported effect sizes. There are three different sources of variance: the between-study level of variance (level 3), the variance between effect sizes extracted from a same study sample (cluster) (level 2), and the variance in the sampling error for each study (level 1). We used the restricted maximum-likelihood (REML) random model to estimate the model parameters. The Knapp-Hartung-correction was applied; thus, a two-sided $t$-test was used for testing our individual regression coefficients rather than a $Z$-test[83]. Meta-analyses and meta-regressions were performed in $R$ using the "rma.mv" function of the metafor package at a 5% significance level. In the meta-regression analysis, the between-study ($\tau^2$) and within-study ($\sigma^2$) variance components were estimated from the complete set of effect sizes and kept constant across moderator levels[80]. All statistical tests reported in this study were two-tailed, and all confidence intervals are two-sided.

We additionally quantified the amount of between-study heterogeneity explained by the moderator set with the pseudo-$R^2$ statistic[84]:

$$\text{PseudoR}^2 = \frac{\tau^2\text{RE} - \tau^2\text{ME}}{\tau^2\text{RE}}$$

Where $\tau^2_{RE}$ is the level-3 variance from the baseline random-effects model and $\tau^2_{ME}$ is the residual level-3 variance from the mixed-effects model that includes all moderators. The statistic reflects the proportion of between-study variance accounted for by the moderators.

### Risk of Bias

We used a version of the revised risk of bias tool for randomized trials (RoB 2)[43] which was adapted to the illusory truth effect experimental design for each study in five areas: (1) risk of bias arising from the randomization process, (2) risk of bias due to deviations from the intended interventions (effect of assignment to intervention), (3) missing outcome data, (4) risk of bias in outcome measurement, (5) risk of bias in selection of the reported results. An overall risk of bias was then computed for each study. This overall risk of bias was coded as low if all RoB 2 domains were low. If up to three RoB 2 domains were of concern, the overall risk of bias was coded as some concerns. It was coded as high if one of the RoB 2 domains was high, or if four out of 5 domains were of concern. Two reviewers (i.e., S.Y. & M.G.) undertook the task of coding independently a random set of 10 included articles. The inter-reviewer agreement for the overall risk of bias measure was strong (Kappa = 0.83). All disagreements between reviewers were discussed until consensus was reached. The remaining articles were coded by S.Y.

### Inclusion and Ethics

This study is a meta-analysis of previously published experimental studies and did not involve new data collection or interactions with human participants. Inclusion of studies was based solely on pre-defined eligibility criteria, without regard to the geographic origin of the research. Ethical approval was not applicable to this study, as it involved only the analysis of existing data. Informed consent and participant compensation were obtained in the original studies, as reported in the respective publications.

### Reporting summary

Further information on research design is available in the Nature Portfolio Reporting Summary linked to this article.

## Data availability

The data generated in this study is available in the Open Science Framework (OSF) repository under accession[85] https://doi.org/10.17605/OSF.IO/2DB8S. No raw individual-level participant data were generated in this study. The processed data underlying all analyses, including extracted effect sizes, moderator coding, and imputed datasets, are available in the OSF repository. No data are subject to ethical, legal, or commercial restrictions.

## Code availability

All analysis scripts used to perform the meta-analyses and generate the figures in this study are available in the OSF repository under accession https://doi.org/10.17605/OSF.IO/2DB8S.

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

## Acknowledgements

We would like to thank Y. Al-Bandakji for her help in double-coding during data extraction, as well as A. De Carvalho for his valuable feedback on systematic review processes. We would also like to thank E.L. Henderson for sharing her materials. S.Y., M.G., M.C., and G.B. received funding from FakeAd ANR grant ANR–21-CE28-0025. S.Y. received fundings from a CIFRE grant 2022/1463 administered by the ANRT (Association Nationale de la Recherche et de la Technologie). The funders had no role in the design and conduct of the study; collection, management, analysis, and interpretation of the data; preparation, review, or approval of the manuscript; or decision to submit the manuscript for publication.

## Author contributions

S.Y. contributed to conceptualization, methodology, formal analysis, investigation, data curation, visualization, writing—original draft

preparation, and writing—review & editing. D.A. contributed to methodology, formal analysis, visualization, and writing—review & editing. M.G. contributed to methodology, investigation, and writing—review & editing. A.C. contributed to formal analysis, visualization, and writing—review & editing. M.C. and G.B. contributed to conceptualization, methodology, supervision, project administration, and writing—review & editing.

## Competing interests

S.Y., M.C., and G.B. were authors of one study included in this review. However, this study was evaluated using the same inclusion criteria and risk of bias assessment as all other studies. The remaining authors declare no competing interests.
