## [Transparent Peer Review file · Nature Communications]

Systematic review and meta-analysis of the evidence for an illusory truth effect and its determinants

Corresponding Author: Professor Grégoire Borst

Version 0:

Reviewer comments:

Reviewer #1

(Remarks to the Author)

At the outset, we note that we have limited expertise in the statistical procedures of meta-analysis; we trust other reviewers will address the technical aspects.

Turning to the psychological issues, this manuscript makes useful contributions to the illusion-of-truth (ITE) literature. Our comments draw attention to aspects that should be addressed in a revision.

Introduction

LL27-32 - These lines (which describe a correction strategy for fake news and the possibility of corrections to backfire) seems tangential to the current focus of the paper. Additionally, the cited paper (Pennycook et al., 2018) doesn't investigate the effects of repeating fake news in the process of debunking it as the sentence seems to imply. I would recommend the authors stick to the focus on the effects of information exposure here (where the Pennycook et al., 2018 would still be relevant).

LL33-34 - regarding "The illusory truth effect or repetition-induced truth effect is not restricted to fake news: any repeated statement tends to be judged truer than novel statements" - given there have been some exceptions noted (e.g., highly implausible/ plausible claims - Pennycook et al., 2018; Fazio, Rand, & Pennycook; 2019; headlines in the form of questions, Calvillo & Harris, 2022), we would tone down this claim a bit from "any repeated statement"

LL64-68 - These lines suggest that inferring truth from processing fluency is always a misattribution. That is not the case. Fluency is a valid correlate of all major truth criteria from coherence to compatibility with one's knowledge, social consensus, and so on. The problem is that people are insensitive to the source of the fluency experience and misread incidental fluency (e.g., from the readability of a print font) as indicative of coherence, consensus, and so on (Schwarz & Jalbert, 2020, in Greifeneder et al., Fake news, Routledge).

LL70-76. – Empirically, manipulations of color contrast have a smaller effect than manipulations of repetition. However, it does not follow from this observation that "conceptual fluency" beats "perceptual fluency." The latter conclusion about types of fluency (rather than specific singular manipulations) requires a calibration of the implementations of color contrast and repetition, ensuring equal subjective experiences of ease of processing. Without this calibration, all conclusions are limited to the manipulation and general inferences about types of fluency are unwarranted (although offered in the papers you cite).

Methods

LL159 following. – What is the rationale for excluding statements that were "not ambiguous" (common knowledge facts) and opinions? It would be worthwhile to include comparisons across these types of statements along with the present results. Piecemeal publication of these aspects in separate papers is not desirable.

In addition, how were "common knowledge facts" defined in making these exclusions? Was a full study excluded if participants judged a mix of "common knowledge facts" and ambiguous information? Studies may just report the mean truth of the selected claims from pretesting with a mix of better-known and more ambiguous information. In addition, studies may

not have norming data collected in the sample the study was conducted in, making the determination of whether or not they include claims that are “common knowledge facts” in that population impossible to assess.

Finally, it is unclear why only ambiguous information is of interest, given the explicit goal of discussing practical implications regarding the current misinformation crises. Does repeating unambiguously false information nevertheless increase acceptance?

LL161-171 - Can you clarify whether all data from a study were excluded if some (but not all) conditions met the exclusion criteria or if only the condition that met the exclusion criteria was excluded while the others were maintained? It currently sounds as if all conditions were excluded which suggests a loss of relevant data.

LL256-262 - The impact of warnings depends on whether the warning occurs at exposure or test. For example, Jalbert et al., (2020) found that warnings only moderate the effect of repetition when they occur at exposure and have no effect when presented only at test (probably because asking for a truth judgment itself acts as a type of warning). There are important theoretical reasons to suspect that the timing of this warning is critical and that warnings at exposure and test should be analyzed separately rather than pooled into the same category.

The ms also flags that many earlier truth effect papers do not contain sufficient information to determine whether or not a study included a warning as part of the standard instructions. Assuming that participants did not receive a warning when the paper doesn't include information regarding these instructions may miscategorize the studies. It would be good to add an analysis that only includes studies with vs. without warnings at exposure to assess the effect of warnings. .

LL229 - The size of the illusory truth effect depends on whether a within-ss or between-ss design is used. The effect is larger when participants judge a mix of repeated and new claims than when they judge only repeated claims (Dechêne et al., 2009 and related studies). It is therefore surprising that this meta-analysis apparently includes within-ss and between-ss studies without consistently differentiating between them. Within vs. between designs should be treated as an important additional moderating variable.

LL301-302 - Why was the response scale coded by odd vs. even? Explaining the reason would be useful.

LL304-305 - Why was data containing a mix of true and false claims excluded from the veracity comparison? This represents a large portion of the data and comparing true vs. false vs. mixed could provide useful insights.

LL379 - We may have overlooked this, but we didn't see whether or not effect sizes were fixed across subgroups? We would expect them to be fixed given recommendations by Borenstein et al. (2009) when the number of subgroups is small.

Results

Overall, the substantive results are useful and make a valuable contribution to the literature. Some details:

LL450 following. – This paragraph is uninformative. We cannot meaningfully relate this material to specific variables, even with the help of the supplementary material.

All p-values should be reported to the same level of specificity (some have 2 decimal places and some have three).

It would also be useful to provide degrees of freedom and corresponding estimates of ds for each subgroup in the main text (or the difference between ds)

Discussion

We appreciate the comparisons of the present results with the earlier meta-analysis by Dechene et al. and are largely in agreement with the interpretations offered.

LL550-551 - Jalbert et al. (2020) discuss ways in which making an initial truth evaluation at exposure (similar to a warning) may decrease the size of the illusory truth effect goes beyond the depth of processing account offered here. Asking people for a truth judgment necessarily entails that some of the claims may not be true.

LL590-597 - The studies included in this paper do not include multiple truth judgments (i.e. an initial one and a later one) but the explanation provided is specific to this scenario. What about when participants do not complete multiple test phases?

LL639-653 – When the fluency experience is (correctly or incorrectly) attributed to an incidental variable (e.g., repetition), the informational value of fluency is undermined. Awareness of the influence is more likely after very short delays (for a discussion, see Schwarz, Jalbert, et al., 2021).

LL692-695 - Several recent papers report that debunking messages that include a repetition of false information do not always result in backfire effects (e.g., Ecker et al., 2020, Briony Swire-Thompson et al., 2020).

These issues can be addressed in a revision that would further increase the contribution of this manuscript.

(Remarks on code availability)

Reviewer #2

(Remarks to the Author)

(Remarks on code availability)

Reviewer #3

(Remarks to the Author)

This is an extremely relevant article for those who approach the field of truth illusions. As a participant in the field, I appreciate the updated meta-analytic review work relative to the 2010 work by Dechene and colleagues, and the systematic map created by Henderson et al. in 2022.

However, despite claiming its relevance, I believe the article has room for improvement, and I would like to see these revisions made before recommending its publication.

My first point refers to the objectives of conducting this meta-analysis (p. 6). These should focus more on the gains it brings rather than on the deficits that previous approaches might present. While gains and deficits are related, I would recommend placing greater emphasis on what is gained. The argument that the advantage comes from greater rigor in estimates, enabled by the developments in the meta-analytic approach, is weak on its own. The studies are now more numerous, with greater variability in conditions, allowing for more consistent estimation of moderators' roles and the inclusion of new moderators that were previously unexplored, among other benefits. These are all relevant arguments in favor of a new meta-analytic study. However, it is precisely in this area where the paper loses some strength—it adds very little to what we already know. There is no reason for this. Since 2010, many other questions have been raised about the effect, and these can be tested in a meta-analytic context. By framing the paper solely around emphasizing the rigor of effect estimation, an opportunity is being missed here.

In my reading I would have liked to find not only evidence that corroborates or challenges Dechene's conclusions but new evidence. For example, Dechene discusses the difference in effect size when the contrast is within-subjects versus between-subjects, highlighting that the fluency/familiarity experience may be relative. Since 2010, some studies have directly addressed this issue, providing data that could be included in this new meta-analysis. With this approach, the present paper would add secure knowledge to the previous paper adding content and not only more rigor in the estimation methods. There are also references in the original paper to differences in effect magnitude between repetition and fluency manipulations, and since then, more studies have been published that use both types of manipulations. I do not see familiarity being contrasted with fluency in this analysis. Another very relevant aspect that could not have been addressed 14 years ago is the nature of the information. When identifying the effect, the statements made were ambiguous. Today, many studies not only present factually false information but also add warnings, etc. These data are not addressed here.

Essentially, what I think diminishes the relevance of this paper is that the moderation analyses are neither directly contrasted with the status quo provided by previous analyses nor clearly theoretically oriented. The analyses would gain relevance if they directly or indirectly answered theoretically significant questions (which would also facilitate and better integrate the data discussion). For instance, more than the instructions and the time spent reading the sentences at the first moment, it is the question of whether these factors may influence the explicit use of memory as information about the validity of a statement. If the effect is smaller when the instructions make it explicit that memory is not informative (with ½ true sentences and ½ false sentences presented at the start) than when nothing is stated, and this happens especially when there is encoding time, such a test could examine whether the effect depends on explicit versus implicit memory. Such kind of questions would induce to test the interaction between moderators and not only their main effects.

In the same line of thought, I believe that the analysis contrasting the magnitude of effects between online and in-person conditions is presented without contextualizing its relevance (pragmatically to alert that the conditions do not allow the effect to be detected, or theoretically because we know the level of motivation of the respondents is different?). Therefore, it lacks a true discussion of the results.

Personally, I find some conclusions drawn from the analysis of studies with very different methodologies concerning. For example, what happens when the authors conclude that the most relevant phase for deep processing is encoding, with retrieval not being relevant. In the retrieval phase, the time of sentence presentation is considered independently of whether they are presented immediately or one month after the study phase. In neither of these contexts does the exposure time influence retrieval conditions. One reason is that the information is still accessible in working memory or almost so, and in the other case, explicit accessibility does not improve with sentence presentation time.

I believe that the text would benefit from better readability if the results of the moderation analyses were presented in Figure 4 within a table, and if on that table they were contrasted with previous meta-analyses, highlighting what differs and what is new.

Please clarify, because I see no reason why the statistics are not presented in a more complete way, with their respective degrees of freedom and effect sizes. I would like to point out that when $t < 1$, the p-value calculation is unstable and unreliable, so it does not need to be presented (see, for example, Voelkle et al., 2007).

Finally, when I finished reading the discussion, I realized that the article could gain more consistency if it were to adopt a theoretical or pragmatic objective instead of an unbalanced mixture of both. When I read it, my interest was theoretical, and I wanted to understand the cognitive mechanisms underpinning the effect. But the article might aim for a pragmatic framework. In that sense, it is important to understand that misinformation goes beyond the truth illusion effect and should not be conflated. But if this is the main concern, it would be important to understand what the moderators say about the possibility of spreading fake news or how we can prevent it.

I also have two small details that could assist in the more detailed revision of the presented text:

Line 37: "after a certain delay" should be changed to "In most studies after a certain..."

Line 39: "... and new statements previously unseen in a random order"

(Remarks on code availability)

Reviewer #4

(Remarks to the Author)

General Assessment

This meta-analysis presents a comprehensive and updated synthesis of the illusory truth effect, incorporating 108 studies and 195 effect sizes—a significant expansion compared to the previous meta-analysis by Dechêne et al. (2010). The authors employ a three-level meta-analysis framework to account for dependencies among effect sizes within studies, use meta-regression to explore sources of heterogeneity, and conduct sensitivity analyses with multiple imputation to address missing data under the MAR assumption. These methodological advancements strengthen the reliability of the findings. From my perspective as a statistical reviewer, the study implements several best practices in meta-analysis. However, there are specific aspects related to effect size calculation, heterogeneity, and small-study effects that could benefit from further refinement and discussion.

Points of Critique

1. Standardized Mean Differences and Imputation of Missing Data

The authors rely on standardized mean differences (Hedges' g) as their primary effect size metric. While group means are often available, the standard deviations and correlations needed for standardizing within-group designs are frequently missing. To address this, the authors use imputation from external sources. However, an alternative approach that might reduce uncertainty in the pooled estimates is the method proposed by Becker, which only requires standard deviations from the first assessment occasion (see Morris & DeShon, 2002). This would eliminate the need for imputed correlations and likely yield more less uncertain effect size estimates.

2. Heterogeneity and Potential Multiplicity in Analytical Methods

High heterogeneity is common in psychological meta-analyses, particularly those involving RCTs. Beyond variation in population, intervention, and outcome characteristics, one likely source of heterogeneity is the diversity of analytical methods used in the original studies to derive effect size estimates. The authors report substantial within- and between-study heterogeneity ($I^2 \approx 71\%-81\%$), which raises interpretational concerns about the aggregated effect size. Additionally, reliance on standardized mean differences may further contribute to heterogeneity, as prior research has suggested (e.g., <https://psycharchives.org/en/item/1b670ec4-fec9-43d8-b72c-ecb85b27d0e5>). I recommend including a critical reflection on how potential hidden multiplicity and standardization might inflate heterogeneity in the discussion section. With regard to the impact of standardization, meta-regression on the coefficient of variation might be considered as another worthwhile sensitivity analysis.

3. Small-Study Effects and Publication Bias

The study acknowledges the presence of small-study effects, which the authors attribute to publication bias. However, while the presence of small-study effects is noted, no corrective methods (e.g., PET-PEESE, selection models) are applied. Given the substantial heterogeneity, publication bias is a serious concern, and the lack of adjustment likely means that the reported effect size estimates are inflated.

Based on the evaluation of small-study effect corrections by Carter, Schönbrodt, Gervais, and Hilgard (2019), I recommend implementing the PEESE method. The authors claim in the discussion section, that such methods cannot be implemented with multi-level meta-analyses. I do not agree, because the PEESE approach is essentially an ad-hoc meta-regression on sample precision. As such it is supposed to provide a pragmatic and conservative correction under conditions of questionable research practices and high heterogeneity.

Additionally, I suggest using this adjusted model as a baseline before adding substantive moderators. This would allow for a clearer interpretation of the impact of study characteristics on the illusory truth effect, free from confounding small-study biases.

4. Reporting of Meta-Regression Results

While the authors conduct a meta-regression to account for moderator effects, there are two areas where reporting could be improved:

- Moderator Impact on Effect Size Scale: Regression coefficients should be explicitly reported in the main text (rather than

supplementary materials) with confidence intervals to provide readers with a clearer understanding of how each moderator influences the standardized mean difference. A modification of Figure 4 to include these coefficients would be particularly useful.

- Proportion of Heterogeneity Explained: The proportion of between-study variance accounted for by the meta-regression should be reported (e.g., pseudo R^2). This would provide insight into the extent to which the included moderators successfully explain the observed heterogeneity.

5. Visualization of Effect Heterogeneity

The authors include a funnel plot for assessing publication bias, that could be improved with regard to the contour enhancement proposed by Peters, Sutton, Jones, Abrams and Rushton (2008). Specifically, the contours are currently constructed around the naïve pooled effect estimate (that is unadjusted for small study bias). Instead, they should be constructed around to $H_0: g = 0$ effect scenario, to facilitate the identification of those studies, that reported effect estimates just below the conventional 0.05% level. Precision-adjusted or naïve pooled effect estimates (and their CI regions) could then simply be added as a top-layer of figure elements.

Additional visualization of the data would further improve interpretability:

- Forest Plots Stratified by Key Moderators: Presenting supplementary forest plots ordered by major moderators (e.g., exposure task and delay, truth warning) would allow readers to see how effect sizes vary within each subgroup.

Conclusion

Overall, this meta-analysis provides a rigorous and updated synthesis of the illusory truth effect. The authors should be commended for their methodological choices, particularly in their use of three-level modeling and sensitivity analyses. However, I encourage further methodological refinement in the areas of effect size calculation, heterogeneity interpretation, and small-study effect adjustments. Addressing these points would enhance the robustness and interpretability of the findings.

Recommendations for Revision

1. Consider Becker's method for standardizing mean differences in within-group designs to reduce imputation uncertainty.
2. Reflect on the impact of standardization and hidden multiplicity on effect size heterogeneity in the discussion.
3. Implement small-study effect corrections, particularly the PEESE method, to provide more conservative effect size estimates.
4. Report regression coefficients and the proportion of heterogeneity explained by meta-regression in the main text.
5. Modify funnel plot and provide supplementary forest plots stratified by moderators.

My own brief screening using PubMed and the search term from the paper returned several new primary studies on the illusory truth effect that have been published since Sep 2022. Although I don't assume, that those new studies will contribute substantially more information than currently compiled by the authors, an update of the meta-analysis would definitely be worthwhile.

By integrating these refinements, this paper will provide an even stronger contribution to the literature on the illusory truth effect.

(Remarks on code availability)

Version 1:

Reviewer comments:

Reviewer #2

(Remarks to the Author)

I was one of the first two reviewers from the original review. I greatly appreciate the efforts the authors have put into updating their manuscript based on the reviewers' suggestions. I believe that significant improvements have been made to this manuscript.

I do have some concerns and suggestions regarding the new "veracity cue" moderator. I also have a number of more minor points following the updates that have been made.

New veracity cues moderators:

In response to our question about the exclusion of common knowledge facts and opinions, the authors added back in opinions (claims), effect sizes previously included for being considered common knowledge (widely known facts, highly implausible claims). In doing these additions, the authors also decided to analyze effect sizes that included veracity cues at exposure.

The addition of the studies using explicit veracity cues was not something we had mentioned, but I believe that including these studies provides really valuable information and greatly appreciate this addition. However, I'm having a hard time following what these categories entail. For example, what was included in "post-rating feedback"? Is this always occurring at the item level? What is the difference between a "positive cue" (category in the results) and a "true cue" (category used elsewhere)? What about studies where some items had cues and others didn't – were these included? In the analysis comparing claims labeled as false or true, is this including all types of "false cues" and "true cues" before/during/after ratings, or just certain ones? The analysis refers to "explicitly labeling claims as false at exposure" but this seems like this would just be a subset of the "false cues" just ones where claims were explicitly labeled as false. The authors also refer to veracity cues as "explicit truth indicators", but some of them (like including "most people know that..." in a claim) do not provide definite information about truth, while others do.

In the discussion, the authors also report veracity cues always attenuating illusory truth: (p. 18 –1 397): "Crucially, our findings suggest that explicit cues about the truth status of a statement significantly reduced the illusory truth effect." However, the results reported indicate that, if anything, explicitly labeling claims as true INCREASES the illusory truth effect. Even if this effect wasn't robust, it doesn't seem to be the case that the authors find labeling claims as true decreases the effects of repetition.

The authors also mention in the discussion (line 408) - "However, variability in the type of veracity cue (e.g., warning labels, post-rating feedback, or epistemic qualifiers) likely contributes to the inconsistent findings, especially given the substantial between-study heterogeneity observed in our analysis." I think this is an important point, and given this, if the authors are to keep in these studies, I would encourage them to i) more clearly define the different types of veracity cues and their categorization and ii) present effect sizes for different types of cues separately.

Additional points

- Moderator justification:

I appreciate the authors adding more justification for their choice of moderators. This information is currently in the methods at the end of the paper, but I think it would be more useful to have this information earlier, either in the section at the end of the intro that lays out the moderators of interest or in the results section as the results are presented.

I also think it would be useful to have the number of effect sizes per category reported with the results along with the number of effect sizes that were not included in that analysis. Given that some of the categories have very few effect sizes – e.g., $n = 6$ for balanced cues and 100-point sliders — and it would help put these results in context.

I think that Table 2 is an especially useful addition. I also think including the number of effect sizes not included in each moderation analysis would be helpful information.

- I'm a bit thrown off by the opening sentences focusing on fake news (and not even fake news being repeated) given the paper is meant to be a review of the illusory truth effect broadly and most of this work doesn't use fake news as stimuli (as the next paragraph discusses).

- p. 3 lines 29-32 – I know the wording of this sentence was updated based on another reviewer's comments, but I'm now having trouble following it. As currently written, this sentence could be read that in some studies there is no test phase? Or that in some studies, participants aren't instructed this?

- p. 12 – lines 241 – The authors first refer to one of the categories of task at exposure as "passive processing" and then later in the paragraph refer to (what I believe to be the same category) as "instructed to read the statements". From the methods section, it seems like passive processing included other tasks like listening to the statements, so updating "instructed to read the statements" to "passive processing" seems more accurate here

- p. 12 line 249 – why does this analysis specifically mention "after controlling for other moderating variables" when the other analyses do not specify this? Were all other moderating variables controlled for?

- p. 13 – line 269-270 – What is the condition being compared to the delay of over a week here? The delay of less than a day?

- p. 14 – lines 304-306- this statement – "Similarly, the proportion of true or false statements showed no influence on the effect size ($t(245) < 1$), indicating that the illusory truth effect is comparable in studies that uses mixed, all-true, or all-false item pools."

Given this was a continuous moderator, would it be more appropriate to report that the illusory truth effect is similar across different proportions of true vs. false claims? Before reading the methods, I thought these were three distinct categories being directly compared to each other rather than a continuous measure.

- I had trouble following the discussion starting on p. 15 line 330. Particularly, I'm not seeing how these findings necessarily challenge the assumption of the referential theory – elaborating on whether or not something is true and elaborating on a statement in some other way with the assumption that it is true should each result in the formation of different links/ strengths of links between referents in memory. The referential theory is not just about deeper processing leading to stronger memory traces, but also whether that processing results in an excitatory or inhibitory link between referents.

- p. 20 lines 447-449– on this statement - “This finding suggests that factors such as experimenter presence, environmental control, or potential distractions in online settings do not substantially influence cognitive processes involved in the illusory truth effect.” – while there may have been no differences observed in the aggregate, as these different potential factors weren’t looked at separately, I would be careful making this conclusion, especially given evidence regarding the impact of divided attention on the size of the truth effect (e.g., - Ly, Bernstein, & Newman, 2024). It could be that different factors influenced participants in different settings in different ways, but these washed out in the overall comparison.

– p 32, lines 734-736 “When not specified, we assumed that the presentation time was similar except if the modality of presentation changed between the exposure and test phases.” – I’m wary about the accuracy of this assumption given it would result in a mis-categorization of my own studies. I think many truth effect studies present claims for a certain amount of time at exposure and specify this, but then at test let participants respond to them in a self-paced way. However, especially if participants were simply reading claims and not making time-restricted judgments at exposure, papers may not explicitly say these test judgments are self-paced because it would be implied that this is the case if no time restrictions are mentioned at any point for judgments.

(Remarks on code availability)

As noted in the original review, I have more limited expertise in the statistical procedures of meta-analysis and trust other reviewers will address the technical aspects. The statistical additions that were made during this revision were largely those I am not familiar with.

Reviewer #3

(Remarks to the Author)

In this resubmission, the authors present a manuscript that differs substantially from the previous version. They have not only expanded the set of studies analyzed but also addressed moderators that are highly relevant for readers seeking to understand the phenomenon(s) under investigation. Additionally, there is a noticeable improvement in the transparency of the procedures, which facilitates a more critical evaluation of the results.

However, my main concern with this version is that it still lacks a clear and explicit statement of the article’s objectives, as well as a more coherent theoretical integration. I could not find a paragraph that clearly articulates the aims of the paper. In the paragraph beginning on line 88, the authors frame their critique of the previous meta-analysis as an “implicit objective,” suggesting their goal is to “do it better.” In my view, this framing is unnecessary. The earlier meta-analysis is simply outdated, and that alone provides sufficient justification for a new one. The intention to improve upon prior work does not need to rely on discrediting it.

To address this issue, I suggest that the authors insert a paragraph between lines 101 and 102 explicitly stating their objectives. These could include (if authors agree): a) updating the meta-analysis with more recent data; b) addressing specific limitations of the previous meta-analysis; c) clarifying the role of moderators relevant to different theoretical perspectives on the effect; d) incorporating an approach with practical relevance, particularly regarding “fake news” and strategies for its mitigation.

I note that the Discussion section opens with and places particular emphasis on the final objective listed above: fake news” and strategies for its mitigation. If this is indeed intended as the article’s primary focus, then the Introduction should be revised to make this emphasis more explicit—guiding the reader to understand the relevance of the illusion of truth effect to the topic of fake news, while also informing them about other contributing factors that influence its success and the strategies available to mitigate it. Alternatively, if the authors prefer to retain the current structure of the Introduction, I recommend that the Discussion be expanded to address all the stated objectives, ensuring alignment between the framing and the conclusions.

I believe the Discussion section would benefit from better organization, supported by a clearer conceptual framing and a more cohesive argument—both of which should be grounded in a well-defined set of objectives (possibly those I have suggested). I am also concerned about the somewhat assertive tone in the Discussion, where certain hypothetical or interpretative points are presented as established facts. While the authors provide their interpretation of the data, they should also acknowledge the possibility of alternative explanations. The Discussion would gain in relevance and scholarly value if the findings were more explicitly connected to theoretical implications—for example, by raising questions about whether the data support or challenge specific frameworks—rather than presenting conclusions without sufficient nuance.

It is clear for me that I believe this article should ultimately be published, as it addresses a relevant topic and is likely to make a meaningful contribution to the field, particularly given its nature as a meta-analysis. However, precisely because of its potential impact, it is essential that the article presents a more coherent set of objectives and that the authors attend to other important details that remain insufficiently clear in the current version.

It is a fact that throughout the article the authors have corrected erroneous statements made about the studies presented or about the field, which has greatly improved this version of the manuscript. Some of these errors escaped me during the first review but were pointed out by my fellow reviewers. I understand that there are many articles involved, and the lack of a conceptual framework for them—one for which methodological and conceptual details of the studies are relevant—causes the authors to overlook the importance of some imprecise statements. This concerns me because such imprecisions often become entrenched misunderstandings of phenomena that can take years to correct, once a respected figure in the field decides to analyze the original sources. Writing a review article thus carries great responsibility.

Below I list some details I have identified, though I fear this list is not exhaustive given the extensive number of articles reviewed:

1. The first example appears in the change made at lines 22–24, where the authors attribute the term “illusory truth effect” to Pennycook et al. (2018). The term “illusion of truth” already appears in the original articles by Bacon and subsequent papers by Begg et al., and Garcia-Marques et al. later shortened to “truth effect” in publications by Unkelbach. Its relationship to “fake news” has a different history, with pragmatic rather than the original theoretical goals (which aimed at understanding

the human mind and not why its sensitive to fake news). Thus, the phrase “Repeated exposure to fake news can increase individuals’ perception of its accuracy, a phenomenon known as the ‘illusory truth effect’” would be more accurate if replaced by “Repeated exposure to fake news can increase individuals’ perception of its accuracy, a phenomenon theoretically related to the ‘illusory truth effect’ (Bacon, Begg et al.).”

2. The authors state that studies on truth effects use “ambiguous statements” (referring to people’s ratings tending toward the midpoint of the scale). I understand this usage, but in fact, those studies frequently use “neutral statements,” which are statements equally likely to be rated as true in both their true and false versions. Given the potential impact of this paper, I believe this distinction should be emphasized, as it may explain why studies sometimes show inconsistent results. One possibility is to state that studies use either ambiguous or neutral statements, with appropriate references.

3. The statement “the illusory truth effect is robust across various populations and tasks” would be better supported by citing Dechêne et al.’s meta-analysis, in addition to the specific studies referenced.

4. Line 45: The sentence “However, repetition does not always increase perceived truth for extremely implausible statements or headlines in the form of questions” suggests this applies only to those cases. Consider revising to: “This is the case, for instance, for extremely implausible statements or headlines in the form of questions.”

5. Line 48: The sentence “The dominant explanation of the illusory truth effect is the fluency account” would be more theoretically accurate if stated as: “Dominant explanations of the illusory truth effect attribute a significant role to processing fluency.” This is because in line 51, the process is described as a misattribution or association with valid cues, which is defended by those who explain the truth effect as involving a memory referent. Thus, although all current theories recognize the role of processing fluency, they differ in how fluency is integrated or interpreted by the cognitive system (as a direct cue, a learned association with validity, an affective experience, a byproduct of familiarity, etc.).

6. Lines 60–63: The hypothesis raised by the authors is plausible. However, please note that there are no data supporting this directly, whereas some data suggest there may be a dissociation between conceptual and perceptual fluency, not merely arising from the strength of the fluency experience. I recommend making this clear, and also explaining why you consider the dissociation an artifact (see for instance Garcia-Marques et al. 2017).

7. Line 344: I believe that there is a serious misunderstanding of the role of memory in the truth effect and the illusion of truth as it is assumed by the referential theory. Better memory is not expected to induce more “illusions” of truth. Instead, stronger memory integration is expected to have this effect only if the information is not already linked to a “validity” judgment in memory that can be retrieved to support the judgment. Also, there is no reason to assume that making a truth judgment implies better memory integration (making truth judgments is far from a synonymous of deep processing). An alternative interpretation to the one offered by the authors is that the effect is reduced in these cases because some participants rely on fluency to make the judgment, while others attempt to recover a previous judgment stored in memory or engaged in a different strategy. Therefore, I recommend that the authors avoid stating that their data “prove” the referential theory wrong. Instead, they could say: “If we assume that making a truth judgment involves a deep encoding process, then the results challenge the referential theory.”

I wish you success with the review and publication of the article, hoping that it will help clarify the literature rather than contribute to further lack of clarity and confusion.

(Remarks on code availability)

Reviewer #4

(Remarks to the Author)

I have reviewed the revised manuscript and the authors’ response to my comments. I appreciate the extensive revisions that have been undertaken. The authors have diligently addressed the major methodological concerns, and the manuscript has been improved considerably as a result.

In particular, I would like to highlight several key improvements:

- The updated literature search (which now includes studies up to early 2025) is a significant addition that enhances the relevance and comprehensiveness of the review.
- The implementation of a PEESE correction to address small-study effects and its use as a baseline for moderator analyses has increased the robustness of the reported findings.
- The discussion section has been strengthened by the inclusion of a more nuanced reflection on heterogeneity, including potential artifacts from standardization and multiplicity.

I have one final suggestion for a minor revision to improve the statistical accuracy of the reporting. It concerns the justification provided for using Borenstein’s formula over Becker’s for the effect size calculation.

The manuscript states that Borenstein’s method was chosen because its estimates were “more conservative”. This is a terminological inaccuracy. Statistically, a wider confidence interval (as produced by Becker’s formula in the authors’ sensitivity analysis) reflects greater uncertainty and is therefore considered more conservative. A narrower confidence interval is more precise, but can also be considered more liberal if that precision is an artifact of imputation, as may be the case here. The greater precision of the Borenstein estimator is likely a consequence of imputing missing correlations, which may lead to an overconfident (i.e., artificially narrow) estimate of the uncertainty.

Recommendation:

I suggest revising the justification on lines 820-821 to transparently acknowledge this trade-off and that the greater precision is contingent on the assumptions made during the imputation of missing correlation coefficients. This would clarify the

rationale while also transparently acknowledging the potential limitations of the chosen methods.

With this adjustment, the manuscript will be a robust and significant contribution to the literature. Hence, I recommend acceptance following this minor revision.

(Remarks on code availability)

Version 2:

Reviewer comments:

Reviewer #2

(Remarks to the Author)

I was Reviewer 2 on the last round of reviews. The authors have done an excellent job addressing my concerns, and I applaud their attention to detail when making these changes.

I have just a few minor points below, but I believe this manuscript is now ready for publication without another full round of review. Congratulations on this work!

- The authors find that the effect size for providing false cues is $g = -0.18$, 95% CI [-0.33, -0.03]. The exploratory analysis by type of cue also found that most cues had confidence intervals below zero. This means that the illusory truth effect was actually reversed, with repeated claims now being judged as less true than new claims. In discussing these effects, however, this reversal is not mentioned, and rather it is just reported that the size of the truth effect was decreased with false cues (e.g., from the discussion "False cues reduced the impact of repeated exposure on perceived truth while true cues did not affect the illusory truth effect"). This type of wording (the false cues decreased the size of the illusory truth effect) implies to me that there is still an illusory truth effect with false cues. I think it would be important and informative when presenting results like this to clearly communicate that it's not just that the size of the illusory truth effect is smaller, but also that it's reversed.

- I didn't check all the observed u that were added, but noticed that for false cues, the authors reported observed $u = 40$ in the results and observed $u = 41$ in the methods. Then, if you add up the observed u for the different types of false cues, they only add up to 37. The true cues, on the other hand, are reported as observed $u = 33$ in the results and observed $u = 32$ in the methods, but if you add up the true cues from the table, those equal 36. There seem to be either typos or coding issues going on here.

- In the abstract – saying "since 2006" in the abstract without context (that the previous meta looked for effects up to 2006) may be confusing to readers. As a very minor note, but in case it's not caught in copy editing, the effect size $g = .37$ is missing a "0" in front of the decimal.

- Minor wording suggestion: On p. 5, line 87 the authors write - "In particular, the illusory truth effect seems to be different" – I think saying "smaller" instead of "different" would work better here.

(Remarks on code availability)

Reviewer #4

(Remarks to the Author)

I thank the authors for their thorough revision. My queries have been satisfactorily addressed.

(Remarks on code availability)

Systematic review and meta-analysis of the evidence for an illusory truth effect and its determinants

Response to Reviewers

We thank the Reviewers for their constructive comments. All modifications in the manuscript were highlighted in yellow color. In the revised manuscript, we implemented the following changes to address the different issues:

1. Expanded our dataset to 366 effect sizes across 182 studies, including studies published up to February 2025.
2. Broadened inclusion criteria, by including opinion-based and non-ambiguous items.
3. Added several new moderators to capture methodological and contextual factors more precisely:
 - Item type: statements, headlines, or claims.
 - Veracity cues: positive, negative, balanced, or none.
 - Truth warning: we separated the coding for exposure and test phases.
 - Proportion of true statements in stimulus pool.
 - Study design: within vs. between-participant repetition manipulation.
4. Clarified our eligibility criteria and inclusion/exclusion of effect sizes drawn from studies with different methodologies based on the available data breakdown.
5. Updated our analytic strategy:
 - Applied small-study effect correction (PEESE) in the main meta-analytic model and subsequent models as a baseline.
6. Clarified our theoretical vs. practical scope by revising the Introduction and Discussion sections to better articulate the rationale and the aims of our meta-analysis.
7. Improved reporting transparency:
 - Standardized the reporting of p-values.
 - Included degrees of freedom, regression coefficients, and subgroup effect size estimates along with their 95% confidence intervals in the main text.
 - Created a summary table comparing moderator results with the previous meta-analysis (Dechêne et al., 2010).
8. Updated our tables and figures:

- Added a systematic contrast of our findings with Dechêne et al. (2010) (Table 2)
 - Modified the funnel plot (Figure 1) to construct the contour around $h = 0$ with precision-adjusted and naïve pooled effect estimates as layers.
 - Modified the visualization of the meta-regression (Figure 3) to describe regression coefficients rather than standardized Z values.
9. Updated the Supplementary Information with:
- A table describing studies by the five RoB2 domains
 - A sensitivity analysis of effect size computation method between Borenstein and Becker's formula
 - Forest plots stratified by key moderators
10. Incorporated clarifications requested by the Reviewers.
11. Reorganized sections and subsections according to Nature Communications' formatting guidelines.

Detailed responses to each point raised by Reviewers are provided below.

Reviewer 1 & Reviewer 2 Comments

• **Overall Comment:** At the outset, we note that we have limited expertise in the statistical procedures of meta-analysis; we trust other reviewers will address the technical aspects. Turning to the psychological issues, this manuscript makes useful contributions to the illusion-of-truth (ITE) literature. Our comments draw attention to aspects that should be addressed in a revision. [...]. Overall, the substantive results are useful and make a valuable contribution to the literature. [...]. We appreciate the comparisons of the present results with the earlier meta-analysis by Dechene et al. and are largely in agreement with the interpretations offered.

Response: We thank the Reviewers for their comments and the suggestions for improving our manuscript.

Introduction

1. **Comment:** LL27-32 - These lines (which describe a correction strategy for fake news and the possibility of corrections to backfire) seems tangential to the current focus of the paper. Additionally, the cited paper (Pennycook et al., 2018) doesn't investigate the effects of repeating fake news in the process of debunking it as the sentence seems to imply. I would recommend the authors stick to the focus on the effects of information exposure here (where the Pennycook et al., 2018 would still be relevant).

Response: We very much agree with the Reviewers' comment that the correction strategies are tangential to the main aim of the meta-analysis. Accordingly, we have revised the Introduction section and now focus only on the relevant aspect of repeated exposure increasing perceived truthfulness of fake news, i.e., namely the illusory truth effect. We no longer discuss the correction strategies and refer to Pennycook et al. (2018) to introduce the illusory truth effect as follows:

- Lines (22-24): "Repeated exposure to fake news can increase individuals' perception of its accuracy ⁶, a phenomenon known as the 'illusory truth effect'"
2. **Comment:** LL33-34 - regarding "The illusory truth effect or repetition-induced truth effect is not restricted to fake news: any repeated statement tends to be judged truer than novel statements" - given there have been some exceptions noted (e.g., highly implausible/

plausible claims - Pennycook et al., 2018; Fazio, Rand, & Pennycook; 2019; headlines in the form of questions, Calvillo & Harris, 2022), we would tone down this claim a bit from “any repeated statement”

Response: Initially, we formulated the sentence broadly because previous literature demonstrates that repetition can increase perceived truthfulness of statements in many forms, depending on the study design (e.g., Garcia-Marques et al., 2015; Lacassagne et al., 2022). Nevertheless, we fully agree that the phrase “any repeated statements” might be overclaimed. We have now toned down the claim as follows:

- Lines (25-26): “The illusory truth effect or repetition-induced truth effect is not restricted to fake news: repeated statement tends to be judged truer than novel statements.”
- Lines (45-46): “However, repetition does not always increase perceived truth for extremely implausible statements^{6,11}, or headlines in form of questions²⁷.”

3. Comment: LL64-68 - These lines suggest that inferring truth from processing fluency is always a misattribution. That is not the case. Fluency is a valid correlate of all major truth criteria from coherence to compatibility with one’s knowledge, social consensus, and so on. The problem is that people are insensitive to the source of the fluency experience and misread incidental fluency (e.g., from the readability of a print font) as indicative of coherence, consensus, and so on (Schwarz & Jalbert, 2020, in Greifeneder et al., Fake news, Routledge).

Response: We fully agree that using processing fluency is not always indicative of misattribution and can often serve as a valid heuristic for judging truth (e.g., Brashier & Marsh, 2020). Specifically, the illusory truth effect arises when individuals incorrectly attribute increases in fluency (e.g., due to repetition alone) to truthfulness. We have thus revised the manuscript accordingly to clarify this sentence as follows:

- Lines (51-53): “While fluency can be a legitimate indicator of truth when it aligns with valid cues^{30,31}, the illusory truth effect arises when people misattribute incidental increases in fluency from mere repetition or font as evidence of truth.”

4. **Comment:** LL70-76. – Empirically, manipulations of color contrast have a smaller effect than manipulations of repetition. However, it does not follow from this observation that “conceptual fluency” beats “perceptual fluency.” The latter conclusion about types of fluency (rather than specific singular manipulations) requires a calibration of the implementations of color contrast and repetition, ensuring equal subjective experiences of ease of processing. Without this calibration, all conclusions are limited to the manipulation and general inferences about types of fluency are unwarranted (although offered in the papers you cite).

Response: The Reviewers rightly note that concluding a direct difference between "conceptual" and "perceptual" fluency based solely on these manipulations might be premature. We have revised the sentence to more accurately reflect the complexity of such comparison by acknowledging that conclusions regarding different types of fluency are limited by the absence of direct empirical comparisons controlling for equivalence in fluency magnitude as follows:

- Lines (55-57): “[...], with studies indicating that manipulation of color contrasts often yield a smaller illusory truth effect than repetition^{33,34}.”
- Lines (60-63): “However, differences in effect size may reflect how strongly each manipulation induces subjective ease of processing, rather than a fundamental disparity between fluency types. Thus, inferences about which type of fluency is more influential remain limited to the specific context studied.

Methods

5. **Comment:** LL159 following. – What is the rationale for excluding statements that were “not ambiguous” (common knowledge facts) and opinions? It would be worthwhile to include comparisons across these types of statements along with the present results. Piecemeal publication of these aspects in separate papers is not desirable.

Response: The primary aim of our meta-analysis was to estimate the illusory truth effect in its most prototypical form (i.e., the influence of repetition on subjective truth judgments when the factual status of a statement is not readily known). We initially excluded certain types of statements for two main reasons:

1. Non-factual items (Exclusion Criterion 1): We excluded statements whose veracity cannot be objectively verified (e.g., opinions, rumors, or beliefs). Including items whose “truth” depends on personal preference or social consensus would introduce additional

variability (e.g., personal attitudes, cultural norms) and confound the standard illusory truth paradigm.

2. Non-ambiguous items (Exclusion Criterion 2): We excluded statements that were cued to be either clearly true or clearly false. Such statements offer explicit veracity cues thereby potentially lowering the overall effect size. Including them could have obscured the underlying processes of interest in the standard paradigm.

After careful consideration of your recommendation, we agree that it would be valuable to investigate whether repetition shapes perceived truth for well-known facts, opinions, or other statement categories with readily verified truth status. Therefore, exclusion criteria 1 was removed, and exclusion criteria 2 was modified to exclude only studies providing veracity cues during the test phase. Specifically, we revisited the effect sizes previously excluded under Criteria 1 or 2 and re-considered them for inclusion into our database. As a result, we retrieved 97 new effect sizes from 20 manuscripts, 11 of which had been fully excluded before. To accommodate these newly included data, we introduced two new moderators. First, an “item type” moderator distinguishes among standard fact-based statements, news headlines, and opinion-based statements. Second, a “cue” moderator captures whether veracity cues were provided (e.g., tags or explicit feedback) during the task. We also revised our “veracity” coding so that, in addition to “true” or “false,” we can track statements with no clearly established factual basis such as rumors or opinion statements (with the coding option “none”). This broader inclusion and expanded coding now allow us to provide a more comprehensive assessment of how repetition affects perceived truth across a vaster range of statement types and contexts. We have revised the following parts of our manuscript as follows:

See in the Method section:

- Lines (542-543): “We excluded studies that involved (1) veracity cues during truth judgment in the test phase, [...]”
- Lines (639-646): “**Item type.** We included item type (e.g., trivia, headlines, opinions) to examine whether the illusory truth effect generalizes across common real-world formats, especially given the prominence of news headlines and unverifiable claims in online misinformation. We coded the item type used in each study to examine differences in effect sizes across three categories: “standard statements” (u = 309), defined as any declarative sentences; “headlines” (u = 43), defined as items formatted

as social media news headlines, and “claims” ($u = 12$) included unverifiable assertions (e.g., any kind of opinions or marketing claims).”

- Lines (683-690): “**Veracity cues.** We included veracity cues as a moderator to examine the impact of explicit truth indicators (e.g., labels or statements like “this is false”) commonly used in misinformation interventions such as fact-checking. We coded each effect size that involved any indication of a statement’s factual status during the exposure phase. These cues could be embedded directly in the statements (e.g., “most people know that,” “certain,” “impossible”), provided as feedback after exposure, or presented as explicit tags labeling statements as true or false. We classified veracity cues into four subgroups: positive cues ($n = 32$), negative cues ($n = 41$), balanced cues ($n = 6$), or uncued ($n = 287$).”
- Lines (757-758): “In case where the statement could not be true or false (e.g., opinion-based statements), the effect size was recoded as “none” ($u = 12$).”

See in the Results section:

- Lines (228-233): “**Item type.** The illusory truth effect was significantly larger in studies using statements ($g = 0.41$, 95% CI [0.33, 0.48]) compared to news headlines ($g = 0.17$, 95% CI [0.01, 0.34]), $b = 0.25$, 95% CI [0.02, 0.47], $t(162) = 2.16$, $p = .033$, but not compared to opinion-based claims ($g = 0.47$, 95% CI [0.20, 0.73], $t(117) < 1$. Effect sizes did not significantly differ between news headlines and opinion-based claims, $t(127) = 1.10$, $p = .275$.”
- Lines (252-262): “**Veracity cues.** Veracity cues significantly moderated the illusory truth effect. Explicitly labelling statements as false at the exposure phase ($g = -0.18$, 95% CI [-0.33, -0.03]) decreased the effect compared to no veracity indication ($g = 0.43$, 95% CI [0.37, 0.50]), $b = -0.63$, 95% CI [-0.81, -0.15], $t(164) = 6.87$, $p < .001$, a balanced mix of true or false cues ($g = 0.28$, 95% CI [-0.08, 0.64]), $b = -0.55$, 95% CI [-0.97, -0.12], $t(298) = 2.55$, $p = .011$, and explicitly labelling statements as true ($g = 0.70$, 95% CI [0.51, 0.88]), $b = -0.90$, 95% CI [-1.10, -0.70], $t(170) = 8.79$, $p < .001$). Additionally, explicitly labelling statements as true yielded a larger illusory truth effect compared to no veracity indication, $b = 0.27$, 95% CI [0.03, 0.50], $t(116) = 2.25$, $p = .027$. However, this was not robust with sensitivity analysis using maximum estimates of missing values, $t(117) = 1.25$, $p = .215$. No other moderating effects were found.”
- Lines (300-307): “**Statements’ veracity.** The size of the illusory truth effect was not modulated the factual truth of the exposed statements. Repetition of true statements (g

= 0.46, 95% CI [0.36, 0.56]) yielded similar effect sizes compared to false statements ($g = 0.33$, 95% CI [0.23, 0.42]), $t(66) < 1$, or non-verifiable statements (e.g., opinions, meaningless statements) ($g = 0.31$, 95% CI [0.12, 0.49]), $t(47) = 1.37$, $p = .175$.”

See in the Discussion section:

- Lines (397-412): “Crucially, our findings suggest that explicit cues about the truth status of a statement significantly reduced the illusory truth effect. Statements labeled as false yielded substantially smaller effects compared to uncued ones. This finding provides additional support for the value of debunking interventions. Explicit markers of whether information is true or false may significantly reduce the illusory truth effect. While findings from primary studies are mixed, our results align with some studies showing that veracity cues diminish the illusory truth effect^{56,57}. Others have reported that such cues attenuate but do not eliminate the effect^{6,58}, and some found no influence at all^{59,60}. Stanley et al. noted⁶⁰ that even when participants remember the cue, they may not use it during the truth judgment phase, suggesting that individuals rely more on fluency than retrieved knowledge. However, variability in the type of veracity cue (e.g., warning labels, post-rating feedback, or epistemic qualifiers) likely contributes to the inconsistent findings, especially given the substantial between-study heterogeneity observed in our analysis. Understanding these mechanisms is crucial for developing effective interventions to mitigate the spread of misinformation⁶¹. Future studies should investigate how different types of veracity cues influence the illusory truth effect.”
- Lines (413-426): “We found no significant moderating effect of the factual veracity of statements on the illusory truth effect, which aligns with findings from most primary studies^{6,62}. The illusory truth effect also appeared robust across different types of repeated items (e.g., statements, headlines, or opinions), their factual veracity, and independently of the proportion of true or false statements in the stimuli pool, confirming its relevance to the current misinformation landscape. Interestingly, studies using news headlines yielded smaller effect sizes than those using standard trivia statements. One possible explanation is that many headline-based studies assessed the illusory truth effect in the context of social media misinformation, where items were not systematically chosen for ambiguity. Moreover, a recent meta-analysis⁶³ showed that participants are generally able to distinguish real from fake news, suggesting that false information in headline formats may be easier to identify than ambiguous trivia items.

As such, news headlines may function as an implicit warning cue, attenuating the illusory truth effect in these contexts.”

6. **Comment:** In addition, how were “common knowledge facts” defined in making these exclusions?

Response: Our objective was to focus on genuinely ambiguous statements, which allow us to isolate and measure the illusory truth effect without the confounding influence of firmly established factual knowledge. For this reason, we excluded effect sizes using non-ambiguous statements (e.g., widely known facts, highly implausible claims, procedures involving cues or feedback regarding the actual truth status) if the study’s explicit primary goal was to investigate the effect of these types of statements on reducing the illusory truth effect. The term “common knowledge facts” was referring to statements which may be used in studies with the goal of measuring the illusory truth effect on unambiguously true statements. However, as noted in the previous comment, references to common knowledge facts have been removed from the revised manuscript.

7. **Comment:** Was a full study excluded if participants judged a mix of “common knowledge facts” and ambiguous information?

Response: Our approach was to retain data only from the ambiguous items whenever possible. Specifically, if a study included both ambiguous and non-ambiguous statements but reported them separately (or if the authors’ raw data allowed us to separate them ourselves), we coded only the ambiguous subset. However, if the study provided only a single, combined measure (i.e., mixing ambiguous and non-ambiguous items) and we could not isolate the ambiguous data, we excluded the study entirely. This principle applied across all our exclusion criteria: we always aimed to code valid effect sizes from eligible subsets but excluded studies in which we could not separate the relevant data. We have revised the manuscript to clarify how we addressed situations in which a part of an eligible study (but not all the study) met an exclusion criteria as follows:

- Lines (605-611): “When a study included multiple conditions such that only some conditions met our inclusion criteria, we extracted data exclusively from those qualifying conditions. Specifically, if a study provided distinct outcome metrics for each condition or made raw data available, we retained only the eligible subsets that satisfied

our criteria and excluded the ineligible ones. In instances where outcomes for eligible and ineligible conditions were reported only as combined results, making it impossible to isolate appropriate data, we excluded the entire study to maintain methodological consistency.”

- 8. Comment:** In addition, studies may not have norming data collected in the sample the study was conducted in, making the determination of whether or not they include claims that are “common knowledge facts” in that population impossible to assess.

Response: We agree that, in the absence of norming data for the specific sample, it can be difficult to identify “common knowledge facts.” However, our exclusion of effect sizes did not depend on the presence of such norming data. Instead, we relied on the authors’ own designations. If the authors explicitly labeled certain items as “known facts” or “highly implausible statements,” we previously excluded those items according to our criteria. Conversely, when the authors referred to their items simply as “general” or “trivia” statements without specifying any pretesting or norming, we included those data in our meta-analysis. However, we now include all types of statements, as specified in the response to comment 5.

- 9. Comment:** Finally, it is unclear why only ambiguous information is of interest, given the explicit goal of discussing practical implications regarding the current misinformation crises. Does repeating unambiguously false information nevertheless increase acceptance?

Response: Initially, we focused on ambiguous statements to examine how repetition alone shapes subjective truth in the absence of firm knowledge, yielding a “pure” measure of the illusory truth effect. However, we acknowledge that examining unambiguously false information is directly relevant to addressing real-world misinformation concerns. With our broadened inclusion criteria, we now incorporate studies presenting unambiguous information. In the revised manuscript, we now include the “veracity cue” moderator which allows us to investigate whether repetition increases truth judgment in unambiguously information (cf. response to comment 5), and discuss the implications of our findings as follows:

- Lines (397-412): “Crucially, our findings suggest that explicit cues about the truth status of a statement significantly reduced the illusory truth effect. Statements labeled as false yielded substantially smaller effects compared to uncued ones. This finding provides additional support for the value of debunking interventions. Explicit markers of whether

information is true or false may significantly reduce the illusory truth effect. While findings from primary studies are mixed, our results align with some studies showing that veracity cues diminish the illusory truth effect^{56,57}. Others have reported that such cues attenuate but do not eliminate the effect^{6,58}, and some found no influence at all^{59,60}. Stanley et al. noted⁶⁰ that even when participants remember the cue, they may not apply it during the truth judgment phase, suggesting that individuals rely more on fluency than retrieved knowledge. However, variability in the type of veracity cue (e.g., warning labels, post-rating feedback, or epistemic qualifiers) likely contributes to the inconsistent findings, especially given the substantial between-study heterogeneity observed in our analysis. Understanding these mechanisms is crucial for developing effective interventions to mitigate the spread of misinformation⁶¹. Future studies should investigate how different types of veracity cues influence the illusory truth effect.”

- 10. Comment:** LL161-171 - Can you clarify whether all data from a study were excluded if some (but not all) conditions met the exclusion criteria or if only the condition that met the exclusion criteria was excluded while the others were maintained? It currently sounds as if all conditions were excluded which suggests a loss of relevant data

Response: When studies included multiple conditions, only some of which met our exclusion criteria (e.g., combining non-ambiguous and ambiguous statements in a within-subject design), we excluded just those ineligible conditions while retaining any that satisfied our criteria. If a study reported separate results for each condition (e.g., means, standard deviations, sample sizes) or provided raw data that let us isolate eligible conditions, we used only those eligible subsets. However, if a study presented only aggregate data that combined eligible and ineligible subsets with no separate data, we had to exclude the entire study to preserve methodological consistency. We followed the same procedure when a study reported separate effect sizes for different conditions (e.g., true vs. false statements). Specifically, we extracted each effect size that met our inclusion criteria. Because multiple effect sizes from the same sample introduce dependency, we employed a multi-level meta-analysis to correctly handle clustered effect sizes from a single study sample. We have revised the manuscript to clarify this point as follows:

- Lines (605-611): “When a study included multiple conditions such that only some conditions met our inclusion criteria, we extracted data exclusively from those qualifying conditions. Specifically, if a study provided distinct outcome metrics for each condition or made raw data available, we retained only the eligible subsets that satisfied

our criteria and excluded the ineligible ones. In instances where outcomes for eligible and ineligible conditions were reported only as combined results, making it impossible to isolate appropriate data, we excluded the entire study to maintain methodological consistency.”

- 11. Comment:** LL256-262 - The impact of warnings depends on whether the warning occurs at exposure or test. For example, Jalbert et al., (2020) found that warnings only moderate the effect of repetition when they occur at exposure and have no effect when presented only at test (probably because asking for a truth judgment itself acts as a type of warning). There are important theoretical reasons to suspect that the timing of this warning is critical and that warnings at exposure and test should be analyzed separately rather than pooled into the same category. The ms also flags that many earlier truth effect papers do not contain sufficient information to determine whether or not a study included a warning as part of the standard instructions. Assuming that participants did not receive a warning when the paper doesn't include information regarding these instructions may miscategorize the studies. It would be good to add an analysis that only includes studies with vs. without warnings at exposure to assess the effect of warnings.

Response: We agree with the Reviewers that the timing of the warning, whether at exposure or test, can significantly affect how it moderates the illusory truth effect. For instance, Jalbert et al. (2020) demonstrated that warnings at exposure effectively reduce illusory truth, whereas warnings introduced only at test appear to have little influence. Initially, we primarily focused on warnings delivered at the exposure phase as many studies did not explicitly state whether there was a warning or not (we supposed that any warnings would be specified in the procedure), and sometimes did not clearly differentiate whether they reiterated the warning at test, making it difficult to cleanly separate “exposure-only” vs. “test-only” warnings. However, we acknowledge that this procedure presents the risk to miscategorize studies. Therefore, we:

1. Separated the coding of the warning during the exposure and the test phase to explore the differences in timing.
2. Conducted an additional sensitivity analysis running a meta-regression comparing studies with no warning vs. warning vs. unspecified warning status to investigate any differences in results.

3. Instead of recoding the absence of information about warning as a study containing no warning, we left those as missing data which is addressed by multiple imputations by chained equations (MICE) to preserve relationship between moderators and the outcome without data losses, while preventing any study to be miscategorized.

We now specify this new coding scheme in our revised manuscript as follows:

See in Methods:

- Lines (667-675): “**Truth warning at the exposure phase.** This moderator reflected whether explicitly alerting participants to potential falsehoods prior to exposure modulates the illusory truth effect. We coded whether participants were explicitly instructed that “some” or “half” of the statements were true or false in the exposure phase, including the presence of cues ($u = 163$), or did not provide any explicit warning ($u = 41$). To prevent any risk of miscategorization, we coded studies that did not specify whether participants were warned about the statement’s veracity or did not provide the full verbatim of the instructions to participants as missing data ($u = 162$). Instructions to rate the truth of statements were not coded as explicit warnings.”
- Lines (676-682): “**Truth warning at the test phase.** As warnings at test and exposure are likely to influence truth judgments through distinct cognitive mechanisms and serve different theoretical functions¹⁸, we coded truth warnings at the exposure and test phases as separate moderators. We followed the same procedure as for the warning during the exposure phase, coding studies in which participants were warned ($u = 66$) or not ($u = 41$) that statements could be false in the test phase. Similarly, we coded missing data ($u = 259$) if the warning status is not specified (or respecified for the test phase).”

See in Results:

- Lines (246-251): “Providing a truth warning did not moderate the illusory truth effect at either phase of the experiment. During the exposure phase, the effect did not differ significantly with a truth warning ($g = 0.34$, 95% CI [0.25, 0.43]) or without one ($g = 0.44$, 95% CI [0.32, 0.57]), after controlling for other moderating variables, $t(61) < 1$. Similarly, the illusory truth effect with a warning ($g = 0.38$, 95% CI [0.27, 0.48]) did not differ compared to no warning ($g = 0.37$, 95% CI [0.26, 0.49]) in the test phase, $t(30) < 1$.”

In the additional analysis the Reviewers suggested on the updated data set, we found no significant difference in the effect size between any warning (e.g., “some” of the statements are false, “half” of the statements are false, or the presence of cues) and no warning whether at the exposure phase or test phase ($ps > .12$). Similarly, effect sizes did not differ comparing studies with unspecified warning (e.g., which were previously coded as “no warning”) and studies clearly stating that they did not warn participants, whether at the exposure ($t(236) < 1$) or the test phase ($t(330) < 1$). However, using our new coding scheme (i.e., leaving unspecified warnings as missing data), we surprisingly found no difference between studies with or without warning (neither at the exposure phase or test phase), which is contrary to both our previous analysis and Jalbert et al.’s (2020) findings. We have thus conducted an additional analysis using the new data set with our previous coding scheme (i.e., recoding unspecified warning as no warning) to examine if the incongruity in our results emerges from modifications in our updated data set (e.g., addition of new studies) and analytical model (e.g., new moderators, and bias-correction) or in our coding scheme. This analysis is consistent with our main analysis, which found no effect of warning during the exposure phase, $t(195) < 1$, or the test phase, $t(263) = 1.62$, $p = .106$, suggesting that the lack of effect was robust to the modification of our coding scheme. Thus, we have revised the interpretation of this finding in the Discussion section as follows:

- Lines (359-367): “Asking participants to make an initial truth judgment may serve as an implicit warning that not all statements are true¹⁸. However, our meta-regression found no overall effect of explicit warnings, whether provided during the exposure or test phase. This contrasts with Jalbert et al.’s findings¹⁸, which showed that explicit warnings reduced the illusory truth effect, but only when given during exposure alongside reading instructions. A plausible explanation is that implicit and explicit warnings may function overlap: once participants are instructed to evaluate veracity, an additional instructional warning may provide little new information, which makes its impact difficult to detect at the aggregate level.”

12. Comment: LL229 - The size of the illusory truth effect depends on whether a within-ss or between-ss design is used. The effect is larger when participants judge a mix of repeated and new claims than when they judge only repeated claims (Dechêne et al., 2009 and related studies). It is therefore surprising that this meta-analysis apparently includes within-ss and

between-ss studies without consistently differentiating between them. Within vs. between designs should be treated as an important additional moderating variable.

Response: We agree that systematically differentiating between within- and between-participants designs is crucial, particularly in light of findings indicating the illusory truth effect can vary by design (e.g., Dechêne et al., 2009). Thus, we included the repetition design as a moderator analysis in our revised manuscript. However, it is worth noting that only 5 out of our 366 effect sizes use a between-participants design, limiting our statistical power to detect meaningful differences. Nevertheless, we highlighted this limitation and discussed implications for interpreting the results given the small number of between-participants studies. We have revised the manuscript accordingly as follows:

- Lines (234-236): “The illusory truth effect did not significantly differ whether repetition was manipulated within-participants ($g = 0.38$, 95% CI [0.30, 0.45]) or between participants ($g = 0.17$, 95% CI [-0.28, 0.61]), $t(262) < 1$.”
- Lines (426-432): “Contrary to earlier primary studies suggesting that the illusory truth effect is smaller in between-participant designs^{64–66}, our meta-regression did not detect a significant difference. However, this result should be interpreted considering the imbalance between the number of effect sizes drawn from within-participants design ($u = 361$) and between-participants design ($u = 5$). This disparity substantially limits statistical power, suggesting that any true difference in effect sizes may simply be too small to detect considering the current data set.”
- Lines (647-652): “**Repetition study design.** To evaluate whether the illusory truth effect varies based on repetition format, we examined whether participants were exposed to repeated and new statements within the same condition or in isolation. We coded whether the effect size was extracted from a within-participant ($u = 361$) or a between-participant ($u = 5$) manipulation of the repetition. In between-participant designs, two groups of participants evaluated either all-repeated or all-new statements during the test phase.”

13. Comment: LL301-302 - Why was the response scale coded by odd vs. even? Explaining the reason would be useful.

Response: We distinguished odd-numbered response scales (which have a “neutral” option) from even-numbered scales (which do not) because having or lacking a midpoint could impact

how participants make their truth judgments. An odd-numbered scale allows individuals to select a neutral middle point, whereas an even-numbered scale forces participants to choose one side of the scale or the other. We therefore coded scales as odd or even to capture this potentially meaningful difference in how participants use the rating scale. We have included this justification in the manuscript as follows:

- Lines (741-745): “We distinguished response scales based on whether they had an odd number of points ($n = 119$) or an even number of points ($n = 241$). The goal was to explore the moderating role of a “neutral” or midpoint option (e.g., 3 on a 5-point scale), versus even-numbered scales that force participants to express a choice that leans in one direction or another (i.e., there is no true midpoint).”

- 14. Comment:** LL304-305 - Why was data containing a mix of true and false claims excluded from the veracity comparison? This represents a large portion of the data and comparing true vs. false vs. mixed could provide useful insights.

Response: We did not exclude data containing a mix of true and false statements from the analyses. Instead, whenever a study did not clearly distinguish between which items were true or false (or did not provide raw data for us to differentiate them), we coded veracity as “missing” and used multiple imputation via MICE to address these missing values. This approach preserves the relationships among variables and mitigates data loss.

Our comparison specifically aimed to see whether the illusory truth effect differs for true versus false statements. Including studies where items are “mixed” (i.e., contain both true and false statements) makes it impossible to isolate a single effect size for true items and another for false items. With our current methodology, we have the advantage of extracting two effect sizes (one for true statements and one for false statements) from a single study using mixed veracity items. Conversely, if we separated all-true, all-false, and mixed designs, we could not identify how much of the overall truth effect arises from each veracity type. Because our analysis requires two clear-cut effect sizes (i.e., one for true statements and one for false statements), mixed designs prevent that distinction and thus cannot serve the purpose of a direct true–false comparison.

Nevertheless, we agree that comparing illusory truth effects across purely true, purely false, and mixed statements could offer valuable insights. To capture this variance, we introduced a new moderator, “proportion of true statements,” which codes the proportion of statements in each study that were factually true. This would provide a comparison of designs using all-true vs.

all-false vs. mixed items, while maintaining a direct comparison of the illusory truth effect for factually true and false statements. We have modified the manuscript accordingly as follows:

- Lines (304-307): “Similarly, the proportion of true or false statements showed no influence on the effect size ($t(245) < 1$), indicating that the illusory truth effect is comparable in studies that employ mixed, all-true, or all-false item pools.”
- Lines (414-418): “The illusory truth effect also appeared robust across different types of repeated items (e.g., statements, headlines, or opinions), their factual veracity, and independently of the proportion of true or false statements in the stimuli pool, confirming its relevance to the current misinformation landscape.”
- Lines (633-637): “When a study included effect sizes potentially matching multiple categories of a moderator, we extracted each effect size individually whenever the data (e.g., distinct outcomes or open datasets) permitted clear separation. In instances where effect sizes were reported in a combined form and could not be disentangled by moderator category, we coded the moderator as missing.”
- Lines (746-753): “**Proportion of true statements.** We included the proportion of true statements to evaluate whether the overall truth base rate within a study influences participants’ sensitivity to repetition. This moderator captures variation in the informational context (e.g., all-true, all-false, or mixed item sets), which may affect the baseline likelihood of endorsing repeated items as true. To capture design variability across all-true, all-false and mixed item sets, we created a continuous moderator (range 0-1) that records the proportion of factually true statements in each study’s stimulus pool (e.g., 0 = all false; 0.50 = 50 % true/50 % false; 1 = all true).”
- Lines (772-774): “This method allowed us to preserve the relationships between variables while avoiding data losses associated with deletion methods.”

To clarify the usefulness of the three-level meta-regression model, we also have added an additional explanation:

- Lines (846-848): “This method appropriately handles the clustering of effect sizes within individual studies and allows for estimation of between-study and within-study variance.”

15. **Comment:** LL379 - We may have overlooked this, but we didn't see whether or not effect sizes were fixed across subgroups? We would expect them to be fixed given recommendations by Borenstein et al. (2009) when the number of subgroups is small.

Response: We thank the Reviewers for raising this point. Borenstein et al. (2009) in Chapter 19 recommended pooling the between-study variance (τ^2) across subgroups when each subgroup contains only a few studies ($\approx \leq 5$), because separate τ^2 estimates would be unstable. In our three-level random-effects model (rma.mv, metafor package), the level-3 between-study variance (τ^2) and the level-2 within-study variance (σ^2) were estimated once from the complete dataset and held constant for every moderator and subgroup analysis, which follows the approach recommended by Borenstein et al. (2009). We clarified this point in the revised manuscript as follows :

- Lines (857-860): "In the meta-regression analysis, the between-study (τ^2) and within-study (σ^2) variance components were estimated from the complete set of effect sizes and kept constant across moderator levels⁸³."

Results

16. **Comment:** LL450 following. – This paragraph is uninformative. We cannot meaningfully relate this material to specific variables, even with the help of the supplementary material.

Response: We appreciate the request for greater clarity. The revised manuscript now describes more clearly our analyses and provides an interpretation of our findings as follows:

- Lines (195-205): "Using the RoB-2 framework, 43 of the 182 studies were judged at low overall risk, while the remaining 139 raised some concerns. Among the five RoB-2 domains, deviation from intended intervention (Domain 2) raised concerns in 86 studies and selection of the reported results (Domain 5) in 138 studies, primarily resulting from lack of pre-registration processes. Supplementary Table 1 describes every study by the five RoB-2 domains. Figure 2 provides a weighted summary. A three-level meta-regression using overall RoB grade as a moderator showed no significant difference between effect sizes from studies at low risk of bias and those with some concerns, $b = 0.12$, 95% CI [-0.02, 0.26], $t(363) = 1.63$, $p = .104$. Thus, the risk-of-bias analyses suggested that the pooled illusory truth effect remained stable after controlling for study-level biases."

17. Comment: All p-values should be reported to the same level of specificity (some have 2 decimal places and some have three).

Response: We thank the Reviewers for their suggestion. We have now standardized the precision of all p-values throughout the manuscript to 3 decimals.

18. Comment: It would also be useful to provide degrees of freedom and corresponding estimates of ds for each subgroup in the main text (or the difference between ds)

Response: We thank the Reviewers for this helpful suggestion. We now report the degrees of freedom and the estimates of Hedge's g and its corresponding 95% confidence intervals for every subgroup comparison in the main text of the revised manuscript as follows:

- Lines (216-221): "To provide an overview of effect sizes within each subgroup, we also report the estimates. Because these values are not adjusted for the other moderators, they are not directly comparable to the coefficients from the multivariate meta-regression."

Please see for an instance of the revision implemented in the Results section as follows:

- Lines (237-245): "A significant moderating effect was found for the content of the task used during the exposure phase. When participants were asked for a truth judgment during the exposure phase ($g = 0.10$, 95% CI [-0.03, 0.23]), the illusory truth effect was consistently reduced compared to irrelevant tasks ($g = 0.42$, 95% CI [0.34, 0.50]), $b = -0.35$, 95% CI [-0.50, -0.20], $t(303) = 4.62$, $p < .001$, or passive processing ($g = 0.47$, 95% CI [0.34, 0.60]), $b = -0.47$, 95% CI [-0.70, -0.23], $t(221) = 3.94$, $p < .001$. However, the illusory truth effect did not significantly vary when the exposure task only instructed participants to read the statements compared to performing an irrelevant task, $t(218) = 1.32$, $p = .190$.

Discussion

19. Comment: LL550-551 - Jalbert et al. (2020) discuss ways in which making an initial truth evaluation at exposure (similar to a warning) may decrease the size of the illusory truth effect goes beyond the depth of processing account offered here. Asking people for a truth judgment necessarily entails that some of the claims may not be true.

Response: We agree that asking participants for an initial truth judgment can operate in ways that go beyond a simple depth-of-processing account. As the Reviewers noted, requesting a truth evaluation at exposure could implicitly warn participants that some statements may be false. In other words, truth judgments during the exposure phase might function in part like a “warning” by prompting participants to treat the task as one involving potential falsehoods rather than simply rating interestingness or familiarity. However, we did not find an effect of warning neither during the exposure phase or test phase in our updated analysis. We have thus revised the Discussion section accordingly as follows:

- Lines (359-367): “Asking participants to make an initial truth judgment may serve as an implicit warning that not all statements are true ¹⁸. However, our meta-regression found no overall effect of explicit warnings, whether provided during the exposure or test phase. This contrasts with Jalbert et al.’s findings ¹⁸, which showed that explicit warnings reduced the illusory truth effect, but only when given during exposure alongside reading instructions. A plausible explanation is that implicit and explicit warnings may function overlap: once participants are instructed to evaluate veracity, an additional instructional warning may provide only few additional information, which makes its impact difficult to detect at the aggregate level.

20. Comment: LL590-597 - The studies included in this paper do not include multiple truth judgments (i.e. an initial one and a later one) but the explanation provided is specific to this scenario. What about when participants do not complete multiple test phases?

Response: In our meta-regression, effect sizes emerging from multiple truth judgment (i.e., an initial one and a later one) was included in the subgroup “truth judgment” of the “exposure task” moderator. In this subgroup, participants were asked to rate the truth of statements in the exposure phase, which half were generally repeated in the test phase. However, there might be a confusion from our exclusion criteria (currently criteria #2), which focused on eliminating studies in which the same items were repeated more than once before the final truth judgment, including multiple test phases that were used to re-test the same set of items more than once. The objective of this exclusion criteria was to control the number of times participants were exposed to a statement, focusing on the illusory truth effect emerging from only one exposure. We did, however, include studies in which participants made an initial truth judgment with a single test phase, or multiple test phases in which repeated statements were only presented once. For instance, while Nadarevic et al.’s (2014) first experiment implemented an exposure phase

(with an initial truth judgment) followed by two test phases (one after 10 minutes and another after one week). Participants did not re-judge the same items in both test phases (i.e., half of the initially exposed statements were repeated in the first test phase, and the other half were judged only in the second test phase). Therefore, each item was effectively repeated just once, which satisfied our inclusion criteria; accordingly, we included such studies and coded their effect sizes separately. In the paragraph the Reviewers referenced, we highlighted that Nadarevic et al. (2014) found an initial truth judgment appeared to suppress the illusory truth effect when the retention interval was short (10 minutes), but that the effect reemerged with a one-week retention interval, leading us to conclude that “over time, the influence of initial truth judgment may decrease, allowing fluency processes to take over”. However, to strengthen the validity of this statement, we now also refer to Brashier et al. (2020)’s study that reports similar findings in four different experiments. The manuscript was revised as follows:

- Lines (367-373): “Importantly, evidence indicates that an initial truth judgment decreases illusory truth effect only when the delay between the exposure and the test phase is short. After a delay of two days, the effect of the initial truth judgment disappeared^{23,51}. This effect was robust after controlling for participants’ memory of the statements, suggesting that the initial truth judgment may engage different cognitive processes that override fluency or coherence effect. Over time, the initial truth judgment may have a smaller impact, allowing fluency processes to take over.”

Furthermore, we have clarified our exclusion criteria 2 as follows:

- Lines (543-545): “(2) multiple exposure to a statement before evaluation (e.g., participants are exposed to a same statement twice or more before test phase)”

- 21. Comment:** LL639-653 – When the fluency experience is (correctly or incorrectly) attributed to an incidental variable (e.g., repetition), the informational value of fluency is undermined. Awareness of the influence is more likely after very short delays (for a discussion, see Schwarz, Jalbert, et al., 2021).

Response: We appreciate the Reviewers drawing attention to the role of attributions in shaping the impact of fluency on truth judgments. With our revised analysis plan and expanded dataset, we no longer observe a clear effect of delay on the effect size. However, we have integrated this explanation in the Discussion section as follows:

- Lines (383-396): “We also observed a relatively smaller truth effect when the test phase followed exposure within the same day versus within the same week, although sensitivity analyses were inconsistent. Primary studies have also reported inconsistent findings. Some suggest that a longer delay was associated with a larger illusory truth effect ¹⁹, possibly because individuals attribute their ease of processing to recent repetition and thus discount fluency as a cue for truth. In contrast, with a longer delay, the source of fluency may be forgotten, which makes fluency more likely to be misattributed to truthfulness ³¹. Other studies found that a smaller effect was associated with a longer delay ^{53,54}, which is consistent with both the fluency account and the referential theory, which predict that fluency or coherence in memory would be reduced over time. However, the moderating effect of delay may interact with other variables influencing the illusory truth effect during the exposure phase, such as the type of task ¹⁹ or the type of repetition ⁵⁵. Given the critical role of the exposure phase, the interaction between delay and exposure conditions may provide key insights into the underlying mechanisms of the illusory truth effect.”

22. Comment: LL692-695 - Several recent papers report that debunking messages that include a repetition of false information do not always result in backfire effects (e.g., Ecker et al., 2020, Briony Swire-Thompson et al., 2020).

Response: We thank the Reviewers for drawing our attention to this literature. We fully agree that the interplay between re-exposure to misinformation and backfire effects seems more nuanced than initially stated. We have revised the Discussion of the illusory truth effect in the context of misinformation to provide a more nuanced account of the backfire effects as follows:

- Lines (510-514): “Metacognitive interventions ⁶⁹, encouraging reflection with accuracy prompts ⁷⁰ or debunking strategies ⁷¹ may effectively counteract the illusory truth effect. Moreover, the concern that repeating false claims during debunking might backfire appears less warranted than previously feared; current evidence suggests that corrective interventions can be effective without amplifying beliefs in the exposed claims ^{72,73}.”
-

Reviewer 3 Comments

Overall Comment: This is an extremely relevant article for those who approach the field of truth illusions. As a participant in the field, I appreciate the updated meta-analytic review work relative to the 2010 work by Dechene and colleagues, and the systematic map created by Henderson et al. in 2022. However, despite claiming its relevance, I believe the article has room for improvement, and I would like to see these revisions made before recommending its publication.

Response: We thank the Reviewer for his/her comment and suggestions for improving our manuscript.

- 1. Comment:** My first point refers to the objectives of conducting this meta-analysis (p. 6). These should focus more on the gains it brings rather than on the deficits that previous approaches might present. While gains and deficits are related, I would recommend placing greater emphasis on what is gained. The argument that the advantage comes from greater rigor in estimates, enabled by the developments in the meta-analytic approach, is weak on its own. The studies are now more numerous, with greater variability in conditions, allowing for more consistent estimation of moderators' roles and the inclusion of new moderators that were previously unexplored, among other benefits. These are all relevant arguments in favor of a new meta-analytic study. However, it is precisely in this area where the paper loses some strength—it adds very little to what we already know. There is no reason for this. Since 2010, many other questions have been raised about the effect, and these can be tested in a meta-analytic context. By framing the paper solely around emphasizing the rigor of effect estimation, an opportunity is being missed here.

Response: We agree that clearly articulating the added value of our new meta-analysis beyond methodological rigor is critical. Indeed, framing the objectives mainly around methodological improvements may overshadow other significant contributions made possible by our updated meta-analysis. As the Reviewer pointed out, since the previous meta-analysis by Dechêne et al. (2010), the literature on the illusory truth effect has grown substantially, offering not only an increased quantity of data but also greater methodological diversity and new theoretical questions. We have now revised the introduction and the discussion sections the added value of this new meta-analysis as follows:

- Lines (102-120): “In the present study, we conducted an updated meta-analysis of the illusory truth effect that synthesizes 182 studies and 366 effect sizes published from 1977 to 2025. Our goals were twofold. First, we aimed to provide a more accurate estimate of the overall effect size by using a three-level random-effects model ^{39,40}, which accounts for the non-independence of multiple estimates derived from the same sample. This model was complemented by (a) study-level quality appraisal using the Risk of Bias 2 tool ⁴¹, (b) tests for publication bias, and (c) extensive sensitivity analyses using multiple imputation strategies for missing data. Second, we substantially expanded the set of moderators, including item type, presence of veracity cues or truth warnings, statement’s veracity, and proportion of true items. The inclusion of these moderators enhances the practical relevance of our findings by identifying contextual features of information exposure that amplify or attenuate the illusory truth effect, offering direct implications for the design of interventions aimed at countering misinformation. These moderators were analyzed simultaneously in a multiple meta-regression to isolate the contribution of each moderator independently of the others. While our selection of moderators was primarily guided by pragmatic and applied considerations, some of them also align with key theoretical distinctions in the literature, offering insight into mechanisms such as fluency-based processing, referential coherence and memory-based correction mechanisms assumed to be at the root of the illusory truth effect.”
- Lines (318-329): “This systematic review and three-level meta-analysis provides robust evidence that repetition increases the perceived truth of information, with growing relevance in today’s media landscape. Recent studies suggested that repeated exposure not only increases the perceived accuracy of misinformation but also increases individuals’ tendency to share it ⁴⁸. This highlights the importance of mitigating the illusory truth effect in the context of the current misinformation crisis. Synthesizing 182 studies (366 effect sizes; N = 31,184), we found a small illusory truth ($g = 0.37$, 95% CI [0.30, 0.44]), after correcting for publication bias. Comparing to the previous meta-analysis ³⁸, our analyses incorporated an expanded set of moderators, many of which have direct practical implications. Together, these moderators accounted for 28% of the between-study variance on average, indicating that their relative importance in moderating the illusory truth effect.”

2. **Comment:** In my reading I would have liked to find not only evidence that corroborates or challenges Dechene's conclusions but new evidence. For example, Dechene discusses the difference in effect size when the contrast is within-subjects versus between-subjects, highlighting that the fluency/familiarity experience may be relative. Since 2010, some studies have directly addressed this issue, providing data that could be included in this new meta-analysis. With this approach, the present paper would add secure knowledge to the previous paper adding content and not only more rigor in the estimation methods.

Response: We thank the Reviewer for pointing this out and fully agree that the between-participant or within-participant manipulation of the repetition in study design can modulate the illusory truth effect as described in some studies (i.e., Dechêne et al., 2009, Garcia-Marques et al., 2016; 2019). As noted in our response to Reviewer 1 and 2 (comment 12), we have now added the "repetition study design" moderator in our three-level meta-regression. However, only 5 out of 366 effect sizes use a between-participants design, limiting our statistical power to detect meaningful differences. In the revised manuscript, we now describe the new moderator and note in the discussion that the evidence is not definitive due to lack of power, see our response to Reviewer 1 and 2 for the changes implemented in the revised manuscript.

3. **Comment:** There are also references in the original paper to differences in effect magnitude between repetition and fluency manipulations, and since then, more studies have been published that use both types of manipulations. I do not see familiarity being contrasted with fluency in this analysis.

We thank the Reviewer for raising this issue. In the literature on the illusory truth effect, the subjective sense of familiarity linked with repetition is generally framed as a manifestation of conceptual fluency (e.g., Silva et al., 2016; Unkelbach et al., 2019). Because our goal was to derive a clean estimate of the repetition-induced truth effect, our eligibility criteria restricted the intervention to verbatim or gist repetition and explicitly excluded studies in which fluency was manipulated independently of repetition (e.g., font clarity, perceptual priming). We agree that systematically contrasting repetition with other fluency routes is theoretically important (Silva et al., 2016, 2017; Unkelbach et al., 2019), but this problematic is outside of the focus of the present meta-analysis.

4. **Comment:** Another very relevant aspect that could not have been addressed 14 years ago is the nature of the information. When identifying the effect, the statements made were

ambiguous. Today, many studies not only present factually false information but also add warnings, etc. These data are not addressed here.

Response: We fully agree that the informational context including the nature of information and potential cues to address the issue of misinformation is increasingly relevant. In addition to the factual veracity of the presented information and the presence of warnings in the instructions (during the exposure or the test phase), we now examine the role of the type of item (e.g., statements, news headlines, opinions) and veracity cues (e.g., uncued, positive cues, negative cues, or balanced cues) in moderating the illusory truth effect on our new analysis. For more details on the changes proposed to the manuscript, see our responses to Reviewer 1 and 2's comments (comments 5, 9 and 11).

5. **Comment:** Essentially, what I think diminishes the relevance of this paper is that the moderation analyses are neither directly contrasted with the status quo provided by previous analyses nor clearly theoretically oriented. The analyses would gain relevance if they directly or indirectly answered theoretically significant questions (which would also facilitate and better integrate the data discussion). For instance, more than the instructions and the time spent reading the sentences at the first moment, it is the question of whether these factors may influence the explicit use of memory as information about the validity of a statement. If the effect is smaller when the instructions make it explicit that memory is not informative (with $\frac{1}{2}$ true sentences and $\frac{1}{2}$ false sentences presented at the start) than when nothing is stated, and this happens especially when there is encoding time, such a test could examine whether the effect depends on explicit versus implicit memory. Such kind of questions would induce to test the interaction between moderators and not only their main effects.

Response: We appreciate the reviewer's comment that the value of moderation analyses is increased when they test theoretically significant questions. While our primary rationale for moderator selection was pragmatic (e.g., we aimed to assess which contextual and methodological features reliably amplify or reduce the illusory truth effect) with an eye toward real-world implications (e.g., misinformation correction, truth judgments in digital environments), several of the moderators we included also provide information on theoretical accounts of the effect. However, testing these theoretical implications is limited by our data presenting substantial heterogeneity and uneven distributions of effect sizes across moderator combinations, thereby limiting the power and interpretability of interaction effects.

The Reviewer specifically proposed investigating whether longer encoding time affects the illusory truth effect differently depending on whether participants are explicitly told that memory may not be a reliable source of truth (e.g., when warned that “half” of the statements are false). We agree that such analysis would shed light on the role of memory on the illusory truth effect. In line with this suggestion, we conducted an exploratory analysis including an interaction term between warning and presentation time during exposure in our meta-regression model. However, neither the interaction nor the main effects reached statistical significance.

A key limitation to this analysis is the sparse and imbalanced cell structure, as shown below:

Warning/Time	<= 5 sec	> 5 sec	Participant-paced	Unspecified
Presence of cues	4	8	3	11
“Half” true/false	5	5	2	2
“Some” true/false	33	11	69	1
No warning	4	4	33	2
Unspecified	30	64	72	2

As this table illustrates, many theoretically meaningful interaction cells contain too few effect sizes for reliable estimation. We therefore see the identification and testing of such interactions as an important direction for future experimental research, where moderators can be systematically manipulated under controlled conditions.

Nevertheless, we agree with the Reviewer that contrasting our results with the previous meta-analysis would be valuable. Thus, we have revised the manuscript to clarify how our moderator choices relate to both practical application and theoretical models, and we now include Table 2 to contrast our findings with Dechêne et al.’s to contextualize our findings within prior meta-analytic work as follows:

- Lines (116-120): “While our selection of moderators was primarily guided by pragmatic and applied considerations, some of them also align with key theoretical distinctions in the literature, offering insight into mechanisms such as fluency-based processing, referential coherence and memory-based correction mechanisms assumed to be at the root of the illusory truth effect ²⁸.”
- Lines (219-220): “Table 2 summarizes the subgroup analyses contrasted with Dechêne et al.’s meta-analysis.”
- Lines (624-630): “Our selection of moderators was primarily guided by pragmatic considerations, with the aim of identifying factors that inform how, when, and for whom

the illusory truth effect is most likely to occur. These included features commonly encountered in online misinformation contexts such as headline formats, exposure duration, or the presence of accuracy cues. At the same time, several moderators also align with key theoretical distinctions in the literature, including fluency-based processing, referential coherence, and memory-based correction mechanisms ²⁸.”

We have also included an additional justification for each moderator in the Methods section as follows:

- Lines (639-642): “We included item type (e.g., trivia, headlines, opinions) to examine whether the illusory truth effect generalizes across common real-world formats, especially given the prominence of news headlines and unverifiable claims in online misinformation.”
- Lines (647-649): “To evaluate whether the illusory truth effect varies based on repetition format, we examined whether participants were exposed to repeated and new statements within the same condition or in isolation.”
- Lines (653-655): “We examined the moderating effect of participants’ task during the initial exposure to determine how different levels of cognitive engagement during encoding influence susceptibility to the illusory truth effect.”
- Lines (667-669): “This moderator reflected whether explicitly alerting participants to potential falsehoods prior to exposure modulates the illusory truth effect.”
- Lines (676-679): “As warnings at test and exposure are likely to influence truth judgments through distinct cognitive mechanisms and serve different theoretical functions ¹⁸, we coded truth warnings at the exposure and test phases as separate moderators.”
- Lines (683-685): “We included veracity cues as a moderator to examine the impact of explicit truth indicators (e.g., labels or statements like “this is false”) commonly used in misinformation interventions such as fact-checking.”
- Lines (691-693): “We examined the influence of delay between exposure and test to assess the temporal durability of the illusory truth effect. This moderator helps understanding the long-term influence of repeated information in both experimental and applied settings.”
- Lines (702-704): “This moderator reflected whether exact repetition (verbatim) and paraphrased repetition (gist) differentially influence perceived truth, as these forms commonly occur in real-world contexts.”

- Lines (708-711): “We included the modality of presentation to understand how the presentation channel affects the illusory truth effect. This moderator accounts for variability in how content is encountered across studies and reflects real-world differences in media consumption.”
- Lines (717-721): “This moderator reflected whether the duration of initial exposure affects the illusory truth effect. Exposure time is a key methodological factor that may affect encoding depth and fluency. This moderator also reflects real-world variability in how long individuals attend to repeated information, making it relevant for interpreting the robustness of the effect in ecological settings.”
- Lines (729-732): “To examine whether the amount of time available for truth judgments moderates the illusory truth effect, we examined the presentation time during the test phase. Varying test durations may influence the extent to which participants rely on quick, fluency-based heuristics versus more deliberative evaluation.”
- Lines (738-745): “We coded whether the response scale was a Likert scale ($u = 269$), a binary scale ($u = 91$) (e.g., yes vs. no), or a slider from 0 to 100 ($u = 6$) to examine whether the precision of the response format used to evaluate truth judgments influenced the effect size. Additionally, we distinguished response scales based on whether they had an odd number of points ($u = 119$) or an even number of points ($u = 241$). The goal was to explore the moderating role of a “neutral” or midpoint option (e.g., 3 on a 5-point scale), versus even-numbered scales that force participants to express a choice that leans in one direction or another (i.e., there is no true midpoint).”
- Lines (746-750): “We included the proportion of true statements to evaluate whether the overall truth base rate within a study influences participants’ sensitivity to repetition. This moderator captures variation in the informational context (e.g., all-true, all-false, or mixed item sets), which may affect the baseline likelihood of endorsing repeated items as true.”
- Lines (754-756): “We investigated whether repetition increases perceived truth for true vs. false statements equally which relates to the illusory truth effect in the misinformation context.”
- Lines (761-765): “We examined the effect of testing environment to assess whether the data collection setting (online vs. face-to-face) introduces variability in effect sizes. This is especially relevant considering the increasing use of online platforms in the illusory truth effect research. It also helps evaluate the generalizability of laboratory findings to digital contexts where repeated misinformation is most encountered.”

6. **Comment:** In the same line of thought, I believe that the analysis contrasting the magnitude of effects between online and in-person conditions is presented without contextualizing its relevance (pragmatically to alert that the conditions do not allow the effect to be detected, or theoretically because we know the level of motivation of the respondents is different?). Therefore, it lacks a true discussion of the results.

Response: We appreciate the Reviewer's comment and agree that the difference in effect sizes between online and in-person studies requires deeper contextualization.

The choice of this moderator was clearly pragmatical. With the growing reliance on online platforms for data collection it is important to assess whether study format introduces systematic variability in effect sizes. We included this moderator for its implications for the generalizability of findings to real-world settings (e.g., online misinformation exposure).

In the original manuscript, we reported that in-person studies tended to show larger illusory truth effects compared to online studies. However, following modifications to our analysis (i.e., inclusion of new studies, additional moderators, and a small-study effect corrections), the difference between online and in-person studies did not reach statistical significance. This suggests that the previously observed difference may not be robust (as suggested by the inconsistent findings in the sensitivity analysis) and could reflect variability unrelated to the testing environment. We believe this absence of a significant effect is still worth highlighting, as it suggests that the illusory truth effect may be robust across testing environments. Thus, we have revised our manuscript accordingly as follows:

- Lines (761-765): "We examined the effect of testing environment to assess whether the data collection setting (online vs. face-to-face) introduces variability in effect sizes. This is especially relevant considering the increasing use of online platforms in the illusory truth effect research. It also helps evaluate the generalizability of laboratory findings to digital contexts where repeated misinformation is most encountered."
- Lines (445-451): "The illusory truth effect seems to operate similarly whether participants complete the experiment online or in laboratory settings. This finding suggests that factors such as experimenter presence, environmental control, or potential distractions in online settings do not substantially influence cognitive processes involved in the illusory truth effect. Considering the increasing reliance on online data collection in psychological research, this consistency supports the generalization of such findings in real-world settings"

7. **Comment:** Personally, I find some conclusions drawn from the analysis of studies with very different methodologies concerning. For example, what happens when the authors conclude that the most relevant phase for deep processing is encoding, with retrieval not being relevant. In the retrieval phase, the time of sentence presentation is considered independently of whether they are presented immediately or one month after the study phase. In neither of these contexts does the exposure time influence retrieval conditions. One reason is that the information is still accessible in working memory or almost so, and in the other case, explicit accessibility does not improve with sentence presentation time.

Response: We thank the Reviewer for highlighting this important methodological point. In our primary three-level meta-regression, we analyzed the main effect of each moderator because cells in crossed designs would contain too few effect sizes to support a reliable interaction test. We agree that this approach has limitations, particularly regarding the detection of potential interaction effects (e.g., between delay and test-phase presentation time).

The Reviewer rightfully notes that the presentation time has theoretically a limited impact on retrieval conditions when the test phase follows immediately, or after a too long period the exposure phase. To address this specific concern, we performed an exploratory analysis of simple effects, excluding effect sizes drawn from studies with no delay and with a delay superior to a week. In this restricted data set, longer presentation time during the test phase still did not reliably modulate the illusory-truth effect compared with participant-paced presentation, $t(101) = 1.72, p = .088$, or shorter presentation time, $t(117) = 1.71, p = .070$.

Within the moderator set and power constraints of our meta-analysis, two exposure phase moderators (i.e., exposure task, presentation time) robustly predicted effect size, whereas no test phase moderators reached conventional significance. While these findings highlight the relative salience of the exposure phase in our data, we acknowledge that it cannot rule out possible contribution of retrieval processes in specific study designs. In the revised manuscript, we modified statements on the importance of retrieval and discussed these limitations as follows:

- Lines (476-483): “Another key limitation of the current meta-analysis is the substantial methodological diversity across the included studies. Procedures varied in initial rating tasks, delay intervals, presentation timing, and other moderators. While our three-level meta-regression is appropriate for estimating overall trends and accounting for clustered data, it lacks power to detect interaction effects given the uneven distribution of study

designs across moderator combinations. Thus, moderator effects should be interpreted as indicators of relative salience rather than definitive estimates, as potential interactions may remain undetected in aggregate-level analyses”

- 8. Comment:** I believe that the text would benefit from better readability if the results of the moderation analyses were presented in Figure 4 within a table, and if on that table they were contrasted with previous meta-analyses, highlighting what differs and what is new.

Response: We thank the Reviewer for his/her suggestion. In our revised manuscript, the results of moderation analyses with the regression coefficient and their confidence intervals are reported in the main text. Additionally, the results are now contrasted with the results of Dechêne et al.’s (2010) meta-analysis in Table 2 presented below. A quick visual summary of the moderators’ effect is still provided in Figure 3, see:

- Lines (219-221): “Table 2 summarizes the subgroup analyses contrasted with Dechêne et al.’s meta-analysis.”

Table 2. Regression analysis results contrasted with Dechêne et al. (2010). All effect sizes and 95% confidence intervals are from bivariate random-effects subgroup analyses based on the between-item criterion. The current results reflect bias-corrected estimates using the PEESE method and account for missing data via multivariate imputation by chained equations, in contrast to the listwise deletion approach used by Dechêne et al. (2010).

	Current moderator analysis					Dechêne et al. (2010)				
	Coding	u	g	95% CI	P -value	Coding	k	d	95% CI	P -value
Item type	Statement	309	0.41	[0.33, 0.48]	< .001	/	/	/	/	/
	Headline	43	0.17	[0.01, 0.34]	.048	/	/	/	/	/
	Claim	12	0.47	[0.20, 0.73]	< .001	/	/	/	/	/
Repetition manipulation	Within-participants	361	0.38	[0.30, 0.45]	< .001	Heterogeneous	65	0.51	[0.44, 0.58]	< .001
	Between-participants	5	0.17	[-0.28, 0.61]	.464	Homogeneous	1	-0.09	[-0.76, 0.55)	n.s.
Exposure task	Truth judgment	78	0.10	[-0.03, 0.23]	.137	High level of processing	51	0.45	[0.37, 0.54]	< .001
	Irrelevant task	213	0.42	[0.34, 0.50]	< .001	Low level of processing	19	0.62	[0.49, 0.75]	< .001
	Passive processing	73	0.47	[0.34, 0.60]	< .001					
Truth warning at exposure	Warning	163	0.34	[0.25, 0.43]	< .001	/	/	/	/	/
	No warning	41	0.44	[0.32, 0.57]	< .001	/	/	/	/	/

Truth warning at test	Warning	66	0.38	[0.27, 0.48]	< .001	/	/	/	/	/
	No warning	41	0.37	[0.26, 0.49]	< .001	/	/	/	/	/
Veracity cues	No cue	287	0.43	[0.37, 0.50]	< .001	/	/	/	/	/
	Balanced cues	6	0.28	[-0.08, 0.64]	.130	/	/	/	/	/
	Positive cue	32	0.70	[0.51, 0.88]	< .001	/	/	/	/	/
	Negative cue	41	-0.18	[-0.33, -0.03]	.020	/	/	/	/	/
Delay between the exposure and test phases	No delay	135	0.46	[0.35, 0.56]	< .001	/	/	/	/	/
	Within day	138	0.33	[0.23, 0.43]	< .001	Within day	25	0.49	[0.37, 0.62]	< .001
	Within week	47	0.36	[0.21, 0.50]	< .001	Within week	14	0.44	[0.29, 0.59]	< .001
	Over a week	41	0.25	[0.06, 0.44]	.008	Longer delay	12	0.49	[0.32, 0.65]	< .001
Repetition type	Verbatim	350	0.36	[0.29, 0.44]	< .001	Verbatim	64	0.53	[0.46, 0.60]	< .001
	Gist	16	0.63	[0.35, 0.91]	< .001	Gist	6	0.22	[-0.02, 0.46]	< .10
Presentation mode	Visual	257	0.38	[0.31, 0.46]	< .001	Visual	38	0.52	[0.42, 0.61]	< .001
	Verbal	62	0.30	[0.12, 0.48]	< .001	Auditory	24	0.52	[0.39, 0.64]	< .001
	Both	17	0.32	[0.02, 0.62]	.071	Mixed	4	0.67	[0.38, 0.95]	< .001
	Participant-paced	179	0.34	[0.26, 0.43]	< .001	Participant-paced	16	0.41	[0.24, 0.58]	< .001
	5 seconds or less	77	0.39	[0.26, 0.52]	< .001					

Presentation time at exposure*	More than 5 seconds	92	0.48	[0.33, 0.63]	< .001	8 seconds or less	5	0.51	[0.21, 0.82]	< .01
Presentation time at test*	Participant-paced	265	0.38	[0.31, 0.46]	< .001	More than 8 seconds	27	0.53	[0.41, 0.65]	< .001
	5 seconds or less	43	0.31	[0.12, 0.49]	< .001					
	More than 5 seconds	54	0.36	[0.16, 0.56]	< .001					
Response scale (1)	/	/	/	/	/	1-7 (false-true)	40	0.41	[0.32, 0.49]	< .001
	/	/	/	/	/	1-7 (true-false)	12	0.65	[0.49, 0.79]	< .001
	Likert scale	269	0.36	[0.29, 0.44]	< .001	1-6 (false-true)	6	0.65	[0.43, 0.88]	< .001
	100-point slider	6	0.23	[-0.17, 0.62]	.256	16cm (false-true)	5	0.65	[0.42, 0.89]	< .001
	Binary scale	91	0.43	[0.30, 0.56]	< .001	Dichotomous	7	0.53	[0.27, 0.79]	< .01
Response scale (2)	Odd	119	0.30	[0.18, 0.43]	< .001	Odd	52	0.47	[0.38, 0.55]	< .001
	Even	241	0.40	[0.32, 0.47]	< .001	Even	13	0.60	[0.42, 0.78]	< .001
Veracity	True	83	0.46	[0.36, 0.56]	< .001	/	/	/	/	/
	False	81	0.33	[0.23, 0.42]	< .001	/	/	/	/	/
	None	12	0.31	[0.12, 0.49]	< .001	/	/	/	/	/
	Face-to-face	196	0.31	[0.20, 0.43]	< .001	Paper and pencil	42	0.59	[0.51, 0.69]	< .001

Testing environment	Online	169	0.40	[0.32, 0.48]	< .001	Computer	14	0.33	[0.16, 0.50]	< .01
--------	-----	------	--------------	--------	----------	----	------	--------------	-------

*In Dechêne et al. (2010), the exposure time was not differentiated between the exposure phase and the test phase.

9. Comment: Please clarify, because I see no reason why the statistics are not presented in a more complete way, with their respective degrees of freedom and effect sizes.

Response: We apologize for the lack of clarity. We agree that providing both the regression coefficients and their associated degrees of freedom can clarify our findings. Previously, we were concerned that reporting degrees of freedom in a multivariate imputation context might be confusing, because they do not align with a typical dataset's interpretation. Specifically, when dealing with missing data via multiple imputation, each imputed dataset is analyzed separately, and the parameter estimates and standard errors are then pooled accounting for within-imputation variance and between-imputation variance. Thus, the resulting pooled degrees of freedom are generally not simply the sample size minus the number of estimated parameters but also account for the variability introduced by multiple imputations (van Buuren, 2018). Similarly, effect sizes in a meta-regression with numerous categorical moderators depend heavily on the chosen baseline. Hence, in our initial draft, we opted to present standardized estimates in a figure. We have clarified this in our manuscript as follows:

- Lines (210-215): "We created 20 imputed data sets, each with a maximum of 20 iterations to ensure convergence of the chained equations. We then pooled the resulting coefficients and their variances according to Rubin's rules to obtain the meta-analytic estimates ⁴⁶. As a result, the adjusted degrees of freedom associated with t-statistics also account for the variability introduced by multiple imputations ⁴⁷."

Additionally, in the revised manuscript, we now include the degrees of freedom and the subgroup effect size estimates in the main text and in Table 2. For example, see:

- Lines (228-233): "The illusory truth effect was significantly larger in studies using statements ($g = 0.41$, 95% CI [0.33, 0.48]) compared to news headlines ($g = 0.17$, 95% CI [0.01, 0.34]), $b = 0.25$, 95% CI [0.02, 0.47], $t(162) = 2.16$, $p = .033$, but not compared to opinion-based claims ($g = 0.47$, 95% CI [0.20, 0.73], $t(117) < 1$. Effect sizes did not significantly differ between news headlines and opinion-based claims, $t(127) = 1.10$, $p = .275$."

10. Comment: I would like to point out that when $t < 1$, the p-value calculation is unstable and unreliable, so it does not need to be presented (see, for example, Voelkle et al., 2007).

Response: We appreciate the Reviewer's suggestion and have removed the p-values associated with a t-statistic < 1 in our revised manuscript.

11. **Comment:** Finally, when I finished reading the discussion, I realized that the article could gain more consistency if it were to adopt a theoretical or pragmatic objective instead of an unbalanced mixture of both. When I read it, my interest was theoretical, and I wanted to understand the cognitive mechanisms underpinning the effect. But the article might aim for a pragmatic framework. In that sense, it is important to understand that misinformation goes beyond the truth illusion effect and should not be conflated. But if this is the main concern, it would be important to understand what the moderators say about the possibility of spreading fake news or how we can prevent it.

Response: We thank the Reviewer for their suggestion. Based on the Reviewer's suggestion, we have revised the manuscript to more clearly articulate the aims of the present meta-analysis and how they are integrated. Specifically, we now clarify at the end of the introduction section that the primary goal of this meta-analysis is to investigate the cognitive mechanisms and boundary conditions of the illusory truth effect, while also examining how these mechanisms relate to the spread of misinformation. While we agree that strictly pragmatic focus can improve the coherence of the article, we believe that our meta-regression should also discuss theoretical mechanisms whenever it is possible. Our Discussion section now more explicitly focuses on the practical implications of the meta-analysis.

We also have added a sentence to the Discussion section to emphasize that repetition-based fluency is only one of many mechanisms contributing to misinformation belief and spread as follows:

- Lines (514-517): "Nevertheless, it is important to note that the illusory truth effect captures only one mechanism (e.g., repetition-based fluency) among many that contribute to the belief and spread of misinformation."

Finally, we wish to note our selection of moderators was guided more by pragmatic considerations than by strict alignment with any one theoretical framework. This is because the moderators that could be reliably extracted across studies do not systematically align with the constructs proposed by existing theoretical models. Moreover, given the current limitations in the number of effect sizes available per moderator combination, we are not able to reliably examine interactions between moderators, which prevents us from explicitly testing the core predictions of theoretical accounts. As a result, while our findings provide theory-relevant

insights (e.g., regarding processing depth, fluency-based processes, encoding conditions...), they should not be interpreted as adjudicating between competing cognitive models. We have clarified this limitation in the manuscript:

- Lines (483-492): “Moderator selection was primarily guided by pragmatic considerations than by alignment to a specific theoretical framework. Thus, they should not be interpreted as direct tests of cognitive models. In fact, the moderators that could be reliably extracted across studies do not systematically align with key constructs from existing theoretical accounts. Moreover, the limited number of effect sizes per moderator combination prevents us from testing interaction effects that would be necessary to evaluate core theoretical predictions. While our findings provide relevant insights to theory-relevant processes (e.g., fluency, depth of processing, encoding conditions), they should not be interpreted as conclusions about the validity of competing cognitive models.”

12. Comment: I also have two small details that could assist in the more detailed revision of the presented text:

Line 37: “after a certain delay” should be changed to “In most studies after a certain...”

Line 39: “... and new statements previously unseen in a random order”

Response: We thank the Reviewer for his/her suggestions. We have revised the manuscript accordingly.

Reviewer 4 Comments

Overall Comment: This meta-analysis presents a comprehensive and updated synthesis of the illusory truth effect, incorporating 108 studies and 195 effect sizes—a significant expansion compared to the previous meta-analysis by Dechêne et al. (2010). The authors employ a three-level meta-analysis framework to account for dependencies among effect sizes within studies, use meta-regression to explore sources of heterogeneity, and conduct sensitivity analyses with multiple imputation to address missing data under the MAR assumption. These methodological advancements strengthen the reliability of the findings.

From my perspective as a statistical reviewer, the study implements several best practices in meta-analysis. However, there are specific aspects related to effect size calculation, heterogeneity, and small-study effects that could benefit from further refinement and discussion.

Overall, this meta-analysis provides a rigorous and updated synthesis of the illusory truth effect. The authors should be commended for their methodological choices, particularly in their use of three-level modeling and sensitivity analyses. However, I encourage further methodological refinement in the areas of effect size calculation, heterogeneity interpretation, and small-study effect adjustments. Addressing these points would enhance the robustness and interpretability of the findings.

Recommendations for Revision

1. Consider Becker's method for standardizing mean differences in within-group designs to reduce imputation uncertainty.
2. Reflect on the impact of standardization and hidden multiplicity on effect size heterogeneity in the discussion.
3. Implement small-study effect corrections, particularly the PEESE method, to provide more conservative effect size estimates.
4. Report regression coefficients and the proportion of heterogeneity explained by meta-regression in the main text.
5. Modify funnel plot and provide supplementary forest plots stratified by moderators.

Response: We thank the Reviewer for his/her appreciation of our work and the suggestions to improve our methodology.

Standardized Mean Differences and Imputation of Missing Data

- 1. Comment:** The authors rely on standardized mean differences (Hedges' g) as their primary effect size metric. While group means are often available, the standard deviations and correlations needed for standardizing within-group designs are frequently missing. To address this, the authors use imputation from external sources. However, an alternative approach that might reduce uncertainty in the pooled estimates is the method proposed by Becker, which only requires standard deviations from the first assessment occasion (see Morris & DeShon, 2002). This would eliminate the need for imputed correlations and likely yield more less uncertain effect size estimates.

Response: We appreciate the Reviewer's insightful suggestion. Indeed, employing Becker's approach is a valuable alternative for computing within-subject effect sizes since it only requires standard deviations from the first assessment, thereby reducing the need to estimate both the pre-post correlation and the second-assessment standard deviation. However, as Morris and DeShon (2002) note, some of these estimations remain necessary for determining the sampling variance of the effect size. To evaluate whether Becker's formula yields less uncertainty, we compared it with Borenstein's formula under our different imputation strategies. Before correction, using Becker's formula under median-imputed values produced an effect size estimate of $g = 0.64$, 95% CI [0.56, 0.72], which ranged from $g = 0.56$, 95% CI [0.49, 0.64] to $g = 0.75$, 95% CI [0.66, 0.85] for the maximum and minimum imputations, respectively. In contrast, Borenstein's formula yielded an estimate of $g = 0.57$, 95% CI [0.51, 0.64] under median imputation, ranging from $g = 0.46$, 95% CI [0.40, 0.51] to $g = 0.65$, 95% CI [0.57, 0.72]. Although these differences are small, Becker's formula ultimately yielded a slightly larger span of confidence-interval bounds for minimum vs. maximum estimates (0.36) compared to that of Borenstein's formula (0.32). Therefore, we opted to continue using Borenstein's formula for our main analyses. A comparison of the results of the pooled estimates of both approaches is now reported in the Supplementary Table 4. We also have revised the manuscript to describe this sensitivity analysis in the main text as follows:

- Lines (815-821): "We also conducted a sensitivity analysis using Becker's formula to compute the standardized mean difference⁸⁵. The alternative formula could reduce uncertainty in the pooled estimates caused by missing outcomes imputation, as it only requires standard deviations from the first assessment. Computation using Becker's formula yielded effect sizes close to those from Borenstein's but with slightly wider confidence intervals (see Supplementary Table 4). Therefore, we reported the more conservative Borenstein estimates in the results.

Heterogeneity and Potential Multiplicity in Analytical Methods

2. **Comment:** High heterogeneity is common in psychological meta-analyses, particularly those involving RCTs. Beyond variation in population, intervention, and outcome characteristics, one likely source of heterogeneity is the diversity of analytical methods used in the original studies to derive effect size estimates. The authors report substantial within- and between-study heterogeneity ($I^2 \approx 71\%-81\%$), which raises interpretational concerns about the aggregated effect size. Additionally, reliance on standardized mean differences may further contribute to heterogeneity, as prior research has suggested (e.g., <https://psycharchives.org/en/item/1b670ec4-fec9-43d8-b72c-ecb85b27d0e5>). I recommend including a critical reflection on how potential hidden multiplicity and standardization might inflate heterogeneity in the discussion section. With regard to the impact of standardization, meta-regression on the coefficient of variation might be considered as another worthwhile sensitivity analysis.

Response: We thank the Reviewer for his/her helpful comment and for highlighting how both standardization and hidden multiplicity can both inflate heterogeneity. To evaluate the impact of standardization, we ran an additional three-level meta-regression with the pooled coefficient of variation as a moderator, see:

- Lines (186-192): “To investigate whether heterogeneity in standard deviations contributed to variation in effect sizes, we conducted a three-level meta-regression using the coefficient of variation (CV) as a moderator. This analysis revealed a significant negative relationship between CV and the standardized mean differences, $b = -0.53$, 95% CI [0.39, 0.58], $t(330) = -4.30$, $p < .001$, indicating that effect sizes were smaller in studies with higher relative dispersion of truth judgments. This finding suggests that variability in outcome precision may partially account for between-study heterogeneity.”

We now discuss this limitation in the revised manuscript as follows:

- Lines (452-475): “Nevertheless, several limitations of the current meta-analysis should be acknowledged. While we applied a PEESE method to gauge and correct potential publication bias, its accuracy deteriorates under high heterogeneity⁶⁷, the corrected estimate should thus be interpreted with caution. Recent methodological works suggest that using standardized mean difference (e.g., Hedge’s g) may inflate between-study

heterogeneity estimates. For instance, heterogeneity in standard deviations alone can raise I^2 even when underlying mean differences are relatively stable⁶⁸. Thus, part of the observed heterogeneity may be a statistical artefact of standardization rather than true differences in the illusory truth effect. To explore this possibility, we examined whether differences in the relative variability of truth judgments across studies contributed to inconsistencies in effect sizes. Results from this additional analysis indicated that studies with more dispersed outcomes tended to report smaller standardized effects, suggesting that statistical noise in measurement precision may inflate heterogeneity estimates. This finding supports the view that variability in study-level standard deviations may contribute to between-study heterogeneity. Furthermore, the potential influence of hidden multiplicity warrants caution. Multiplicity often leads to selective reporting, where researchers may preferentially publish outcomes that align with their hypotheses, while omitting null or contrary results. Many included experiments reported multiple eligible effect sizes (e.g., across different time points: immediate vs. delayed test). While we extracted all eligible effects and used a multilevel meta-analytic approach to account for within-study dependence, this does not fully eliminate the risk of bias introduced by multiplicity. The analytic choices made by original authors (e.g., which outcomes to report or emphasize), as well as those made during data extraction, may still influence aggregated effect estimates despite statistical adjustment.”

Small-Study Effects and Publication Bias

- 3. Comment:** The study acknowledges the presence of small-study effects, which the authors attribute to publication bias. However, while the presence of small-study effects is noted, no corrective methods (e.g., PET-PEESE, selection models) are applied. Given the substantial heterogeneity, publication bias is a serious concern, and the lack of adjustment likely means that the reported effect size estimates are inflated. Based on the evaluation of small-study effect corrections by Carter, Schönbrodt, Gervais, and Hilgard (2019), I recommend implementing the PEESE method. The authors claim in the discussion section, that such methods cannot be implemented with multi-level meta-analyses. I do not agree, because the PEESE approach is essentially an ad-hoc meta-regression on sample precision. As such it is supposed to provide a pragmatic and conservative correction under conditions of questionable research practices and high heterogeneity.

Response: We appreciate the Reviewer’s recommendation and agree that some form of small-study effect adjustment is warranted, especially given our finding of potential publication bias

and substantial heterogeneity. Initially, we were unsure about applying corrective methods such as PEESE to a three-level meta-analysis, as they were developed primarily for two-level models and can perform less reliably under high heterogeneity (e.g., Stanley, 2017). Following the Reviewer's suggestion, we have now implemented the PEESE correction to provide a more conservative assessment of how publication bias might inflate our observed effects. We thus reported these PEESE-adjusted estimates alongside our main results, discussing any implications for interpreting our overall findings as follows:

- Lines (158-161): "Following the evidence of funnel-plot asymmetry, we implemented a Precision-Effect Estimate with Standard Error (PEESE) correction to obtain a bias-adjusted overall effect."
- Lines (166-168): "The PEESE-corrected overall effect size ranged from $g = 0.35$, 95% CI [0.29, 0.42] to $g = 0.44$, 95% CI [0.37, 0.51] depending on the values imputed for missing standard deviations ($u = 136$) and missing correlation coefficients ($u = 282$)."
- Lines (453-455): "While we applied the PEESE method to gauge and correct potential publication bias, its accuracy deteriorates under high heterogeneity ⁶⁷, the corrected estimate should thus be interpreted with caution."

4. **Comment:** Additionally, I suggest using this adjusted model as a baseline before adding substantive moderators. This would allow for a clearer interpretation of the impact of study characteristics on the illusory truth effect, free from confounding small-study biases.

Response: We appreciate the Reviewer's suggestion. In our revision, we used the PEESE-adjusted model as a baseline prior to introducing moderators, which allows us to explore the role of potential moderators while controlling for small-study biases. We also have revised the manuscript as follows:

- Lines (160-161): "Furthermore, this adjusted model served as the baseline for all subsequent moderator analyses to support an unbiased interpretation of moderator effects."

Reporting of Meta-Regression Results

5. **Comment:** Moderator Impact on Effect Size Scale: Regression coefficients should be explicitly reported in the main text (rather than supplementary materials) with confidence intervals to provide readers with a clearer understanding of how each moderator influences

the standardized mean difference. A modification of Figure 4 to include these coefficients would be particularly useful.

Response: We agree that regression coefficients and their corresponding confidence intervals greatly help readers in understanding how each moderator influences the standardized mean difference. We have revised Figure 3 (previously figure 4) to describe the regression coefficients and their 95% confidence intervals rather than the Z-Values. Additionally, we included these coefficients and confidence intervals in the main text of our revised manuscript. For example, see:

- Lines (237-245): “A significant moderating effect was found for the content of the task used during the exposure phase. When participants were asked for a truth judgment during the exposure phase ($g = 0.10$, 95% CI [-0.03, 0.23]), the illusory truth effect was consistently reduced compared to irrelevant tasks ($g = 0.42$, 95% CI [0.34, 0.50]), $b = -0.35$, 95% CI [-0.50, -0.20], $t(303) = 4.62$, $p < .001$, or passive processing ($g = 0.47$, 95% CI [0.34, 0.60]), $b = -0.47$, 95% CI [-0.70, -0.23], $t(221) = 3.94$, $p < .001$. However, the illusory truth effect did not significantly vary when the exposure task only instructed participants to read the statements compared to performing an irrelevant task, $t(218) = 1.32$, $p = .190$.”
- Lines (1146-1153): “**Figure 3.** Bar plot of moderator analysis of the illusory truth effect. The regression coefficient values and their 95% confidence intervals were computed using median estimates imputation for missing outcomes of interest. The reference group for each categorical moderator was selected based on the most frequently observed subgroup in the dataset. Negative coefficients indicate a reduction in the illusory truth effect relative to the reference group, positive coefficients indicate an increase. All coefficients are corrected for small-study effects using the PEESE method (precision-effect estimate with standard error).”

6. **Comment:** Proportion of Heterogeneity Explained: The proportion of between-study variance accounted for by the meta-regression should be reported (e.g., pseudo R^2). This would provide insight into the extent to which the included moderators successfully explain the observed heterogeneity.

Response: We fully agree that reporting the proportion of heterogeneity explained by the moderators (e.g., pseudo- R^2) can clarify how well our chosen variables account for the between-

study variance. In the revised manuscript, we have included such measure by comparing the variance components from our baseline three-level model to those from our final meta-regression model, see:

- Lines (311-315): “Averaged across the 20 multiple imputed datasets, the pseudo-R² for the between-study variance was 0.28 (range = 0.26-0.30). On average, 28 % of the between-study heterogeneity relative to the PEESE-adjusted baseline model was accounted for by the moderator set.”
- Lines (327-329): “Together, these moderators explain 28% of the between-study variance, indicating that their relative importance in moderating the illusory truth effect.”
- Lines (861-867): “We additionally quantified the amount of between-study heterogeneity explained by the moderator set with the pseudo-R² statistic ⁸⁷:

$$Pseudo R^2 = \frac{\tau^2_{RE} - \tau^2_{ME}}{\tau^2_{RE}}$$

Where τ^2_{RE} is the level-3 variance from the baseline random-effects model and τ^2_{ME} is the residual level-3 variance from the mixed-effects model that includes all moderators. The statistic reflects the proportion of between-study variance accounted for by the moderators.”

Visualization of Effect Heterogeneity

7. **Comment:** The authors include a funnel plot for assessing publication bias, that could be improved with regard to the contour enhancement proposed by Peters, Sutton, Jones, Abrams and Rushton (2008). Specifically, the contours are currently constructed around the naïve pooled effect estimate (that is unadjusted for small study bias). Instead, they should be constructed around to H0: $g = 0$ effect scenario, to facilitate the identification of those studies, that reported effect estimates just below the conventional 0.05% level. Precision-adjusted or naïve pooled effect estimates (and their CI regions) could then simply be added as a top-layer of figure elements.

Response: We appreciate the Reviewer’s suggestion to improve our funnel plot. We agree that centering the funnel plot around $g = 0$ can facilitate the identification of small-studies bias. Following the Reviewer’s recommendations, we revised the Figure 3 by:

1. Reconstructing the funnel plot so that its contours are based on $H_0: g = 0$ rather than on the naïve pooled effect estimate.
2. Adding the naïve and PEESE-adjusted pooled estimate and its confidence interval as a top-layer of figure elements.

We also revised the interpretation in regard to the new figure in the revised manuscript as follows:

- Lines (149-158): “Visual inspection of the contour-enhanced funnel plot ⁴² revealed asymmetry in the distribution of effect sizes (Figure 3). Most imprecise studies (lower part of the plot) fall in the $p < .05$ regions to the right of zero, whereas few comparable studies appear in the adjacent above the conventional 0.05% area. This pattern suggests that non significant or negative small study results are underrepresented. Additionally, most highly precise estimates at the top of the funnel cluster around the bias-adjusted effect size, whereas the unadjusted pooled effect estimate falls farther to the right, implying inflation by small-study effects. This observation was confirmed using an Egger’s regression test ⁴³ accounting for dependencies across effect sizes⁴⁴. Results indicate asymmetry in the funnel plot, $t(364) = 9.77, p < .001$, which suggests the presence of small-study effects, one possible component of publication bias ⁴⁵.”
- Lines (1133-1137): “**Figure 1.** Contour-enhanced funnel plot to visualize publication bias in the illusory truth effect literature. Each point represents a single effect size estimate. The contour shading reflects thresholds for statistical significance based on null-centered z-values. Dashed vertical lines mark the pooled point estimates from the unadjusted (blue) and PEESE-adjusted (red) models, with their 95% confidence intervals regions.”

- 8. Comment:** Additional visualization of the data would further improve interpretability: Forest Plots Stratified by Key Moderators: Presenting supplementary forest plots ordered by major moderators (e.g., exposure task and delay, truth warning) would allow readers to see how effect sizes vary within each subgroup.

Response: We agree that stratified forest plots greatly enhance clarity by illustrating how effect sizes vary under different experimental conditions. In the revised manuscript, we have included separate forest plots in the Supplementary Information, grouped by major moderators (i.e., item

type, exposure task, veracity cues, and presentation time at exposure). These subgroup-specific plots should provide a clear visual comparison of effect sizes within each category, see:

- Lines (314-315): “Supplementary Figures 1-17 display forest plots stratified by item type, exposure task, veracity cues, and presentation time at exposure.”

9. Comment: My own brief screening using PubMed and the search term from the paper returned several new primary studies on the illusory truth effect that have been published since Sep 2022. Although I don't assume, that those new studies will contribute substantially more information than currently compiled by the authors, an update of the meta-analysis would definitely be worthwhile.

Response: We conducted an updated literature search on February 24th, 2025, using the same databases and keywords covering the period from September 2022. This additional screening allowed us to identify several relevant studies published after our initial search, including 64 new effect sizes from 38 studies reported in 19 manuscripts. In addition to the revision of our eligibility criteria (see response to Reviewers 1 and 2's comment 2), the revised meta-analysis now includes 366 effect sizes from 182 individual studies extracted from 99 articles. We have revised our manuscript and the flow diagram of study selection (Figure 4) to describe these newly included studies as follows:

- Lines (4-6): “In light of a growing number of studies published since 2006 and concern of publishing biases, we conducted a meta-analysis on 182 studies and 366 effect sizes (N = 31,184 participants) published from 1977 to 2025.”
- Lines (102-104): “In the present study, we conducted an updated meta-analysis of the illusory truth effect, synthesizing 182 studies and 366 effect sizes published between 1977 and 2025.”
- Lines (134-144): “The current meta-analysis included n = 99 articles describing k = 182 studies from which u = 366 effect sizes were extracted from 31,184 participants. Of the 93 articles identified in Henderson et al.⁸, 27 were excluded according to our inclusion and exclusion criteria. Compared to a previous meta-analysis conducted by Dechêne et al.³⁸, 6 out of 25 articles were excluded. Included articles were published between 1977 and 2025, with a median year of 2019. Most studies were published in a journal (n = 77) while other were unpublished studies (n = 9) or data set (n = 1), PhD (n = 6) or Master's thesis (n = 5) and conference presentations (n = 1). Most studies were

conducted in Europe ($n = 49$) and in North America ($n = 47$), with a few in Oceania ($n = 3$), Asia ($n = 1$), and South America ($n = 1$). References for all studies included in the meta-analysis are provided in the Supplementary Materials.”

- Lines (585-593): “We conducted an update to the systematic review on 24 February 2025, using the same databases, keyword string, and screening procedures that were applied to the original review. Limiting to records published after September 2022 returned 2433 titles/abstracts. After removing 1234 duplicates with Zotero’s automatic tool and 26 duplicates through manual inspection, 1173 unique records remained. Title and abstract-screening excluded 1117 of these records, leaving 56 additional articles to be assessed for eligibility, 10 of which could not be retrieved even after messaging the authors. Figure 4 provides the flow diagram of systematic review and study selection, and Supplementary Table 3 lists all excluded effect sizes and studies with justifications.”

We also described the change in our results with the addition of new studies published after 2022, the revision of our eligibility criteria, the addition of publication bias correction and new moderators, see:

- Lines (6-11): “After correcting for small-study effects, we observed a small illusory truth effect ($g = .37$, 95% confidence interval [0.30, 0.44]), with a substantial within and between-study heterogeneity. Here, we show that multiple variables accounted for such heterogeneity, including the type of item, the exposure task, the use of veracity cues, and the duration of presentation on first exposure to the statement.”
- Lines (121-130): “In this work, we confirm that repeated exposure produces a small but robust increase in perceived truth (bias-corrected $g = 0.37$, 95% CI [0.30, 0.44]). The effect is systematically moderated by the nature of the item type and the design of the initial exposure, particularly the initial task, the presentation duration and the presence of explicit veracity cues. The meta-regression indicates that the combined moderators account for approximately 28% of the between-study variance, highlighting their explanatory value. Together, these findings suggest the importance of designing interventions that target the initial moments of exposure. Encouraging early truth evaluation or embedding explicit accuracy signals during initial encounters with information may significantly reduce the later influence of repetition.”
- Lines (164-172): “The three-level meta-analysis of the illusory truth effect for 366 effect sizes retrieved from 182 studies described in 99 unique articles yielded an estimated

small to medium size effect. The PEESE-corrected overall effect size ranged from $g = 0.35$, 95% CI [0.29, 0.42] to $g = 0.44$, 95% CI [0.37, 0.51] depending on the values imputed for missing standard deviations ($u = 136$) and missing correlation coefficients ($u = 282$). This interval was obtained from a sensitivity analysis using the minimum and maximum estimated values (i.e., minimum/maximum of previously observed values) for imputed standard deviations and correlation coefficients. When the median estimated values were taken, we found an effect size $g = 0.37$, 95% CI [0.30, 0.44] (see Table 1).”

- Lines (175-181): “Depending on the computed missing values, the estimated variance components varied for τ^2 Level 2 = 0.054 to 0.101 and τ^2 Level 3 = 0.099 to 0.148 after accounting for small-study bias. The related I^2 value indicated that 26% to 36% of the total variance could be attributed to within-study differences in effect sizes, and that between-study heterogeneity explained $I^2 = 36\%$ to 70% of the total variation. We found that the three-level model provided a significantly better fit compared to a two-level model with level 3 heterogeneity constrained to zero ($\chi^2(1) = 32.65, p < .001$).”
- Lines (182-185): “The strong within and between-study heterogeneity, $Q(364) = 4445, p < .001$, calls for caution in interpreting the estimated overall effect size and suggests that differences in the methods used in the different studies may be at the root of the observed heterogeneity (see the meta-regression below for such analysis).
- Lines (323-325): “Synthesizing 182 studies (366 effect sizes; $N = 31,184$), we found a small illusory truth ($g = 0.37$, 95% CI [0.30, 0.44]), after correcting for publication bias.”
- Lines (499-506): “In conclusion, this updated meta-analysis confirms the robustness of the illusory truth effect, with a small but consistent overall effect ($g = 0.37$) after correcting for publication bias. While repetition reliably increases perceived truth, the effect varies substantially across study designs. Critical moderators are largely associated with the first exposure including the exposure task, the presence of veracity cues and the length of presentation time. Importantly, the effect remained robust across content types, repetition formats, and testing environment, emphasizing its real-world relevance to information exposure. These findings have clear implications for misinformation interventions.”

Systematic review and meta-analysis of the evidence for an illusory truth effect and its determinants

Response to Reviewers

We thank the Reviewers for their constructive comments. All modifications in the manuscript were highlighted in yellow color. During the process of re-examining the coding based on Reviewers' comments, we identified one error in our dataset related to the coding of one of the moderators. We sincerely apologize for this oversight. The error has been corrected in the final version of the manuscript. Importantly, this correction did not affect the pattern, strength, or interpretation of any of the findings reported in the original submission. In the revised manuscript, we implemented the following changes based on Reviewers' valuable suggestions:

1. Clarified the objectives of the meta-analysis in the Introduction and reorganized the Discussion section to ensure that (a) the interpretation of results is directly aligned with these aims and (b) the discussion of moderators are more coherently grouped.
2. Strengthened the conceptual framework by introducing a new Table 1, which defines all moderators and provides their coding schemes as well as their conceptual rationale.
3. Conducted additional exploratory analyses focusing on the moderating effect of cue type by examining the interaction between cue type and cue valence.
4. Standardized the coding approach by clarifying moderator categories and adding study-level descriptions of each moderator in the Results section to improve transparency.
5. Removed the "balanced cues" category for the cue valence moderator, in accordance with our general approach for handling effect sizes that could belong to multiple categories.
6. Improved the transparency of results by reporting the number of effect sizes per moderator category and by clarifying the handling of missing data through multivariate imputation.

Detailed responses to each point raised by Reviewers are provided below.

Reviewer 2 Comments

• **Overall Comment:** I was one of the first two reviewers from the original review. I greatly appreciate the efforts the authors have put into updating their manuscript based on the reviewers' suggestions. I believe that significant improvements have been made to this manuscript. I do have some concerns and suggestions regarding the new "veracity cue" moderator. I also have a number of more minor points following the updates that have been made.

As noted in the original review, I have more limited expertise in the statistical procedures of meta-analysis and trust other reviewers will address the technical aspects. The statistical additions that were made during this revision were largely those I am not familiar with.

Response: We thank the Reviewer for his/her comments and the suggestions for improving our manuscript.

New veracity cues moderators:

1. **Comment:** In response to our question about the exclusion of common knowledge facts and opinions, the authors added back in opinions (claims), effect sizes previously included for being considered common knowledge (widely known facts, highly implausible claims). In doing these additions, the authors also decided to analyze effect sizes that included veracity cues at exposure. The addition of the studies using explicit veracity cues was not something we had mentioned, but I believe that including these studies provides really valuable information and greatly appreciate this addition.

Response: We thank the Reviewer for the positive feedback and are pleased that the inclusion of the veracity cue moderator was found to be valuable.

2. **Comment:** However, I'm having a hard time following what these categories entail. For example, what was included in "post-rating feedback"? Is this always occurring at the item level?

Response: We thank the Reviewer for highlighting this important point. We recognize that our description of veracity cue categories may have caused confusion, and we appreciate the opportunity to clarify.

In our meta-analysis, we did not use “post-rating feedback” as a formal moderator or coding category. Rather, this term was used descriptively to refer to how certain primary studies implemented veracity cues. Specifically, the literature includes a variety of cue types that differ in both timing (e.g., concurrent with exposure, immediately post-exposure, or after a block of trials) and format (e.g., source cues, labels, factual feedback).

To address this concern, we have revised the manuscript as follows:

- The new Table 1 includes an explicit definition of the veracity cue moderator: “The valence of indicator of factuality during exposure.” Categories are also labeled more clearly as: no cue, true cues, false cues.
- Lines (456-462): “To explore how variability in the type of veracity cue (e.g., source reliability cues, epistemic qualifiers, labels, immediate or delayed feedback) contribute to inconsistent findings in the literature, we conducted an exploratory analysis. We found that the moderating influence of veracity cues may depend on cue type, with stronger effects of feedback, source reliability cues, and epistemic qualifiers compared to labels. However, the reliability of these conclusions is limited by the small number of effect sizes per category, and they should thus be interpreted with caution (see Supplementary Materials).”

3. **Comment:** What is the difference between a “positive cue” (category in the results) and a “true cue” (category used elsewhere)?

Response: We thank the Reviewer for drawing attention to this important inconsistency and apologize for the confusion caused. The term “positive cue” was used to refer to cues suggesting that a statement is true, what we elsewhere referred to as a “true cue”. To ensure clarity and consistency, we have now standardized the terminology throughout the manuscript. Specifically, we consistently use the terms “true cue” and “false cue” to refer to indicators suggesting a statement is true or false, respectively. We have removed the terms “positive cue” and “negative cue” entirely to avoid ambiguity.

4. **Comment:** What about studies where some items had cues and others didn’t – were these included?

Response: We thank the Reviewer for raising this important point. Yes, studies that included a mix of cued and uncued items were included in our meta-analysis. As described in the Methods

section under Extraction of Moderating Variables, this situation corresponds to cases where a single study matched multiple categories of a moderator (e.g., “true/false cues” and “no cues”). Our procedure was as follows:

“When a study included effect sizes potentially matching multiple categories of a moderator, we extracted each effect size individually whenever the data (e.g., distinct outcomes or open datasets) permitted clear separation. In instances where effect sizes were reported in a combined form and could not be disentangled by moderator category, we coded the moderator as missing.”

For instance, Mitchell et al. (2005) presented participants with statements paired with different types of cues (e.g., true, false, or neutral) manipulated within-participants. The study reported the average proportion of “true” responses separately for each cue condition (i.e., true cues, false cues, uncued), as well as for new items. This allowed us to extract three distinct effect sizes based on cue condition. Importantly, the statistical dependency between these effect sizes resulting from the fact that they involved the same participants was appropriately modeled in our multilevel analysis.

However, if the study had only reported an aggregated illusory truth effect that collapsed across cue types (i.e., without separating true, false, and neutral conditions), we would have extracted this single effect size and treated the cue moderator for that effect size as missing.

Upon further reflection, we also decided to revise our initial coding strategy. Specifically, we removed the previously defined “balanced cues” category, which had been used to label studies that employed cues but did not report separate effect sizes for true and false conditions. We concluded that this category introduced unnecessary ambiguity and was inconsistent with our methodological approach. Accordingly, we removed the “balanced cues” category and coded the cue moderator as missing for these effect sizes.

5. **Comment:** In the analysis comparing claims labeled as false or true, is this including all types of “false cues” and “true cues” before/during/after ratings, or just certain ones?

Response: We thank the Reviewer for this important question. The analysis comparing claims labeled as false versus true includes all types of “true cues” and “false cues”, regardless of their timing or format. This reflects the broad operationalization of cue valence across studies in our meta-analysis. We have clarified this in the revised manuscript as follows:

- Lines (266-281): “**Veracity cues.** Veracity cues refer to any explicit or implicit indicators of a statement’s truth status, such as labels (“disputed by a third-party fact-

checker”), epistemic qualifiers (“it is unlikely that...”), source reliability cues (“all statements voiced by female are false”), or corrective feedback (e.g., immediately after initial exposure, or at the end of exposure phase). In our coding scheme, we classified these cues by valence (e.g., true, false, or no cue) regardless of the type of cue. The valence of veracity cues significantly moderated the illusory truth effect. Across cue types (e.g., any timing or format), providing false cues during the exposure phase ($g = -0.18$, 95% CI [-0.33, -0.03]) decreased the effect compared to no cues ($g = 0.43$, 95% CI [0.36, 0.50]), $b = -0.66$, 95% CI [-0.83, -0.49], $t(219) = 7.70$, $p < .001$, and providing true cues ($g = 0.70$, 95% CI [0.51, 0.88]), $b = -0.86$, 95% CI [-1.04, -0.68], $t(244) = 9.47$, $p < .001$). Additionally, explicitly labelling statements as true yielded a larger illusory truth effect compared to no veracity indication, $b = 0.20$, 95% CI [0.01, 0.40], $t(229) = 2.05$, $p = .042$. However, this was not robust with sensitivity analysis using maximum estimates of missing values, $t(275) = 1.15$, $p = .253$. No other moderating effects were found.”

6. **Comment:** The analysis refers to “explicitly labeling claims as false at exposure” but this seems like this would just be a subset of the “false cues” just ones where claims were explicitly labeled as false.

Response: We appreciate the Reviewer’s observation and apologize for the ambiguous wording. To clarify, our analysis indeed included all types of false cues, not just those that explicitly labeled claims as false at exposure. We have revised the manuscript to reflect this more precisely. Specifically, we now describe the effect of false cues in general, as follows:

- Lines (272-276): “The valence of veracity cues significantly moderated the illusory truth effect. Across cue types (e.g., any timing or format), providing false cues during the exposure phase [...] decreased the effect compared to no cues [...], and providing true cues [...].”
- Lines (453-454): “False cues reduced the impact of repeated exposure on perceived truth while true cues did not affect the illusory truth effect.”

7. **Comment:** The authors also refer to veracity cues as “explicit truth indicators”, but some of them (like including “most people know that...” in a claim) do not provide definite information about truth, while others do.

Response: We thank the Reviewer for this observation. We agree that the cues we examined vary in the extent to which they provide definitive information about truth. For instance, phrases like “most people know that...” imply consensus rather than objective verification. To reflect this distinction more accurately, we have revised the manuscript as follows:

- Lines (731-733): “We included the valence of veracity cues as a moderator to examine the impact of cues that signal truth or falsity, which are commonly used in misinformation interventions such as fact-checking.”

8. **Comment:** In the discussion, the authors also report veracity cues always attenuating illusory truth: (p. 18 –1 397): “Crucially, our findings suggest that explicit cues about the truth status of a statement significantly reduced the illusory truth effect.” However, the results reported indicate that, if anything, explicitly labeling claims as true INCREASES the illusory truth effect. Even if this effect wasn’t robust, it doesn’t seem to be the case that the authors find labeling claims as true decreases the effects of repetition.

Response: We apologize for the oversight. The original sentence in the discussion was indeed imprecise and could be misinterpreted. Our results indicate that false veracity cues consistently attenuated the illusory truth effect, whereas true cues showed a trend toward amplifying it, though this effect did not remain statistically significant in the sensitivity analyses. To clarify this in the revised manuscript, we have updated the discussion to more accurately reflect the differential effects of cue valence as follows:

- Lines (452-454): “Crucially, our findings suggested that the valence of veracity cues significantly moderate the illusory truth effect. False cues reduced the impact of repeated exposure on perceived truth while true cues did not affect the illusory truth effect.”

9. **Comment:** The authors also mention in the discussion (line 408) - “However, variability in the type of veracity cue (e.g., warning labels, post-rating feedback, or epistemic qualifiers) likely contributes to the inconsistent findings, especially given the substantial between-study heterogeneity observed in our analysis.” I think this is an important point, and given this, if the authors are to keep in these studies, I would encourage them to i) more clearly

define the different types of veracity cues and their categorization and ii) present effect sizes for different types of cues separately.

Response: We thank the Reviewer for this constructive suggestion. As noted in the Discussion, we indeed expect that the variability in the type of veracity cues may contribute to the heterogeneity observed across studies. Our primary motivation for examining cue valence as a moderator was to evaluate the potential of different veracity cues to serve as effective interventions, either mitigating or amplifying the effect of repetition in perceived truth. However, we agree that further differentiating cues by type could yield important insights into which interventions are most effective.

That said, analyzing cue type independently of cue valence poses interpretive challenges. Cue valence is typically manipulated within participants, while cue type varies between studies. Thus, collapsing across valence to report cue-type effects would obscure meaningful distinctions. To address this, and in response to the reviewer’s suggestion, we conducted an additional analysis using a recoded moderator that jointly accounts for cue type and valence.

We identified 5 types of cues used in literature during our re-examination of the articles:

1. Epistemic qualifiers (e.g., using qualifiers such as “it is certain that...” or “it is widely known that...” before the statement),
2. Source reliability cues (e.g., instructing participants that all statements voiced by male/female are true/false),
3. Labels (e.g., statements presented with “true” or “false” labels or tags),
4. Immediate feedback (e.g., feedback given immediately after initial rating), and
5. Delayed feedback (e.g., feedback given at the end of exposure phase, or at the beginning of test phase).

Below, we present the descriptive effect sizes (PEESE-corrected) for each cue type and valence:

Category		Number of effect sizes (u)	Effect size (g)	95% CI
No cue		287	0.43	[0.36, 0.50]
Epistemic qualifiers	True cue	10	0.16	[-0.21, 0.52]
	False cue	7	0.01	[-0.27, 0.29]
Source reliability	True cue	16	0.64	[0.40, 0.87]
	False cue	16	-0.21	[-0.44, 0.02]

Labels	True cue	1	-0.60	[-2.25, 1.06]
	False cue	5	0.27	[-0.03, 0.57]
Immediate feedback	True cue	2	0.67	[0.30, 1.44]
	False cue	2	-0.58	[-1.12, -0.04]
Delayed feedback	True cue	7	0.75	[0.33, 1.17]
	False cue	7	-0.90	[-1.31, -0.50]

Comparing to no cues condition, we found significant reductions in the illusory truth effect in effect sizes using false cues based on:

- Delayed feedback: $g = -1.12$, 95% CI [-1.51, -0.73], $t(292) = 5.69$, $p < .001$
- Immediate feedback: $g = -0.91$, 95% CI [-1.44, -0.39], $t(293) = 3.31$, $p = .001$
- Source reliability: $g = -0.73$, 95% CI [-0.95, -0.50], $t(319) = 6.33$, $p < .001$
- Epistemic qualifiers: $g = -0.57$, 95% CI [-0.87, -0.29], $t(273) = 3.85$, $p < .001$

In contrast, false label cues did not significantly reduce the effect, $b = 0.04$, 95% CI [-0.29, 0.38], $t(197) < 1$.

Comparing cue types directly, false delayed feedback was significantly more effective than false epistemic qualifiers, $b = -0.55$, 95% CI [-1.04, -0.06], $t(263) = 2.20$, $p = .029$.

Regarding true cues, we found a significant increase in the illusory truth effect when using true cues with delayed feedback, $b = 0.49$, 95% CI [0.09, 0.89], $t(295) = 2.42$, $p = .016$.

All the described effects were robust in sensitivity analyses. However, some cue type and valence combinations were represented by very few studies, which limits the reliability and generalizability of these specific estimates. For this reason, we have included the full analysis in the Supplementary Materials, as it supports our main conclusions while providing additional detail without altering the overall pattern of results.

The revised manuscript now reads:

- Lines (266-281): “Veracity cues refer to any explicit or implicit indicators of a statement’s truth status, such as labels (“disputed by a third-party fact-checker”), epistemic qualifiers (“it is unlikely that...”), source reliability cues (“all statements voiced by female are false”), or corrective feedback (e.g., immediately after initial exposure, or at the end of exposure phase). In our coding scheme, we classified these cues by valence (e.g., true, false, or no cue) regardless of the type of cue. The valence of veracity cues significantly moderated the illusory truth effect. Across cue types (e.g., any timing or format), providing false cues during the exposure phase ($g = -0.18$, 95%

CI [-0.33, -0.03]; observed $u = 40$) decreased the effect compared to no cues ($g = 0.43$, 95% CI [0.36, 0.50]; observed $u = 287$), $b = -0.66$, 95% CI [-0.83, -0.49], $t(219) = 7.70$, $p < .001$, and providing true cues ($g = 0.70$, 95% CI [0.51, 0.88]; observed $u = 33$), $b = -0.86$, 95% CI [-1.04, -0.68], $t(244) = 9.47$, $p < .001$). Additionally, explicitly labelling statements as true yielded a larger illusory truth effect compared to no veracity indication, $b = 0.20$, 95% CI [0.01, 0.40], $t(229) = 2.05$, $p = .042$. However, this was not robust with sensitivity analysis using maximum estimates of missing values, $t(275) = 1.15$, $p = .253$. No other moderating effects were found.

- Lines (282-292): “We also conducted an exploratory analysis to assess the effect of the type of cue on the illusory truth effect. We found a significant decrease in the illusory truth effect when using false cues involving epistemic qualifiers ($g = -0.57$, 95% CI [-0.87, -0.29]; observed $u = 7$), source reliability cues ($g = -0.73$, 95% CI [-0.95, -0.50]; observed $u = 16$), immediate feedback ($g = -0.91$, 95% CI [-1.44, -0.39]; observed $u = 2$), and delayed feedback ($g = -1.12$, 95% CI [-1.51, -0.73]; observed $u = 7$). Among these, false delayed feedback was significantly more effective than false epistemic qualifiers in reducing the effect ($b = -0.55$, 95% CI [-1.04, -0.06], $t(263) = 2.20$, $p = .029$). In contrast, false labels showed no reliable effect ($g = 0.04$, 95% CI [-0.29, 0.38]; observed $u = 5$). Given the small number of effect sizes in several categories, these findings should be interpreted cautiously (see Supplementary Materials).”
- Lines (452-465): “Crucially, our findings suggested that the valence of veracity cues significantly moderate the illusory truth effect. False cues reduced the impact of repeated exposure on perceived truth while true cues did not affect the illusory truth effect. This finding provides additional support for the value of debunking interventions. To explore how variability in the type of veracity cue (e.g., source reliability cues, epistemic qualifiers, labels, immediate or delayed feedback) contribute to inconsistent findings in the literature, we conducted an exploratory analysis. We found that the moderating influence of veracity cues may depend on cue type, with stronger effects of feedback, source reliability cues, and epistemic qualifiers compared to labels. However, the reliability of these conclusions is limited by the small number of effect sizes per category, and they should thus be interpreted with caution (see Supplementary Materials). Understanding these mechanisms is crucial for developing effective interventions to mitigate the spread of misinformation 19. Future studies should systematically investigate how different types of veracity cues influence the illusory truth effect.”

- Lines (731-736): “**Veracity cues.** We included the valence of veracity cues as a moderator to examine the impact of cues that signal truth or falsity, which are commonly used in misinformation interventions such as fact-checking. We coded each effect size that involved any indication of a statement’s factual status during the exposure phase. We classified veracity cues into three subgroups: true cues (n = 32), false cues (n = 41), or uncued (n = 287).”

Moderator justification

- 10. Comment:** I appreciate the authors adding more justification for their choice of moderators. This information is currently in the methods at the end of the paper, but I think it would be more useful to have this information earlier, either in the section at the end of the intro that lays out the moderators of interest or in the results section as the results are presented.

Response: We thank the Reviewer for this suggestion. We agree that presenting the rationale for our choice of moderators earlier in the manuscript improves clarity and allows readers to better follow the logic of our analyses, especially given that the full Methods section appears at the end. In response, we have introduced a new Table 1 at the end of the Introduction. This table summarizes all moderators included in the meta-analysis, along with their definitions, coding scheme, and conceptual rationale. We have also included a study-level description of each moderator in the Results section. Please see for an instance of the description implemented in the Results section:

- Lines (228-232): “**Item type.** Studies on the illusory truth effect use a variety of materials such as trivia statements, topic-specific or general knowledge statements, social media news headlines, conspiracy theories, political claims, marketing advertisements, and meaningless sentences. In our coding, these were grouped in three categories: standard statements, headlines, and claims.”
- Lines (238-240): “**Repetition study design.** Repetition can be manipulated either within-participants (e.g., each participant judges both repeated and new items), or between-participants (e.g., one group judges only repeated items and another only new items).”

Table 1. Description of moderators included in the meta-analysis, with definitions, coding scheme, and conceptual rationale..

Moderator	Definition	Coding Scheme	Conceptual Rationale
Item type.	The format of the statements used.	Categories: standard statements (declarative sentences), headlines (social media or news-style headlines), claims (unverifiable assertions such as opinions or marketing slogans).	Tests the generalizability of the illusory truth effect across different content formats.
Repetition study design.	The experimental design used to manipulate repetition.	Categories: within-participant (repeated and new items judged by the same participant) or between-participant (separate groups judged repeated or new items).	Clarifies whether awareness of repetition influences subsequent truth judgments.
Instructions during the exposure phase.	The task participants completed during the exposure phase.	Categories: truth judgment, irrelevant task (cognitive engagement unrelated to truth such as interestingness rating, categorizing...), passive processing (no instructions or low-effort task such as reading, listening...).	Evaluates how levels of cognitive engagement during encoding influence susceptibility to repetition.
Truth warning at the exposure phase.	The presence or absence of explicit instructional warning before first exposure that some statements might be false.	Categories: warning, no warning.	Determines whether warnings before first exposure attenuate the illusory truth effect.

Truth warning at the test phase.	The presence or absence of explicit instructional warning before the test phase that some statements might be false.	Categories: warning, no warning.	Clarifies whether the timing of warnings (exposure vs. test) affects belief correction.
Veracity cues.	The valence of indicator of factuality during exposure.	Categories: no cue, true cues, false cues.	Examines whether veracity cues mitigate the illusory truth effect.
Delay between exposure and test phases.	The time interval between the exposure and test phases.	Categories: no delay, within day, up to one week, more than one week.	Evaluates whether the illusory truth effect decays across time intervals.
Repetition type.	The nature of repetition between exposure and test phases.	Categories: verbatim (exact repetition), gist (paraphrased repetition).	Tests whether the effect extends to semantically coherent but non-verbatim repetitions (gist).

Modality of presentation.	The sensory channel through which statements were presented.	Categories: visual, verbal, both.	Evaluates the robustness of the effect across sensory channels and media contexts.
Presentation time during exposure.	The time duration of exposure to each statement at the exposure phase.	Categories: 5 seconds or less, more than 5 seconds, participant-paced.	Investigates whether exposure duration influences the illusory truth effect (e.g., fast vs. extended processing).
Presentation time during test.	The time duration allowed for participants to perform truth judgment for each statement at test phase.	Categories: 5 seconds or less, more than 5 seconds, participant-paced.	Examines whether response time constraints (e.g., fast vs. deliberative judgments) affect the illusory truth effect.
Response scale (type /structure).	The format of the scale used for truth judgments in the test phase.	Categories: binary, Likert, 100-point slider.	Tests how response format (binary vs. graded scales) influences measured effect sizes.
		Categories: even, odd.	

Proportion of true statements.	The proportion of factually true items in the stimulus pool.	Continuous variable from 0 to 1.	Clarifies whether varying base rates of truth moderate susceptibility to repetition.
Veracity.	The factual status of the item.	Categories: true, false, none (for unverifiable, meaningless or opinion-based statements).	Examines whether repetition biases judgments equally for true, false, and unverifiable statements.
Testing environment.	The context in which data collection occurred.	Categories: online, face-to-face.	Evaluates whether laboratory findings generalize to online settings where misinformation is prevalent.

11. **Comment:** I also think it would be useful to have the number of effect sizes per category reported with the results along with the number of effect sizes that were not included in that analysis. Given that some of the categories have very few effect sizes – e.g., $n = 6$ for balanced cues and 100-point sliders — and it would help put these results in context.

Response: We thank the Reviewer for this valuable suggestion. We agree that reporting the number of effect sizes per category is important for interpreting the strength and generalizability of moderator effects. In response, we have now included the number of observed effect sizes per category in the presentation of each moderator analysis.

Regarding the number of effect sizes that were not included due to missing moderator data: rather than excluding these effect sizes from the analyses, we addressed missing data by computation using multivariate imputation by chained equations (mice package in R). This statistical approach estimates missing moderator values based on patterns in the observed data (e.g., other moderator variables and effect sizes), allowing us to retain all studies in the analysis while reducing potential bias. As such, no effect sizes were excluded from the moderator analyses, even when a specific moderator was missing from some studies. We revised the manuscript to explain this method more clearly as follows:

- Lines (205-210): “To handle missing values in moderator variables, we used multivariate imputation by chained equations (mice)⁴⁷. This method fills in missing moderator values by estimating them based on patterns in the observed data (e.g., other moderators and effect sizes), preserving the relationships among variables and reducing the loss of information that would result from excluding cases with missing values.”

Also, please see for an instance of the revision implemented in the Results section as follows:

- Lines (232-237): “The illusory truth effect was significantly larger in studies using standard statements ($g = 0.41$, 95% CI [0.33, 0.48]; observed $u = 309$) compared to headlines ($g = 0.17$, 95% CI [0.01, 0.34]; observed $u = 43$), $b = 0.25$, 95% CI [0.01, 0.48], $t(147) = 2.10$, $p = .038$, but not compared to opinion-based claims ($g = 0.47$, 95% CI [0.20, 0.73]; observed $u = 12$), $t(147) < 1$. Effect sizes did not significantly differ between news headlines and opinion-based claims, $t(138) < 1$.”

- 12. Comment:** I think that Table 2 is an especially useful addition. I also think including the number of effect sizes not included in each moderation analysis would be helpful information.

Response: We thank the Reviewer for this feedback and appreciate the suggestion to improve Table 3 (formerly Table 2). As noted in our response to Comment 11, we used multivariate imputation to handle missing moderator data. This approach allowed us to retain all effect sizes in the analyses by estimating missing values based on observed data patterns, avoiding the exclusion of effect sizes due to missingness.

In Table 3, we report the number of effect sizes with observed values for each category. This provides readers with a clear view of how much direct data informs each moderator estimate. Since we also report the total number of effect sizes included in the meta-analysis, readers can easily infer the number of imputed cases for each moderator by comparing the observed counts to the total.

Imputed values are not analyzed independently, but rather are incorporated into the pooled estimates using standard procedures that already account for the added uncertainty due to missingness. Therefore, we chose not to include the number of imputed values directly in the table or in the main text in order to maintain visual clarity and avoid redundancy.

Nevertheless, if the Reviewer think that explicitly including the number of imputed cases would enhance clarity or transparency, we would be happy to provide this information.

Additional points

- 13. Comment:** I'm a bit thrown off by the opening sentences focusing on fake news (and not even fake news being repeated) given the paper is meant to be a review of the illusory truth effect broadly and most of this work doesn't use fake news as stimuli (as the next paragraph discusses).

Response: We thank the Reviewer for this observation. We agree that the original opening can be confusing by placing emphasis on fake news, which is not the exclusive focus of the illusory truth literature. Our intention was to frame the review in a way that highlights the societal relevance of the phenomenon, but we acknowledge that doing so at the outset created a potential disconnect with the wider empirical base.

In response, we have revised the introduction to open with a more direct and accurate definition of the illusory truth effect. Only after establishing this foundation do we introduce misinformation and fake news as important, real-world applications of the effect. We believe

this revised structure more clearly aligns the introduction with the content and scope of the review. The revised introduction now reads:

- Lines (15-34): “The illusory truth effect, or repetition-induced truth effect is refers to the tendency for individuals to judge repeated statements as more true than novel statements¹. In a typical experiment, participants are first exposed to a set of ambiguous statements (i.e., statements for which participants are unlikely to know the actual truth status, as established by pretests) or neutral statements (i.e., statements equally likely to be rated as true in both their true and false version) during the exposure phase. At the test phase, participants are instructed to judge the truth of another set of statements, usually consisting of statements repeated from the exposure phase and new statements previously unseen in a random order. The illusory truth effect is typically operationalized as the difference in truth ratings between repeated and novel items. While the design of primary studies varies considerably, the illusory truth effect appears robust across populations and types of content^{2,3}. The effect is observed regardless of the of the population type^{4,5} (clinical vs typical), or of age^{6,7} (from 5 to 80 years old). Additionally, the illusory truth effect seems to appear regardless of the source of the information⁸, or whether it is a trivia statement^{9,10}, a general knowledge question^{7,11}, a (fake) news headline^{12,13}, or less-tangible information such as a rumor¹⁴, an opinion in an argument¹⁵, or a marketing claim¹⁶. However, the effect of repetition on perceived truth was not consistently observed. For instance, repetition did not reliably increase the perceived truth for extremely implausible statements^{12,17}, or headlines in form of questions¹⁸.
- Lines (35-43): “The rise of digital media and the rapid pace of online communication have amplified concerns about how repetition might increase belief in misinformation¹⁹. Repeated exposure to false news headlines not only increases perceived accuracy but can also raise the intention to share such content on social media²⁰. While individual factors such as motivation, cognitive abilities, thinking style, or knowledge influence belief in misinformation^{21,22}, these factors appear to have little impact on the illusory truth effect. The effect is robust across individual differences in cognitive abilities¹⁰ and persists even when participants are incentivized for accuracy²³, or possess relevant factual knowledge^{7,24}.”
- Lines (44-57): “Several intervention strategies have been tested to reduce or eliminate the illusory truth effect. These include warning participants about the effect itself^{25,26},

alerting them prior to initial exposure that some of the information might be false ²⁷, or asking them to rate the accuracy of information during initial exposure ^{11,28}. However, other approaches have produced mixed results. While labeling false information as “disputed by a third-party fact-checker” had little impact, more explicit labels (e.g., “false”) successfully eliminated the effect ²⁹. Similarly, the benefit of labeling source reliability may depend on whether participants recall this information during judgment ^{8,30,31}. Given that false information can spread more widely and rapidly than true information online ³², understanding the determinants of the illusory truth effect has become increasingly urgent. Such insights could clarify the conditions under which repetition amplifies perceived truth, thereby guiding the design of targeted interventions to reduce the risk of increasing gullibility to misinformation when people are massively exposed repeatedly to them.”

14. **Comment:** p. 3 lines 29-32 – I know the wording of this sentence was updated based on another reviewer’s comments, but I’m now having trouble following it. As currently written, this sentence could be read that in some studies there is no test phase? Or that in some studies, participants aren’t instructed this?

Response: We thank the reviewer for highlighting this unintended ambiguity. Our previous revision may have implied that some studies omit a test phase or omit truth judgments, which was not our intention. To clarify, we have rewritten the passage to describe the structure of a typical illusory truth effect experiment and to make clear that both exposure and test phases are standard. The revised sentence now reads:

- Lines (16-23): “In a typical experiment, participants are first exposed to a set of ambiguous statements (i.e., statements for which participants are unlikely to know the actual truth status, as established by pretests) or neutral statements (i.e., statements equally likely to be rated as true in both their true and false version) during the exposure phase. At the test phase, participants are instructed to judge the truth of another set of statements, usually consisting of statements repeated from the exposure phase and new statements previously unseen in a random order.”

15. **Comment:** p. 12 – lines 241 – The authors first refer to one of the categories of task at exposure as “passive processing” and then later in the paragraph refer to (what I believe to

be the same category) as “instructed to read the statements”. From the methods section, it seems like passive processing included other tasks like listening to the statements, so updating “instructed to read the statements” to “passive processing” seems more accurate here

Response: We agree that our prior wording was inconsistent and potentially confusing, as "passive processing" was indeed a broader category that included tasks beyond reading (e.g., no instructions, listening, head movements). We have revised the sentence accordingly to consistently use the term "passive processing" throughout the paragraph. The revised paragraph now reads:

- Lines (244-255): “**Participants’ task during exposure phase.** In primary studies, the task participants perform during initial exposure to information varies widely, from passively reading statements, to making unrelated judgments (e.g., rating interestingness, categorizing), to explicitly evaluating truthfulness. A significant moderating effect was found for the content of the task used during the exposure phase. When participants were asked for a truth judgment during the exposure phase ($g = 0.10$, 95% CI [-0.03, 0.23]; observed $u = 78$), the illusory truth effect was consistently reduced compared to irrelevant tasks ($g = 0.42$, 95% CI [0.34, 0.50]; observed $u = 213$), $b = -0.32$, 95% CI [-0.47, -0.16], $t(264) = 4.02$, $p < .001$, or passive processing ($g = 0.47$, 95% CI [0.34, 0.60]; observed $u = 73$), $b = -0.46$, 95% CI [-0.70, -0.23], $t(200) = 3.94$, $p < .001$. However, the illusory truth effect did not significantly vary when the exposure task involved passive processing compared to performing an irrelevant task, $t(195) = 1.69$, $p = .093$.”

16. **Comment:** p. 13 – line 269-270 – What is the condition being compared to the delay of over a week here? The delay of less than a day?

Response: We thank the reviewer for this observation. We agree that the comparison involving the “delay over a week” condition was unclear. The intended comparison was with the “within a day” delay condition. We have now revised the text to make this comparison explicit as follows.

- Lines (303-304): “However, no significant difference was found when comparing studies with a same day delay to those with a delay over a week.”

17. Comment: p. 14 – lines 304-306- this statement – “ Similarly, the proportion of true or false statements showed no influence on the effect size ($t(245) < 1$), indicating that the illusory truth effect is comparable in studies that uses mixed, all-true, or all-false item pools.”

Given this was a continuous moderator, would it be more appropriate to report that the illusory truth effect is similar across different proportions of true vs. false claims? Before reading the methods, I thought these were three distinct categories being directly compared to each other rather than a continuous measure.

Response: We thank the reviewer for this observation. We agree that our original phrasing could have incorrectly suggested a categorical comparison across three item pool types. That was not an accurate reflection of our analytic approach, which treated the proportion of true versus false statements as a continuous moderator. We have revised the sentence to more precisely reflect this by stating that “the illusory truth effect did not vary systematically with the proportion of true or false items,” thereby avoiding the implication of categorical comparisons as follows:

- Lines (365-368): “Similarly, the proportion of true or false statements showed no influence on the effect size ($b = -0.02$, 95% CI $[-0.39, 0.35]$; observed $u = 339$), $t(175) < 1$, indicating that the illusory truth effect is similar across different proportions of true vs. false statements.”

18. Comment: I had trouble following the discussion starting on p. 15 line 330. Particularly, I’m not seeing how these findings necessarily challenge the assumption of the referential theory – elaborating on whether or not something is true and elaborating on a statement in some other way with the assumption that it is true should each result in the formation of different links/ strengths of links between referents in memory. The referential theory is not just about deeper processing leading to stronger memory traces, but also whether that processing results in an excitatory or inhibitory link between referents.

Response: We thank the Reviewer for this helpful clarification regarding the scope of the referential theory. We agree that our original discussion did not adequately reflect that the theory addresses not only the strength of links between referents in memory but also their valence (excitatory vs. inhibitory), depending on how the information is processed. We have revised the manuscript as follows:

- Lines (431-439): “From a referential theory perspective, truth judgments may create links between referents in memory whose valence (e.g., excitatory vs. inhibitory) depends on whether the information is encoded as true or false. In contrast, in irrelevant-task or passive-processing conditions, statements may be implicitly assumed to be true, leading to predominantly excitatory links. Furthermore, individual retrieval strategies may also influence the illusory truth effect: some participants may attempt to retrieve their previous judgment stored in memory, whereas others may primarily rely on fluency or adopt a different strategy.”

19. **Comment:** p. 20 lines 447-449– on this statement - “This finding suggests that factors such as experimenter presence, environmental control, or potential distractions in online settings do not substantially influence cognitive processes involved in the illusory truth effect.” – while there may have been no differences observed in the aggregate, as these different potential factors weren’t looked at separately, I would be careful making this conclusion, especially given evidence regarding the impact of divided attention on the size of the truth effect (e.g., - Ly, Bernstein, & Newman, 2024). It could be that different factors influenced participants in different settings in different ways, but these washed out in the overall comparison.

Response: We thank the Reviewer for pointing out this important point. We fully agree that our study does not allow us to draw firm conclusions about the role of specific contextual factors (e.g., experimenter presence or distractions). Although our findings suggest general consistency of the illusory truth effect across lab and online settings, we acknowledge that the aggregate-level comparison may obscure the influence of distinct factors that differ between these environments (e.g., divided attention). To reflect this important nuance and avoid overgeneralization, we have revised the manuscript as follows:

- Lines (489-494): “Null effects for moderators such as the modality of presentation, the response scale or testing environment were consistent with earlier meta-analytic findings ², suggesting that the illusory truth effect extends across commonly used research contexts. However, recent evidence suggests that divided attention (e.g., experimenter presence, environmental distractions) may still influence the illusory truth effect and warrant closer examination in future research ⁶¹.”

20. Comment: p 32, lines 734-736 “When not specified, we assumed that the presentation time was similar except if the modality of presentation changed between the exposure and test phases.” – I’m wary about the accuracy of this assumption given it would result in a mis-categorization of my own studies. I think many truth effect studies present claims for a certain amount of time at exposure and specify this, but then at test let participants respond to them in a self-paced way. However, especially if participants were simply reading claims and not making time-restricted judgments at exposure, papers may not explicitly say these test judgments are self-paced because it would be implied that this is the case if no time restrictions are mentioned at any point for judgments.

Response: We thank the Reviewer for this observation. We agree that assuming identical presentation durations between exposure and test phases could lead to misclassification. This is especially plausible in studies where exposure is time-limited, but the test phase allows for self-paced judgments without clearly stating so.

To address this concern, we re-examined all studies in our dataset that were originally coded as having matched, time-limited durations for both exposure and test. We identified 81 effect sizes fitting this criterion, of which 49 did not provide explicit information regarding test phase presentation time.

To assess whether reclassifying these ambiguous cases would affect our findings, we conducted a sensitivity analysis by recoding the 49 effect sizes as “participant-paced.” The results of this analysis were consistent with those from our original model, suggesting that the overall moderating effect of test-phase presentation time is robust to this reclassification.

The output of this sensitivity analysis remained consistent with the original dataset. However, we think that the Reviewer’s reasoning is totally valid. Therefore, we conserved the assumption that judgments are participant-paced in the test phase if unspecified in our main analysis model.

Nevertheless, we acknowledge the Reviewer's concern and believe that it is more appropriate to adopt a conservative and theoretically grounded assumption. Therefore, in our revised main analysis, we now assume that presentation time at test is participant-paced when not explicitly stated. We have updated the Methods section accordingly and now report the sensitivity analysis in the Supplementary Information. The revised methods now reads:

- Lines (780-781): "When not specified, we assumed that the presentation time was participant-paced."
-

Reviewer 3 Comments

Overall Comment: In this resubmission, the authors present a manuscript that differs substantially from the previous version. They have not only expanded the set of studies analyzed but also addressed moderators that are highly relevant for readers seeking to understand the phenomenon(s) under investigation. Additionally, there is a noticeable improvement in the transparency of the procedures, which facilitates a more critical evaluation of the results. However, my main concern with this version is that it still lacks a clear and explicit statement of the article's objectives, as well as a more coherent theoretical integration.

Response: We thank the Reviewer for his/her comment and suggestions for improving our manuscript.

- 1. Comment:** I could not find a paragraph that clearly articulates the aims of the paper. In the paragraph beginning on line 88, the authors frame their critique of the previous meta-analysis as an "implicit objective," suggesting their goal is to "do it better." In my view, this framing is unnecessary. The earlier meta-analysis is simply outdated, and that alone provides sufficient justification for a new one. The intention to improve upon prior work does not need to rely on discrediting it.

Response: We agree that a forward-looking framing is more appropriate and appreciate the opportunity to improve the tone and clarity of our introduction. In response, we have revised the paragraph to remove language implying that our work is intended as a corrective to earlier research. The updated text now emphasizes the rationale for a new meta-analysis based on the field's expansion and methodological diversity, without discrediting prior work. The revised paragraph now reads:

- Lines (92-98): "This first meta-analysis on the illusory truth effect provided a valuable foundation by synthesizing early evidence for the phenomenon. Since then, the field has expanded rapidly, with increasing diversity in study designs, populations, and materials, warranting an updated and more comprehensive synthesis. At the same time, concerns have been raised about the evidential quality of the literature: a recent review found that 96% of studies on the illusory truth effect reported statistically significant results, raising the possibility of publication bias ³."

2. **Comment:** To address this issue, I suggest that the authors insert a paragraph between lines 101 and 102 explicitly stating their objectives. These could include (if authors agree): a) updating the meta-analysis with more recent data; b) addressing specific limitations of the previous meta-analysis; c) clarifying the role of moderators relevant to different theoretical perspectives on the effect; d) incorporating an approach with practical relevance, particularly regarding "fake news" and strategies for its mitigation.

Response: We appreciate the Reviewer's suggestion regarding the articulation of our objectives. We agree that clearly articulating our objectives strengthens the coherence and accessibility of the manuscript. In response, we have added a new section to the introduction that explicitly outlines four core aims of our meta-analysis, aligning closely with the structure suggested by the Reviewer.

We also took this opportunity to clarify the rationale behind our moderator selection, which reflects the applied orientation of the project. As the literature has grown increasingly heterogeneous in study design, populations, and materials (e.g., Henderson et al., 2022), we saw an opportunity to identify boundary conditions and practical implications. Accordingly, many of our moderators were chosen based on their real-world relevance and empirical prevalence, with a subset also aligned with established theoretical frameworks.

While we slightly revised the Reviewer's suggested phrasing of objective (c), this was done to better reflect our actual scope (e.g., focusing on generalizability and boundary conditions) without diminishing the importance of theory. We also clarified that we primarily interpret the results within the processing fluency account and the referential theory of the illusory truth effect, while also considering alternative explanations for moderators not directly addressed by these frameworks

By anchoring our discussion in established theoretical frameworks, we aim to offer a rigorous and practically relevant synthesis of the current evidence, while providing a strong empirical basis for refining or expanding theory in future work. The revised paragraph in the introduction reads:

- Lines (99-116): "The present meta-analysis was designed to address these developments through four core objectives. First, to provide an updated and comprehensive estimate of the illusory truth effect by synthesizing the greatly expanded body of research with 182 studies and 366 effect sizes published between 1977 and 2025. Second, to improve the estimation of the effect size of the illusory truth effect and the potential moderators of such effect by applying more advanced techniques than the ones available when the

previous meta-analysis was conducted. These include three-level random-effects models to account for the non-independence of multiple estimates from the same sample^{40,41}, publication bias diagnostics ⁴², risk of bias assessments (RoB2 ⁴³), multivariate imputation for missing moderator data, and sensitivity analyses to evaluate the impact of missing outcome data. Third, to examine a broader set of moderators to investigate the boundary conditions of the illusory truth effect (see Table 1). Moderator selection prioritized variables with practical relevance (e.g., misinformation susceptibility) and empirical prevalence. The results are discussed within the processing fluency account and the referential theory of the truth effect, while also considering alternative explanations for moderators not directly addressed by these frameworks. Fourth, drawing from this moderator analysis, to identify those with direct implications for misinformation mitigation, including the design of effective interventions.”

3. Comment: I note that the Discussion section opens with and places particular emphasis on the final objective listed above: fake news" and strategies for its mitigation. If this is indeed intended as the article's primary focus, then the Introduction should be revised to make this emphasis more explicit—guiding the reader to understand the relevance of the illusion of truth effect to the topic of fake news, while also informing them about other contributing factors that influence its success and the strategies available to mitigate it. Alternatively, if the authors prefer to retain the current structure of the Introduction, I recommend that the Discussion be expanded to address all the stated objectives, ensuring alignment between the framing and the conclusions.

Response: We thank the Reviewer for this observation. We agree that our primary applied focus aligns most strongly with objective (d): using moderator patterns to inform intervention strategies in real-world settings. We also agree that the relevance of the illusory truth effect to the problem of misinformation should be emphasized earlier in the manuscript to better prepare the reader for our Discussion. In response, we revised the Introduction to explicitly connect the illusory truth effect to repeated misinformation, outline other factors that influence belief in misinformation, and provide an overview of intervention strategies. The revised introduction now reads:

- Lines (35-43): “The rise of digital media and the rapid pace of online communication have amplified concerns about how repetition might increase belief in misinformation ¹⁹. Repeated exposure to false news headlines not only increases perceived accuracy but

can also raise the intention to share such content on social media ²⁰. While individual factors such as motivation, cognitive abilities, thinking style, or knowledge influence belief in misinformation ^{21,22}, these factors appear to have little impact on the illusory truth effect. The effect is robust across individual differences in cognitive abilities ¹⁰ and persists even when participants are incentivized for accuracy ²³, or possess relevant factual knowledge ^{7,24}.”

- Lines (44-57): “Several intervention strategies have been tested to reduce or eliminate the illusory truth effect. These include warning participants about the effect itself ^{25,26}, alerting them prior to initial exposure that some of the information might be false ²⁷, or asking them to rate the accuracy of information during initial exposure ^{11,28}. However, other approaches have produced mixed results. While labeling false information as “disputed by a third-party fact-checker” had little impact, more explicit labels (e.g., “false”) successfully eliminated the effect ²⁹. Similarly, the benefit of labeling source reliability may depend on whether participants recall this information during judgment ^{8,30,31}. Given that false information can spread more widely and rapidly than true information online ³², understanding the determinants of the illusory truth effect has become increasingly urgent. Such insights could clarify the conditions under which repetition amplifies perceived truth, thereby guiding the design of targeted interventions to reduce the risk of increasing gullibility to misinformation when people are massively exposed repeatedly to them.”

4. **Comment:** I believe the Discussion section would benefit from better organization, supported by a clearer conceptual framing and a more cohesive argument—both of which should be grounded in a well-defined set of objectives (possibly those I have suggested)

Response: We appreciate the Reviewer’s suggestion regarding the organization and conceptual framing of the Discussion. We agree that a clearer structure, explicitly linked to our stated objectives, would enhance the coherence of our argument. In response, we have reorganized the Discussion to align with the four core objectives introduced in the Introduction. The revised manuscript now: (a) begins by summarizing the updated effect size estimate, (b) highlights methodological improvements and their implications for effect size precision, (c) discusses moderator patterns and their theoretical implications, and (d) expands on the applied relevance for misinformation mitigation, before concluding with the limitations of the current meta-

analysis and directions for future research. With the effort to align with our third objective (i.e., examine a broader set of moderators to investigate the boundary conditions of the illusory truth effect), we also have revised the structure of how moderating effects are interpreted by regrouping four categories of moderators in their relation to: stimulus characteristics (e.g., stimulus type, type of repetition, factual truth status, and proportion of true items), exposure phase (e.g., participants' task, presentation time, veracity cues, instructional warnings), test phase (e.g., judgment time, instructional warnings), and study design (e.g., delay between phases, manipulation design of repetition, modality of presentation, response scale, testing environment). The revised manuscript now reads:

- Lines (381-388): “This systematic review and three-level meta-analysis provides robust evidence that repetition increases the perceived truth of information, with growing relevance in today’s media landscape. Synthesizing evidence from 182 studies (366 effect sizes; $N = 31,184$), our analysis provides robust confirmation of the illusory truth effect, indicating a small effect size ($g = 0.37$, 95% CI [0.30, 0.44]), after correcting for publication bias. These findings suggest that repetition reliably increases perceived truth across diverse methodological contexts, highlighting the relevance for contemporary concerns regarding misinformation.”
- Lines (389-398): “Our methodological approach improved upon the previous meta-analysis by incorporating more recent statistical techniques, including a three-level random-effects model to account for within-study dependence of effect sizes, multivariate imputation to handle missing moderator data, and sensitivity analyses to evaluate the impact of missing outcome data. We also assessed study-level quality using the Risk of Bias 2 tool, which indicated that most studies raised some concerns primarily due to a lack of pre-registration. However, effect sizes did not differ significantly by overall risk of bias level, suggesting that the pooled estimate was not substantially biased by study quality. In addition, while publication bias diagnostics revealed evidence of small-study effects, the illusory truth effect remained robust after applying appropriate correction methods.”
- Lines (399-404): “Beyond the overall effect, substantial between-study heterogeneity in effect sizes warranted moderator analyses to clarify the conditions under which repetition influences the perceived truth. To structure this discussion, we group moderators by stimulus characteristics, exposure phase, test phase, and study design. Averaged across multiple imputed datasets, the moderator set explained approximately

37% of the observed variance. This indicates meaningful heterogeneity beyond sampling error.”

- Lines (405-421): “Regarding stimulus characteristics, the illusory truth effect was robust across content types (e.g., trivia statements, news headlines, opinions), repetition formats (verbatim vs. gist), factual truth status (true vs. false), or proportion of true items. The absence of moderation by truth status aligns with prior findings and reinforces the idea that fluency affects perceived truth independently of objective accuracy ^{30,49-51}. Similarly, the comparable effects observed for verbatim and gist repetitions are consistent with both the fluency account and referential theory. While the former emphasizes the role of conceptual (vs. perceptual) processing ease ⁵², the latter attributes the effect to stronger semantic coherence across repetitions ^{38,39}. We also found that studies using news headlines tend to yield smaller effect sizes than those using standard trivia statements. One possible explanation is that many headline-based studies assessed the illusory truth effect in the context of social media misinformation, where items were not systematically chosen for neutrality or ambiguity. Moreover, a recent meta-analysis ⁵³ found that participants were generally able to distinguish real from fake news. One possibility is that the format of social media content may cue readers to engage in more cautious evaluation. This could reduce reliance on fluency during truth judgments, thereby attenuating the illusory truth effect.”
- Lines (422-439): “In contrast, several moderators related to the exposure phase significantly influenced the magnitude of the illusory truth effect. Asking participants to perform a truth judgment in the exposure phase was associated with a weaker illusory truth effect compared to irrelevant tasks or passive processing. One explanation, consistent with the fluency discount hypothesis, is that individuals prompted with an initial truth judgment may activate relevant knowledge, thereby reducing reliance on fluency as a cue of truth. This attenuation appears to be limited in time, emerging primarily when the test phase follows shortly after exposure and dissipating after approximately two days ^{11,54}. Importantly, this pattern remains after controlling for participants’ memory of the statements, suggesting that factors beyond simple memory retention are involved. From a referential theory perspective, truth judgments may form links between referents in memory that differ in valence (e.g., excitatory vs. inhibitory) depending on whether information is encoded as true or false. In contrast, statements may be implicitly assumed true in irrelevant task or passive processing conditions, leading to uniformly excitatory links.”

- Lines (440-451): “Additionally, longer presentation time at the exposure phase (more than 5 seconds) was associated with a larger truth effect, although this relationship was only marginal in one of the sensitivity analyses. To our knowledge, no prior work has systematically isolated the exposure phase to examine how stimulus duration influences the illusory truth effect. This potential relationship has thus not been reported previously. From a processing fluency perspective, longer exposure may facilitate more complete perceptual encoding, making the statement easier to process at test and thereby more likely to be judged as true. From a referential theory perspective, extended exposure may provide more opportunity to activate and integrate related concepts in memory, increasing the coherence of the referential network associated with the statement. Future work should systematically investigate whether limiting presentation time during initial exposure could effectively attenuate the illusory truth effect.”
- Lines (452-465): “Crucially, our findings suggested that the valence of veracity cues significantly moderate the illusory truth effect. False cues reduced the impact of repeated exposure on perceived truth while true cues did not affect the illusory truth effect. This finding provides additional support for the value of debunking interventions. To explore how variability in the type of veracity cue (e.g., source reliability cues, epistemic qualifiers, labels, immediate or delayed feedback) contribute to inconsistent findings in the literature, we conducted an exploratory analysis. We found that the moderating influence of veracity cues may depend on cue type, with stronger effects of feedback, source reliability cues, and epistemic qualifiers compared to labels. However, the reliability of these conclusions is limited by the small number of effect sizes per category, and they should thus be interpreted with caution (see Supplementary Materials). Understanding these mechanisms is crucial for developing effective interventions to mitigate the spread of misinformation ¹⁹. Future studies should systematically investigate how different types of veracity cues influence the illusory truth effect.”
- Lines (466-473): “Surprisingly, we found no effect of explicit instructional warnings at the exposure phase on the illusory truth effect. This contrasts with Jalbert et al.’s findings ²⁷, which showed that explicit warnings reduced the illusory truth effect, but only when given during exposure alongside reading instructions. A possible explanation is that implicit warnings (e.g., initial truth ratings) and explicit warnings may overlap. Once participants are instructed to evaluate veracity, an additional instructional warning

may provide only few additional information, which makes its impact difficult to detect at the aggregate level.”

- Lines (474-494): “In addition, we found little evidence that variables in the test phase (e.g., warnings, presentation time) reliably moderated the illusory truth effect. These null results align with prior work showing that time pressure or warnings during the test phase do not alter the effect ^{27,55}, suggesting that repetition may exert its strongest influence during initial exposure rather than later evaluation. Finally, the effect was generally consistent across study design features (e.g., delay length, repetition design, modality of presentation, response scale, testing environment). Although some primary studies have reported smaller effects in between-participant designs⁵⁶⁻⁵⁸, these patterns did not replicate in our aggregate analyses and were likely constrained by the limited number of studies in some categories (e.g., only five effect sizes extracted from between-participant designs). Similarly, we observed a relatively smaller truth effect when the test phase followed exposure within the same day versus within the same week, although sensitivity analyses were inconsistent. Primary studies suggested that the moderating effect of delay may interact with other variables influencing the illusory truth effect during the exposure phase, such as the type of task ⁵⁹ or the type of repetition ⁶⁰. As such, it may be difficult to isolate the sheer effect of delay in aggregate data. Null effects for moderators such as the modality of presentation, the response scale or testing environment were consistent with earlier meta-analytic findings ², suggesting that the illusory truth effect extends across commonly used research contexts. However, recent evidence suggests that divided attention (e.g., experimenter presence, environmental distractions) may still influence the illusory truth effect and warrant closer examination in future research ⁶¹.”
- Lines (495-510): “Synthesizing across moderators, the results indicate that illusory truth effect may be shaped primarily by processes occurring during the initial encounter with information, and several moderators point to potential leverage for interventions. Performing an initial truth judgment was associated with a weaker illusory truth effect, suggesting that accuracy-focused evaluation at first exposure may help to attenuate the influence of repetition. This interpretation is consistent with intervention studies showing that accuracy prompts and early veracity judgments can reduce susceptibility to misinformation ^{11,28}. Similarly, the presence of falsity cues was reliably linked to a reduced effect, pointing to the potential utility of clear validity signals during initial encounter with potential misinformation ⁵¹. In contrast, longer exposure durations

tended to amplify the effect, raising the possibility that stimulus timing may be an overlooked target for intervention, though this relationship warrants further investigation.”

- Lines (507-516): “Together, these findings indicate that interventions may be most effective when they encourage accuracy-focused processing and make falsity salient at the point of first exposure. Promising approaches include metacognitive prompts ²⁶, encouraging early reflection with accuracy checks ²⁸ and debunking strategies ⁶². Importantly, the concern that repeating false claims during debunking might backfire appears less warranted than previously feared; current evidence suggests that corrective interventions can be effective without amplifying beliefs in the exposed claims^{63,64}. At the same time, the illusory truth effect captures only one mechanism among many that contribute to the belief and spread of misinformation. While it cannot solve misinformation alone, reducing the illusory truth effect is a practical way to limit its impact.”
- Lines (564-571): “In conclusion, this updated meta-analysis confirms the robustness of the illusory truth effect, with a small but consistent overall effect ($g = 0.37$) after correcting for publication bias. While repetition reliably increases perceived truth, the effect varies substantially across study designs. Our findings highlight the central role of initial exposure in shaping later truth judgments, with robust moderators including the type of item, participants’ task during the exposure phase, and the valence of veracity cues. These insights have direct practical relevance for developing scalable interventions to limit the impact of repeated misinformation.”

5. **Comment:** I am also concerned about the somewhat assertive tone in the Discussion, where certain hypothetical or interpretative points are presented as established facts. While the authors provide their interpretation of the data, they should also acknowledge the possibility of alternative explanations. The Discussion would gain in relevance and scholarly value if the findings were more explicitly connected to theoretical implications—for example, by raising questions about whether the data support or challenge specific frameworks—rather than presenting conclusions without sufficient nuance.

Response: We thank the Reviewer for this important observation regarding tone and interpretative balance in the Discussion. We agree that interpretative points should be presented

with appropriate nuance and in the context of alternative explanations. In the revision of the organization of the Discussion, we were careful not to present our interpretation as facts. We also grounded our discussion of findings supporting or challenging specific frameworks or primary research. Please see for an instance of the revision implemented in the revised Discussion:

- Lines (407-413): “The absence of moderation by truth status aligns with prior findings and reinforces the idea that fluency affects perceived truth independently of objective accuracy ^{30,49–51}. Similarly, the comparable effects observed for verbatim and gist repetitions are consistent with both the fluency account and referential theory. While the former emphasizes the role of conceptual (vs. perceptual) processing ease ⁵², the latter attributes the effect to stronger semantic coherence across repetitions ^{38,39}.”
- Lines (415-421): “One possible explanation is that many headline-based studies assessed the illusory truth effect in the context of social media misinformation, where items were not systematically chosen for neutrality or ambiguity. Moreover, a recent meta-analysis ⁵³ found that participants were generally able to distinguish real from fake news. One possibility is that the format of social media content may cue readers to engage in more cautious evaluation. This could reduce reliance on fluency during truth judgments, thereby attenuating the illusory truth effect.”
- Lines (423-439): “Asking participants to perform a truth judgment in the exposure phase was associated with a weaker illusory truth effect compared to irrelevant tasks or passive processing. One explanation, consistent with the fluency discount hypothesis, is that individuals prompted with an initial truth judgment may activate relevant knowledge, thereby reducing reliance on fluency as a cue of truth. This attenuation appears to be limited in time, emerging primarily when the test phase follows shortly after exposure and dissipating after approximately two days ^{11,54}. Importantly, this pattern remains after controlling for participants’ memory of the statements, suggesting that factors beyond simple memory retention are involved. From a referential theory perspective, truth judgments may form links between referents in memory that differ in valence (e.g., excitatory vs. inhibitory) depending on whether information is encoded as true or false. In contrast, statements may be implicitly assumed true in irrelevant task or passive processing conditions, leading to uniformly excitatory links. Furthermore, individual retrieval strategies may also influence the illusory truth effect: some participants may

attempt to retrieve their previous judgment stored in memory, whereas others may primarily rely on fluency or adopt a different strategy.”

- Lines (445-449): “From a processing fluency perspective, longer exposure may facilitate more complete perceptual encoding, making the statement easier to process at test and thereby more likely to be judged as true. From a referential theory perspective, extended exposure may provide more opportunity to activate and integrate related concepts in memory, increasing the coherence of the referential network associated with the statement.”

6. Comment: It is clear for me that I believe this article should ultimately be published, as it addresses a relevant topic and is likely to make a meaningful contribution to the field, particularly given its nature as a meta-analysis. However, precisely because of its potential impact, it is essential that the article presents a more coherent set of objectives and that the authors attend to other important details that remain insufficiently clear in the current version. It is a fact that throughout the article the authors have corrected erroneous statements made about the studies presented or about the field, which has greatly improved this version of the manuscript. Some of these errors escaped me during the first review but were pointed out by my fellow reviewers. I understand that there are many articles involved, and the lack of a conceptual framework for them—one for which methodological and conceptual details of the studies are relevant—causes the authors to overlook the importance of some imprecise statements. This concerns me because such imprecisions often become entrenched misunderstandings of phenomena that can take years to correct, once a respected figure in the field decides to analyze the original sources. Writing a review article thus carries great responsibility.

Response: We sincerely thank the Reviewer for emphasizing the importance of precision and clarity in review work. We fully agree that review articles carry a special responsibility to represent the primary literature accurately, and that even small inaccuracies can perpetuate misunderstandings in the field. In response to the Reviewer’s concerns, we have thoroughly revised the manuscript to:

- Clarify the objectives by adding an explicit paragraph in the Introduction presenting four aims: updating the pooled effect size, implementing methodological improvements,

examining a broader set of moderators, and discussing applied implications for misinformation.

- Strengthen the conceptual framework by introducing Table 1, which links each moderator to its conceptual rationale.
- Align the Discussion with the objectives. The Discussion now (a) opens with the updated pooled estimate and its context, (b) interprets methodological improvements and bias diagnostics, (c) synthesizes moderator patterns with theoretical implications, and (d) expands on applied relevance for misinformation mitigation, followed by limitations and future directions.
- Calibrate the interpretative tone by avoiding assertive phrasing, explicitly acknowledging alternative explanations, and noting where aggregate patterns may mask contextual differences.
- Improve conceptual accuracy by defining moderators, standardizing terminology across the manuscript, and re-examining cited theoretical sources to ensure our interpretations do not extend beyond what is supported in the original literature.

7. **Comment:** Below I list some details I have identified, though I fear this list is not exhaustive given the extensive number of articles reviewed: 1. The first example appears in the change made at lines 22–24, where the authors attribute the term “illusory truth effect” to Pennycook et al. (2018). The term “illusion of truth” already appears in the original articles by Bacon and subsequent papers by Begg et al., and Garcia-Marques et al. later shortened to “truth effect” in publications by Unkelbach. Its relationship to “fake news” has a different history, with pragmatic rather than the original theoretical goals (which aimed at understanding the human mind and not why its sensitive to fake news). Thus, the phrase “Repeated exposure to fake news can increase individuals’ perception of its accuracy, a phenomenon known as the ‘illusory truth effect’” would be more accurate if replaced by “Repeated exposure to fake news can increase individuals’ perception of its accuracy, a phenomenon theoretically related to the ‘illusory truth effect’ (Bacon, Begg et al.)”

Response: We thank the Reviewer for this observation and apologize for any confusion caused by the original phrasing. Our intent was not to attribute the term “illusory truth effect” to Pennycook et al. (2018). Rather, we cited their study as an example of how this effect manifests

in the context of misinformation and fake news. The term itself, as the Reviewer rightly notes, has a longer history tracing back to early work by Bacon, Begg, Hasher et al.

To ensure full clarity and avoid potential misinterpretation, we have revised the Introduction to explicitly cite the foundational work by Hasher et al. (1977), which first demonstrated that repeated statements are more likely to be judged as true than novel ones. We now introduce the illusory truth effect in this original theoretical context, with no attribution of terminology to later studies as follows:

- Lines (15-16): “The illusory truth effect, or repetition-induced truth effect refers to the tendency for individuals to judge repeated statements as more true than novel statements
1”

8. Comment: 2. The authors state that studies on truth effects use “ambiguous statements” (referring to people’s ratings tending toward the midpoint of the scale). I understand this usage, but in fact, those studies frequently use “neutral statements,” which are statements equally likely to be rated as true in both their true and false versions. Given the potential impact of this paper, I believe this distinction should be emphasized, as it may explain why studies sometimes show inconsistent results. One possibility is to state that studies use either ambiguous or neutral statements, with appropriate references.

Response: We thank the Reviewer for highlighting this important distinction. Our wording was based on the description provided in the previous meta-analysis by Dechêne et al. (2010), which characterized a typical illusory truth effect experiment as involving “a set of ambiguous statements (i.e., statements whose truth status is unknown to participants, as established by pretests)”.

However, we agree that the term “neutral statements” may be more precise in many cases, particularly where stimuli are designed to be equally likely rated as true regardless of their actual truth status. Accordingly, we have revised the relevant passage to specify that studies typically use either “ambiguous” or “neutral” statements as follows:

- Lines (16-20): “In a typical experiment, participants are first exposed to a set of ambiguous statements (i.e., statements for which participants are unlikely to know the actual truth status, as established by pretests) or neutral statements (i.e., statements

equally likely to be rated as true in both their true and false version) during the exposure phase.”

9. **Comment:** 3. The statement “the illusory truth effect is robust across various populations and tasks” would be better supported by citing Dechêne et al.’s meta-analysis, in addition to the specific studies referenced.

Response: We thank the Reviewer for this suggestion. We agree that citing Dechêne et al.’s (2010) meta-analysis provides important support for the claim that the illusory truth effect is robust across different populations and task types. Their work offers foundational evidence based on the literature available at the time and complements the more recent studies we had initially cited. In response, we have now added a citation to Dechêne et al. alongside the existing references to strengthen the empirical basis for this statement as follows:

- Lines (25-26): While the design of primary studies varies considerably, the illusory truth effect appears robust across populations, tasks, and types of content ^{2,3}.

10. **Comment:** 4. Line 45: The sentence “However, repetition does not always increase perceived truth for extremely implausible statements or headlines in the form of questions” suggests this applies only to those cases. Consider revising to: “This is the case, for instance, for extremely implausible statements or headlines in the form of questions.”

Response: We thank the Reviewer for this suggestion. We agree that the original phrasing could be misread as implying that these are the only exceptions to the illusory truth effect, whereas they are better understood as illustrative examples. In response, we have revised the relevant passage as follows:

- Lines (31-34): “However, the effect of repetition on perceived truth was not consistently observed. For instance, repetition did not reliably increase the perceived truth for extremely implausible statements ^{12,17}, or headlines in form of questions ¹⁸.”

11. **Comment:** 5. Line 48: The sentence “The dominant explanation of the illusory truth effect is the fluency account” would be more theoretically accurate if stated as: “Dominant explanations of the illusory truth effect attribute a significant role to processing fluency.”

This is because in line 51, the process is described as a misattribution or association with valid cues, which is defended by those who explain the truth effect as involving a memory referent. Thus, although all current theories recognize the role of processing fluency, they differ in how fluency is integrated or interpreted by the cognitive system (as a direct cue, a learned association with validity, an affective experience, a byproduct of familiarity, etc.).

Response: We thank the Reviewer for this suggestion. We agree that the original phrasing oversimplified the theoretical landscape and did not fully capture the diversity of perspectives on the illusory truth effect. As the Reviewer notes, while processing fluency plays a central role across most contemporary accounts, theories vary in how fluency is integrated. We have revised the sentence to reflect this nuance more accurately.

- Lines (58-61): “Previous research has proposed several theoretical frameworks to explain why repetition increases perceived truth of information ³³. Dominant explanations of the illusory truth effect attribute a significant role to processing fluency, which is defined as the perceived ease with which information is processed ³⁴.”

12. **Comment:** 6. Lines 60–63: The hypothesis raised by the authors is plausible. However, please note that there are no data supporting this directly, whereas some data suggest there may be a dissociation between conceptual and perceptual fluency, not merely arising from the strength of the fluency experience. I recommend making this clear, and also explaining why you consider the dissociation an artifact (see for instance Garcia-Marques et al. 2017).

Response: We thank the Reviewer for this important theoretical clarification. We agree that empirical work, including the study by Garcia-Marques et al. (2017), supports a dissociation between conceptual and perceptual fluency. We appreciate the opportunity to clarify our position.

Our original wording may have unintentionally suggested that we considered this dissociation an artifact, which was not our intention. Rather, our aim was to point out that different fluency manipulations (e.g., repetition, font clarity, color contrast) all tend to increase perceived truth, they did it to varying degrees (e.g., Silva et al., 2016, 2017). However, it was possible that the differences in observed effect sizes could reflect differences in the strength of the subjective fluency experience they induce rather than fundamental differences between fluency types.

Upon reflection, we acknowledge that disentangling conceptual and perceptual fluency involves nuanced theoretical issues that are beyond the scope of our meta-analysis, which does

not empirically assess fluency subtypes. That said, we believe it remains important to provide a general overview of processing fluency as a mechanism underlying the illusory truth effect, both to contextualize the phenomenon and to interpret our findings through the lens of existing theoretical accounts. As such, we have revised the introduction to present a more general overview of processing fluency as a mechanism behind the illusory truth effect.

The manuscript now reads:

- Lines (58-67): “Previous research has proposed several theoretical frameworks to explain why repetition increases perceived truth of information³³. Dominant explanations of the illusory truth effect attribute a significant role to processing fluency, which is defined as the perceived ease with which information is processed³⁴. According to this view, repeated exposure to information increases processing fluency, which in turn increases the perceived truth. Supporting this framework, prior research has shown that the illusory truth effect can be reversed when participants form an implicit association between fluency and falseness³⁵. While fluency can be a legitimate indicator of truth when it aligns with valid cues^{36,37}, the illusory truth effect arises when people misattribute incidental increases in fluency from mere repetition as evidence of truth.”

13. **Comment:** 7. Line 344: I believe that there is a serious misunderstanding of the role of memory in the truth effect and the illusion of truth as it is assumed by the referential theory. Better memory is not expected to induce more “illusions” of truth. Instead, stronger memory integration is expected to have this effect only if the information is not already linked to a “validity” judgment in memory that can be retrieved to support the judgment. Also, there is no reason to assume that making a truth judgment implies better memory integration (making truth judgments is far from a synonymous of deep processing). An alternative interpretation to the one offered by the authors is that the effect is reduced in these cases because some participants rely on fluency to make the judgment, while others attempt to recover a previous judgment stored in memory or engaged in a different strategy. Therefore, I recommend that the authors avoid stating that their data “prove” the referential theory wrong. Instead, they could say: “If we assume that making a truth judgment involves a deep encoding process, then the results challenge the referential theory.”

Response: We thank the Reviewer for clarifying the interpretation of the referential theory and for noting where our earlier phrasing could have been misleading. We fully agree that stronger memory integration is only expected to increase the illusory truth effect when the information is not already linked to a stored validity judgment. We also agree that making a truth judgment does not necessarily imply deep processing (although prior work may have assumed this relationship; Dechêne et al., 2010).

In the revised manuscript, we have rephrased this paragraph to avoid implying that our data challenges the referential theory, and to acknowledge the role individual retrieval strategies. The revised section now reads:

- Lines (431-439): “From a referential theory perspective, truth judgments may create links between referents in memory whose valence (e.g., excitatory vs. inhibitory) depends on whether the information is encoded as true or false. In contrast, in irrelevant-task or passive-processing conditions, statements may be implicitly assumed to be true, leading to predominantly excitatory links. Furthermore, individual retrieval strategies may also influence the illusory truth effect: some participants may attempt to retrieve their previous judgment stored in memory, whereas others may primarily rely on fluency or adopt a different strategy.”

14. **Comment:** I wish you success with the review and publication of the article, hoping that it will help clarify the literature rather than contribute to further lack of clarity and confusion.

Response: We appreciate the Reviewer’s encouraging closing remarks and fully share the goal of ensuring that this work contributes to clarification of the literature. We have revised the manuscript to present our synthesis with greater precision, a clearer conceptual framework, and accurate representation of prior studies and theoretical perspectives. We hope that these changes will help the article serve as a clear and reliable reference for future research on the illusory truth effect.

Reviewer 4 Comments

Overall Comment: I have reviewed the revised manuscript and the authors' response to my comments. I appreciate the extensive revisions that have been undertaken. The authors have diligently addressed the major methodological concerns, and the manuscript has been improved considerably as a result.

In particular, I would like to highlight several key improvements:

- The updated literature search (which now includes studies up to early 2025) is a significant addition that enhances the relevance and comprehensiveness of the review.
- The implementation of a PEESE correction to address small-study effects and its use as a baseline for moderator analyses has increased the robustness of the reported findings.
- The discussion section has been strengthened by the inclusion of a more nuanced reflection on heterogeneity, including potential artifacts from standardization and multiplicity.

I have one final suggestion for a minor revision to improve the statistical accuracy of the reporting. It concerns the justification provided for using Borenstein's formula over Becker's for the effect size calculation. [...]

With this adjustment, the manuscript will be a robust and significant contribution to the literature. Hence, I recommend acceptance following this minor revision.

Response: We thank the Reviewer for his/her appreciation of our work and the suggestions to improve our methodology.

Standardized Mean Differences and Imputation of Missing Data

1. **Comment:** The manuscript states that Borenstein's method was chosen because its estimates were "more conservative". This is a terminological inaccuracy. Statistically, a wider confidence interval (as produced by Becker's formula in the authors' sensitivity analysis) reflects greater uncertainty and is therefore considered more conservative. A narrower confidence interval is more precise, but can also be considered more liberal if that precision is an artifact of imputation, as may be the case here. The greater precision of the Borenstein estimator is likely a consequence of imputing missing correlations, which may lead to an overconfident (i.e., artificially narrow) estimate of the uncertainty.
Recommendation:

I suggest revising the justification on lines 820-821 to transparently acknowledge this trade-off and that the greater precision is contingent on the assumptions made during the

imputation of missing correlation coefficients. This would clarify the rationale while also transparently acknowledging the potential limitations of the chosen methods.

Response: We thank the reviewer for this important clarification regarding the use of the term "conservative" and the implications of imputing missing correlations. We agree that describing the Borenstein method as "more conservative" was imprecise and have revised the justification to better reflect the statistical trade-off involved. The revised text now acknowledges the limitations and motivates the choice based on interpretability and convergence properties as follows:

- Lines (860-869): "We also conducted a sensitivity analysis using Becker's formula to compute the standardized mean difference, which only requires standard deviations from the first assessment and therefore avoids the need to impute correlations between repeated measures ⁷⁸. This approach yielded effect sizes close to those from Borenstein's formula but with slightly wider confidence intervals, reflecting greater uncertainty (see Supplementary Table 5). We selected Borenstein's method for the main analyses because it yielded more stable estimates across studies, which facilitates interpretation and comparability of effect sizes in the synthesis. However, we acknowledge that its greater precision is contingent on the imputation of missing correlations, which may lead to potentially overconfident uncertainty estimates."

Systematic review and meta-analysis of the evidence for an illusory truth effect
and its determinants

Response to Reviewers

We thank the Reviewers for their evaluation of our manuscript and for their constructive feedback. All modifications in the manuscript have been highlighted in yellow color.

Detailed responses to each point raised by Reviewers are provided below.

Reviewer 2 Comments

Overall Comment: I was Reviewer 2 on the last round of reviews. The authors have done an excellent job addressing my concerns, and I applaud their attention to detail when making these changes. I have just a few minor points below, but I believe this manuscript is now ready for publication without another full round of review. Congratulations on this work!

Response: We thank the Reviewer for their positive and encouraging assessment of our revised manuscript. We are pleased that we were able to address their previous concerns and appreciate their constructive feedback throughout the review process.

New veracity cues moderators:

- Comment:** The authors find that the effect size for providing false cues is $g = -0.18$, 95% CI $[-0.33, -0.03]$. The exploratory analysis by type of cue also found that most cues had confidence intervals below zero. This means that the illusory truth effect was actually reversed, with repeated claims now being judged as less true than new claims. In discussing these effects, however, this reversal is not mentioned, and rather it is just reported that the size of the truth effect was decreased with false cues (e.g., from the discussion “False cues reduced the impact of repeated exposure on perceived truth while true cues did not affect the illusory truth effect”). This type of wording (the false cues decreased the size of the illusory truth effect) implies to me that there is still an illusory truth effect with false cues. I think it would be important and informative when presenting results like this to clearly communicate that it’s not just that the size of the illusory truth effect is smaller, but also that it’s reversed.

Response: We thank the Reviewer for this comment. We agree that our previous wording may underplay the effect reversal observed in the subgroup analysis. While these subgroup estimates are descriptive and not adjusted for the full set of moderators included in the meta-regression, a negative estimate with confidence intervals entirely below zero indeed indicates that repeated statements were judged as less true than new statements on average in studies implementing false cues. We have therefore revised the Discussion to explicitly state that false cues not only reduce but reverse the illusory truth effect at the subgroup level as follows:

- Lines (453-460): “In particular, false cues not only attenuated but, on average, reversed the illusory truth effect, such that repeated statements were judged as less true than new

statements. In contrast, true cues did not reliably alter the magnitude of the illusory truth effect. This pattern suggests that explicit falsity cues can override the influence of repetition, providing further support for the potential effectiveness of debunking interventions. However, because this reversal is observed in subgroup estimates that are not adjusted for the full set of moderators, it should be interpreted as descriptive of existing study patterns.”

2. **Comment:** I didn't check all the observed u that were added, but noticed that for false cues, the authors reported observed $u = 40$ in the results and observed $u = 41$ in the methods. Then, if you add up the observed u for the different types of false cues, they only add up to 37. The true cues, on the other hand, are reported as observed $u = 33$ in the results and observed $u = 32$ in the methods, but if you add up the true cues from the table, those equal 36. There seem to be either typos or coding issues going on here.

Response: We thank the Reviewer for carefully noting these inconsistencies and apologize for the oversight. We would like to clarify that the inconsistency in the Methods section originates from the same coding error in the veracity-cue moderator that we identified and reported during the previous round of peer review. As stated in our earlier response, this coding error was corrected in the dataset and all analyses, and it did not affect the pattern, magnitude, or interpretation of any results.

In addition, the Reviewer refers to a discrepancy in the subgroup analysis by cue type reported in the Supplementary Information. This was due to a typographical swap in the counts for epistemic qualifiers (true vs. false cues), which has now been corrected (false cues = 10; true cues = 7).

We have now harmonized the reported counts across the Methods, Results, and Supplementary Information.

3. **Comment:** In the abstract – saying “since 2006” in the abstract without context (that the previous meta looked for effects up to 2006) may be confusing to readers. As a very minor note, but in case it's not caught in copy editing, the effect size $g = .37$ is missing a “0” in front of the decimal.

Response: We thank the reviewer for noting these points. We have clarified the reference to “since 2006” in the abstract by explicitly indicating that it refers to the publication date of the

previous meta-analysis. We have also corrected the effect size notation to include a leading zero ($g = 0.37$). The revised abstract now reads:

- Lines (4-9): “In light of a growing number of studies published since the previous meta-analysis in 2006 and concern of publishing biases, we conduct a meta-analysis on 182 studies and 366 effect sizes ($N = 31,184$ participants) published from 1977 to 2025. After correcting for small-study effects, we observe a small illusory truth effect ($g = 0.37$, 95% confidence interval $[0.30, 0.44]$), with a substantial within and between-study heterogeneity.”

4. **Comment:** Minor wording suggestion: On p. 5, line 87 the authors write - “In particular, the illusory truth effect seems to be different” – I think saying “smaller” instead of “different” would work better here.

Response: We thank the Reviewer for this helpful suggestion. We have replaced “different” with “smaller” to more precisely reflect the quantitative comparison between within-item and between-item effect sizes.

Reviewer 4 Comments

Overall Comment: I thank the authors for their thorough revision. My queries have been satisfactorily addressed.

Response: We thank the Reviewer for their positive assessment and are pleased that our revisions satisfactorily addressed their concerns.